# Entorhinal grid-like codes for visual space during memory formation

Luise P. Graichen [1] ✉, Magdalena S. Linder[1], Lars Keuter [1,2], Ole Jensen [3], Christian F. Doeller [4,5], Claus Lamm [1], Tobias Staudigl [6,7,10] & Isabella C. Wagner [1,7,8,9,10] ✉

Eye movements, such as saccades, allow us to gather information about the environment and, in this way, can shape memory. In non-human primates, saccades are associated with the activity of grid cells in the entorhinal cortex. Grid cells are essential for spatial navigation, but whether saccade-based grid-like signals play a role in human memory formation is currently unclear. Here, human participants undergo functional magnetic resonance imaging and continuous eye gaze monitoring while studying scene images. Recognition memory is probed immediately thereafter. Results reveal saccade-based grid-like codes in the left entorhinal cortex that are specific to later remembered trials during study, a finding that we replicate with an independent data set. The grid-related effects are time-locked to activation increases in the frontal eye fields. Unexpectedly, lower saccade-based grid-like codes are associated with better subsequent recognition memory performance. Our findings suggest an entorhinal map of visual space that is timed with neural activity in oculomotor regions, and negatively associated with subsequent memory. Grid-like codes, entorhinal cortex, saccades, frontal eye fields (FEF), memory, functional magnetic resonance imaging (fMRI)

Humans move their eyes to gather information about the environment. During natural viewing, eye movements, such as saccades, help to shift attention to the relevant features of a visual scene and can thereby shape memory formation[1–5]. Saccades are associated with the activity of grid cells in the entorhinal cortex, which are known to play a role in spatial navigation[6,7]. Grid cells express multiple firing fields that are hexagonally arranged[8]. Killian and colleagues discovered visual grid cells in the entorhinal cortex of non-human primates that responded to multiple gaze positions[9] and saccade directions[10]. In

humans, it has been suggested that the firing properties of grid cell populations might relate to so-called "grid-like representations" or "grid-like codes", which can be obtained non-invasively with functional magnetic resonance imaging (fMRI) and correspond to the strength of the hexadirectional modulation of the fMRI signal[11,12]. Grid-like codes were linked to saccades during free visual search[6] and controlled visual tracking of a moving target[7]. Staudigl and colleagues employed magnetoencephalography (MEG) and reported a grid-like modulation of broadband high-frequency activity that was linked to saccades as

[1]Department of Cognition, Emotion, and Methods in Psychology, Faculty of Psychology, University of Vienna, Vienna, Austria. [2]University Medical Center Hamburg-Eppendorf, Institute of Systems Neuroscience, Hamburg, Germany. [3]Wellcome Centre for Integrative Neuroscience, Oxford Centre for Human Brain Activity, Department of Psychiatry, University of Oxford, Oxford, United Kingdom. [4]Department of Psychology, Max Planck Institute for Human Cognitive and Brain Sciences, Leipzig, Germany. [5]Kavli Institute for Systems Neuroscience, Centre for Neural Computation, The Egil and Pauline Braathen and Fred Kavli Centre for Cortical Microcircuits, Jebsen Centre for Alzheimer's Disease, Norwegian University of Science and Technology, Trondheim, Norway. [6]Department of Psychology, Ludwig-Maximilians-Universität München, Munich, Germany. [7]Donders Institute for Brain, Cognition, and Behaviour, Radboud University, Nijmegen, The Netherlands. [8]Vienna Cognitive Science Hub, University of Vienna, Vienna, Austria. [9]Centre for Microbiology and Environmental Systems Science, University of Vienna, Vienna, Austria. [10]These authors contributed equally: Tobias Staudigl, Isabella C. Wagner. ✉e-mail: luise.philine.graichen@univie.ac.at; isabella.wagner@univie.ac.at

individuals viewed scene images[13]. Grid cells are thought to provide an internal spatial map of the current surroundings[14] that was shown to relate to spatial memory performance[11,15–17]. Similarly, visual grid cells may provide us with a mental map to organize the spatial relationships of visually presented content. By encoding translational vectors between salient stimulus features, they may guide memory formation[1,18]. However, an effect of visual grid cells on memory has so far only been shown in non-human primates, where saccade-related grid cell activity revealed neural adaption (i.e., decreased activity) upon stimulus repetition[9,10], serving as an index for memory[19,20]. This leaves open the question whether saccade-based grid-like codes in the human entorhinal cortex are relevant to memory formation.

The entorhinal cortex provides the major input into the hippocampus and is structurally connected to adjacent medial temporal areas[21]. In humans, it connects to the parahippocampal cortex, which is engaged in visual scene processing[22]. In rodents and non-human primates, the entorhinal cortex receives input from primary and secondary visual areas[23,24]. Brain regions associated with vision and oculomotion, such as the visual cortex and the frontal eye fields (FEF), play a central role in generating and coordinating saccades[25,26]. Recent studies revealed synchronized grid-like codes in human entorhinal and ventromedial prefrontal cortices that shared a similar grid orientation[27], as well as entorhinal-neocortical connectivity that was modulated by the magnitude of entorhinal grid-like codes[17]. These findings indicate that the entorhinal cortex might serve as a hub region, orchestrating information flow within entorhinal-neocortical networks[28,29]. However, how saccades and associated grid-like codes are integrated into this interregional dialogue remains to be elucidated.

To tackle these open questions, we examined data from two independent studies that were performed at different measurement sites (thus, yielding a "Donders" and a "Vienna" data set with $N = 48$ and $N = 50$, respectively). In both studies, human participants viewed scene images while we tracked their eye movements and measured fMRI. Immediately thereafter, participants completed a recognition memory task and discriminated previously studied ("old") from novel scenes (Fig. 1B). The Vienna study also included a delayed test session in the behavioral laboratory after one week. We hypothesized that saccades during scene viewing should be coupled to grid-like codes in the entorhinal cortex and that the grid-related signals should be linked to individual variations in recognition memory performance. We further expected saccade-based grid-like codes in the entorhinal cortex to be tied to the activity in oculomotor regions that are known to be involved in saccade generation and coordination.

## Results

### Recognition memory performance and saccades

In both the Donders and Vienna study, participants completed a recognition memory task (Fig. 1B) during which they studied scene images (study period) and were asked to discriminate previously viewed ("old") from novel ("new") scenes immediately after study (test period). Participants of the Donders study recognized more than two-thirds of the scene images correctly (~70% of 200 scenes, mean ± SEM, $140.91 ± 5.62$) and performed, on average, 7.58 (± 0.40) saccades per scene. To examine how saccade-based grid-like codes related to recognition memory, we first assessed the link between eye movements and memory formation. Consistent with previous literature on the role of eye movements in memory encoding[4,30–32], we found that the average saccade numbers during later remembered scenes ($7.95 ± 0.40$) were significantly higher than during later forgotten ones ($6.64 ± 0.43$; paired-sample $t$-test, $N = 32$; $t(31) = 8.262$, Cohen's $d = 1.46$, 95% confidence interval (CI) = [0.99, 1.63], $p_{two-tailed} < 0.0001$). In line with these results, we observed a positive correlation between individual recognition memory performance ($d$-prime, $1.82 ± 0.11$; Fig. 1C, left panel) and saccade numbers during

later remembered scenes ($N = 32$, $r_{Pearson} = 0.51$, 95% CI = [0.20, 0.73], $p_{two-tailed} = 0.003$; Fig. 1D). Thus, participants who used more saccades to explore later remembered scenes performed better in distinguishing old from novel material during the subsequent recognition memory test.

Similar to the Donders study, participants of the Vienna study recognized more than two-thirds of the scene images correctly (89% of 96 scenes, mean ± SEM, $85.91 ± 1.47$) and produced, on average, 4.69 (± 0.14) saccades per scene. Again, average saccade numbers during later remembered scenes ($4.74 ± 0.14$) differed significantly from later forgotten ones ($4.31 ± 0.23$; paired-sample $t$-test, $N = 46$; $t(42) = 2.796$, $d = 0.43$, 95% CI = [0.13, 0.78], $p_{two-tailed} = 0.008$), and were significantly associated with individual recognition memory performance ($d$-prime, $N = 46$, $r_{Pearson} = 0.46$, 95% CI = [0.20, 0.66], $p_{two-tailed} = 0.001$; Supplementary Results S1, Supplementary Fig. S1). Overall, participants of the Vienna study showed high recognition memory performance ($d$-prime, $3.24 ± 0.13$; Fig. 1C, right panel), significantly better compared to participants of the Donders study (Donders: $N = 32$, Vienna: $N = 46$; Welch two-sample $t$-test, $t(75.91) = −8.594$, $d = 1.87$, 95% CI = [−1.75, −1.09], $p_{two-tailed} < 0.0001$). This likely reflects that participants of the Vienna study viewed fewer scenes, were not faced with an additional (cover) task during encoding, and completed no distractor task between study and test periods (s. Methods).

### Saccades are associated with grid-like codes in the entorhinal cortex

Next, we tested whether saccades were linked to entorhinal grid-like codes as participants viewed scenes during the study period. Based on previous findings in humans[6,7,13] and non-human primates[9], we expected significantly increased grid-like codes (i.e., testing for a 6-fold symmetrical modulation of the fMRI signal) in the entorhinal cortex time-locked to saccades. We turned toward the fMRI data and, for each participant, analyzed the saccades that occurred during scene viewing with respect to their directional angle. We then split the data of each individual into independent data halves (a) to estimate the individual, saccade-based grid orientation (i.e., the phase of the hexadirectional fMRI signal) in the first half of the data, and (b) to test the estimated grid orientation in the second half of the data (Fig. 2A, and see Methods section for details). This allowed us to quantify the amount of saccade-based grid-like codes, such that a closer alignment between saccade direction and individual grid orientation should correspond to a stronger hexadirectional fMRI signal.

In the Donders study data, we found significantly increased grid-like codes in the bilateral entorhinal cortex, our predefined region-of-interest (ROI), for the 6-fold symmetrical model that were time-locked to saccades while participants studied the scene images ($N = 29$; bilateral entorhinal cortex: mean ± SEM, $0.084 ± 0.029$, Wilcoxon test, $V = 333.5$, $p_{one-tailed} = 0.006$, $d = 0.53$; Fig. 2B, left panel). If this effect was related to grid-like codes, we reasoned that it should not be present for different symmetrical models (testing for a 4-, 5-, 7-, or 8-fold symmetry). Indeed, when repeating the analysis with these different symmetries, results showed no evidence for significantly increased grid-like codes in the entorhinal cortex (Fig. 2C). We also tested whether grid-like codes were present in other areas known to be involved in memory and visuo-oculomotor processing (such as the hippocampus, anterior thalamus, FEF, and visual cortex). There were no significantly increased grid-like codes in any of these regions (Supplementary Results S2, Supplementary Fig. S2), corroborating that our findings were specific to the entorhinal cortex, in which grid cells were previously reported[11]. Additional follow-up analysis showed that the effect was more prominent in the left than in the right entorhinal cortex ($N = 29$; left entorhinal cortex: mean ± SEM, $0.068 ± 0.034$, Wilcoxon test, $V = 300$, $p_{one-tailed} = 0.038$, $d = 0.37$; right entorhinal cortex: $0.051 ± 0.031$, Wilcoxon test, $V = 275$, $p_{one-tailed} = 0.052$, $d = 0.30$; Fig. 2B, right panel).

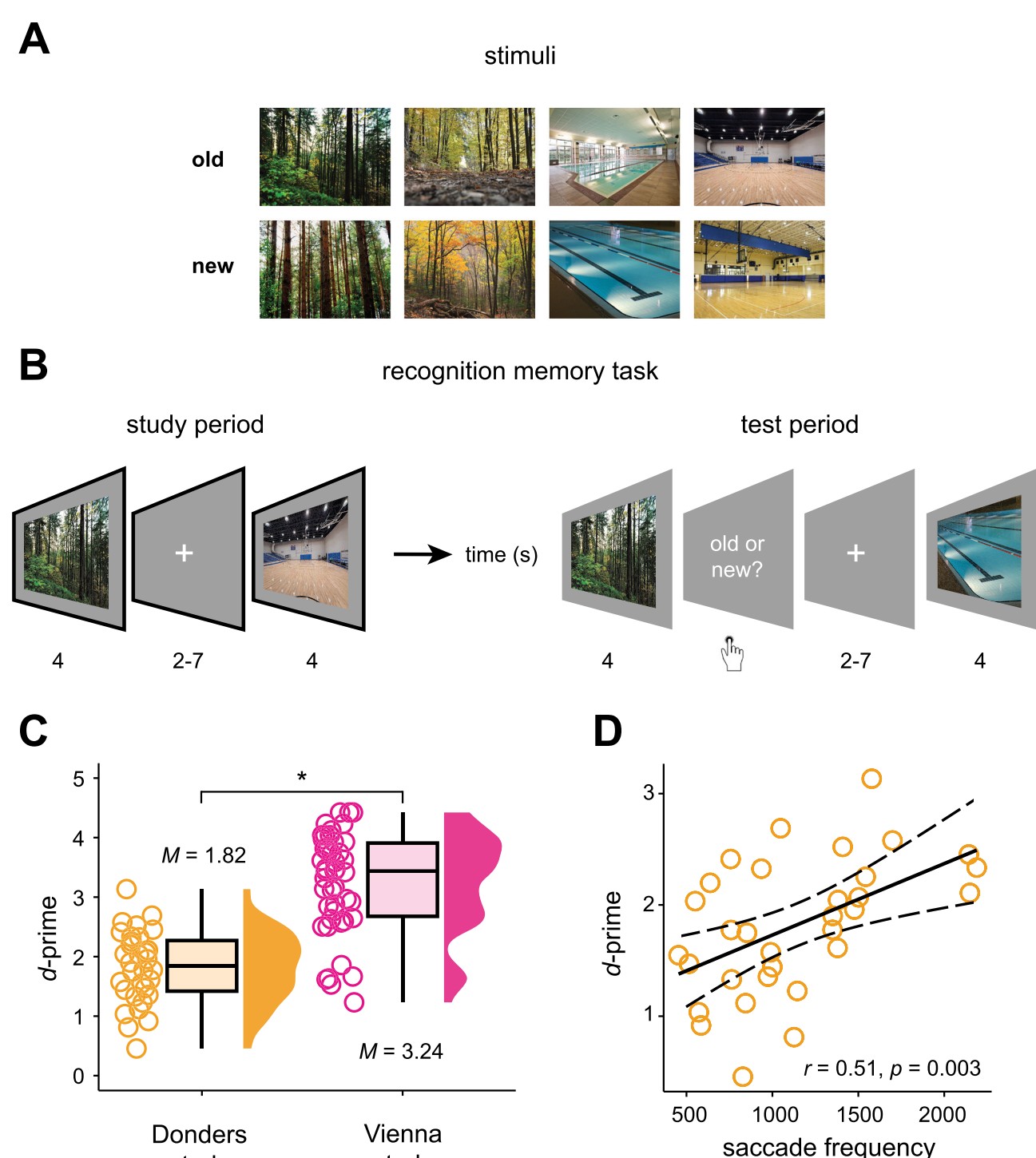

**Fig. 1 | Recognition memory performance and saccades. A** Examples of scenes that participants viewed during the study period ("old" scenes) and during the test period, together with novel ones ("new" scenes), are illustrated here with Creative Commons licensed images, as the original study stimuli cannot be published. **B** Recognition memory task used in the Donders and Vienna study. Note that in the Vienna study, participants viewed each scene for 3 s (study period) and could provide their answer for 2 s (test period). The Vienna study also incorporated an additional test period that was performed in the behavioral lab after one week (delayed test, not depicted in the figure). **C** Data points show the participant-specific d-prime values for both the Donders (orange, $N = 46$) and the Vienna study (pink, $N = 32$), and boxplots show the median (upper and lower borders mark the interquartile range, whiskers show minimum and maximum non-outlier values). The performance discrepancy between Donders and Vienna studies (Welch two-sample t-test, two-tailed, $p < 0.0001$) likely arose from the Vienna study including fewer scenes, no additional task during encoding, and no distractor task between study and test (s. Methods). **D** The scatter plot shows the Pearson correlation between the total number of saccades per participant (saccade frequency) and d-prime for the Donders study (two-tailed, $N = 32$, $p = 0.003$; for the Vienna study, see Supplementary Results S1, Supplementary Fig. S1A). The solid line indicates the regression fit, and the dashed lines mark the confidence interval (95% CI). *Significant at $p < 0.05$. Source data are provided as a Source Data file.

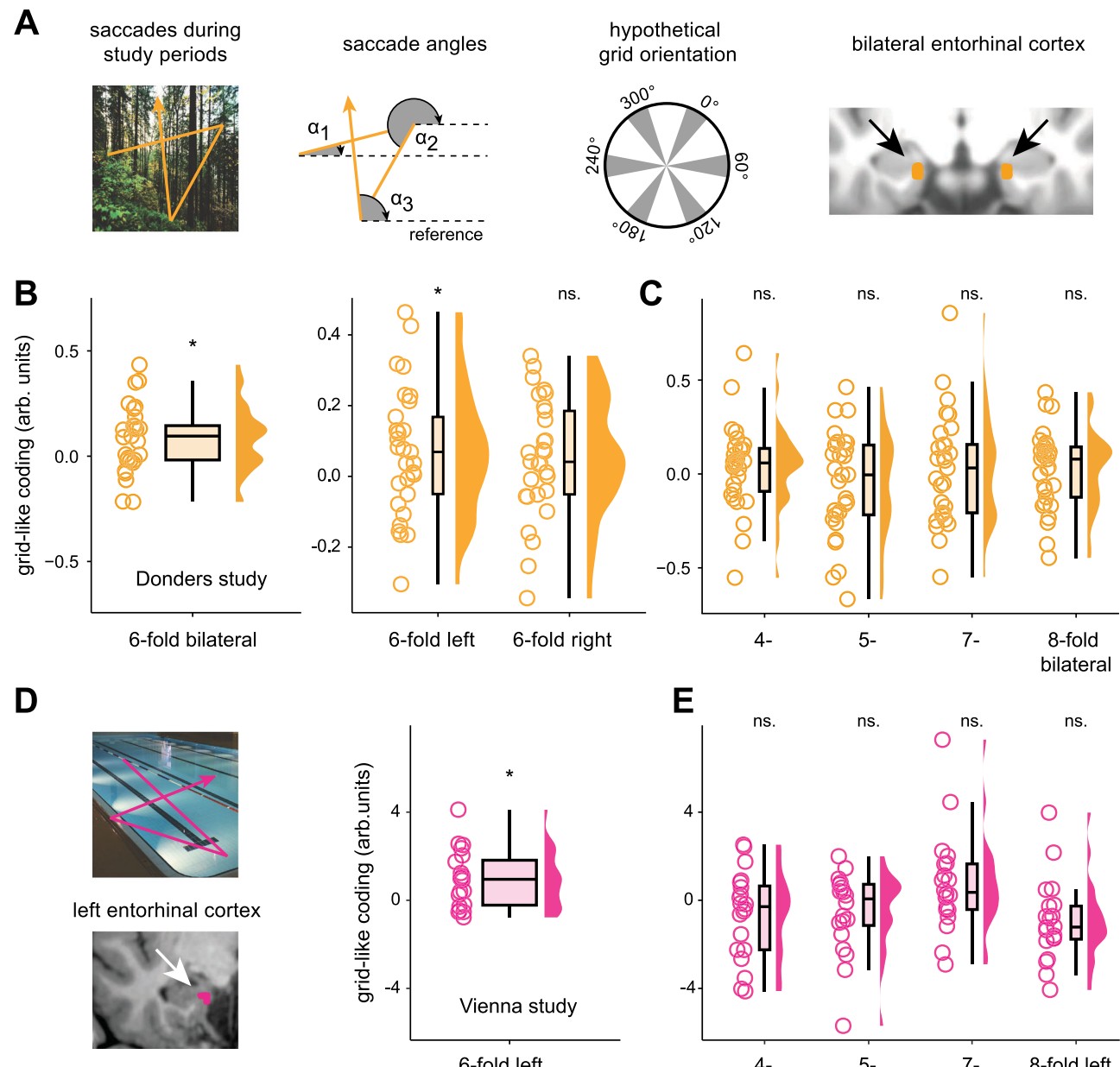

**Fig. 2 | Saccades associated with grid-like codes in the entorhinal cortex.**
**A** Donders study (orange), from left to right: Schematic saccade trajectory during the study period (exemplarily overlaid onto a scene image). Saccade directional angles referenced to an arbitrary screen point (dashed lines, angles α1-α3 exemplarily shown in gray). Hypothetical grid orientation in 360° space with the main grid axes in gray. Bilateral entorhinal cortex region-of-interest (ROI, in orange) projected onto the normalized T1-weighted structural image. We expected increased bilateral entorhinal cortex signal for saccades aligned to individual grid orientations. **B** left panel: Magnitude of 6-fold grid-like codes in the bilateral entorhinal cortex (arb. units = arbitrary units; $N = 29$, $p = 0.006$). **B** right panel: Magnitude of 6-fold grid-like codes in left and right entorhinal cortices ($N = 29$, left: $p = 0.038$, right: $p = 0.052$). **C** Grid magnitudes for control symmetries (4-, 5-, 7-, and 8-fold periodicities, bilateral entorhinal cortex, $N = 29$, $p > 0.05$). **D** Vienna

study (pink), left panel: Schematic saccade trajectory during study periods. Left entorhinal cortex ROI (in pink) superimposed on the T1-weighted structural image of one participant (grid-like codes for this study were analyzed in subject-native space). **D** right panel: Magnitude of 6-fold grid-like codes in the left entorhinal cortex ($N = 20$, $p = 0.003$). **E**: Grid magnitudes for control symmetries (4-, 5-, 7-, and 8-fold periodicities, left entorhinal cortex, $N = 20$, $p > 0.05$). Data points show individual grid magnitudes during study periods, and boxplots show the median (upper and lower borders mark the interquartile range, whiskers show minimum and maximum non-outlier values). Statistical analysis was performed using one-tailed Wilcoxon tests. Unadjusted $p$-values are reported, with effects below $p < 0.01$ meeting the Bonferroni-corrected significance criterion (s. Methods). *Significant at $p < 0.05$; ns. not significant. Source data are provided as a Source Data file.

We performed additional control analyses to rule out alternative explanations for our findings. First, we repeated the main analysis but reversed estimation and test data halves (thus, estimating individual grid orientations on the second and testing grid-like codes in the first data half). This gave virtually identical results, highlighting that our findings were independent of the specific data partitioning scheme ($N = 29$; bilateral entorhinal cortex: mean ± SEM, $0.069 ± 0.029$,

Wilcoxon test, $V = 340$, $p_{one-tailed} = 0.004$, $d = 0.45$; Supplementary Results S3A, Supplementary Fig. S3A). Second, to show that grid-like codes were not linked to the number of saccades that participants made, we correlated the average saccade numbers across all trials with the individual magnitude of entorhinal grid-like codes. Results confirmed that grid-like code magnitude was not related to saccade numbers across participants ($N = 29$; bilateral entorhinal

cortex: $p_{two\text{-}tailed} = 0.386$). Third, the estimation of grid orientations can be biased by the saccade durations (in ms) along the different saccade directions on the computer screen (individual grid orientations for a specific saccade direction cannot be estimated if the participant never made saccades in that direction). We found a significant bias in saccade durations across directional bins in 8 out of 29 participants (all $p < 0.05$). For this subset of participants, we randomly excluded 10% of saccades in an iterative process until we could ensure an even distribution of saccade durations across the different saccade directions (all $p > 0.05$), and repeated the main analysis. This confirmed our result of increased saccade-based grid-like codes in the entorhinal cortex (Supplementary Results S3B, Supplementary Fig. S3B). Fourth, we checked whether significantly increased grid-like codes were tied to an overall increase in the entorhinal cortex BOLD signal that could reflect individual differences in the signal-to-noise ratio of the fMRI data (thus, answering the question of whether stronger grid-like codes would be linked to a higher overall BOLD signal in the entorhinal cortex), but this was not the case ($p = 0.083$, Supplementary Results S4, Supplementary Fig. S4).

Building on the abovementioned findings from the Donders study, we leveraged the data from the Vienna study to test for saccade-based grid-like codes in this independent participant sample. Since grid-like codes were more prominent in the left entorhinal cortex (Fig. 2B, right panel), we focused the analysis on the left and right entorhinal cortices separately (we only performed this analysis on a subset of 20 participants for which the entorhinal cortex masks comprised at least 14 voxels to match the entorhinal cortex mask size of the Donders study, see Methods section for details, and as we have done previously[17]). Once again, we detected significant saccade-based grid-like codes in the left but not in the right entorhinal cortex while participants studied scene images ($N = 20$; left entorhinal cortex: mean ± SEM, $0.942 ± 0.291$, Wilcoxon test, $V = 176$, $p_{one\text{-}tailed} = 0.003$, $d = 0.72$; right entorhinal cortex: mean ± SEM, $0.595 ± 0.608$, Wilcoxon test, $V = 130$, $p_{one\text{-}tailed} = 0.184$, $d = 0.22$; Fig. 2D). Grid-like codes were not significant for any of the control symmetries (Fig. 2E). In summary, across two independent data sets, we found significantly increased saccade-based grid-like codes in the left entorhinal cortex while participants studied scene images.

## Saccade-based grid-like codes in the entorhinal cortex are lower at better recognition memory

Our main goal was to clarify the role of saccade-based grid-like codes in memory formation. While hexadirectional modulation of the fMRI signal in the entorhinal cortex is a well-established finding (e.g., Doeller et al.[11]) and has been observed in relation to saccade activity during free visual search[6,7], a direct link to human memory formation is missing. Previous studies reported mixed results regarding the relationship between entorhinal grid-like codes and behavior. In short, increased grid-like codes were associated with better spatial navigation performance of human participants in an object-location memory task (i.e., lower drop error when placing an object at its correct location[11,15,16]). When observing a virtual demonstrator, increased grid-like codes were associated with lower navigation performance[17], and a similarly negative relationship was found for directional coding in the human medial temporal lobe[33]. Saccade-based grid-like codes were shown to be positively associated with self-reported navigation ability[6]. In non-human primates, only some of the detected visual grid cells showed neural adaption (i.e., decreased activation) upon stimulus repetition[4,9], leaving it open whether saccade-based entorhinal grid-like codes are related to human memory formation as well.

Using data from the Donders study, we tested whether individual variations in the magnitude of saccade-based grid-like codes during study periods would scale with individual differences in recognition memory performance (as indexed by $d$-prime). Results showed a significantly negative association between grid-like codes

and $d$-prime values ($N = 29$, $r_{Pearson} = -0.51$, 95% CI = $[-0.741, -0.182]$, $p_{two\text{-}tailed} = 0.004$; Fig. 3A). In other words, lower saccade-based grid-like codes were correlated with better recognition memory performance across participants. Several control analyses confirmed this brain-behavior relationship. First, we could show that the result did not stem from specific saccade patterns. Participants with better recognition memory performance (higher $d$-prime) might have made shorter saccades, potentially causing lower grid-like codes, but this was not the case ($p = 0.717$; Supplementary Results S5, Supplementary Fig. S5A; for a more detailed analysis of the relationship between entorhinal grid-like codes, saccade characteristics, and visual exploration see also Supplementary Results S6 & S7, Supplementary Figs. S6, S7). Second, the result could have been driven by differences in entorhinal BOLD activation. Participants with higher $d$-prime could have shown lower BOLD changes during the study period, as better memory can be associated with activation decreases in the involved areas[34], resulting in lower grid-like codes. However, this was also not the case ($p = 0.094$; Supplementary Results S5, Supplementary Fig. S5B).

We repeated the same analysis with data from the Vienna study, for which recognition memory performance was measured twice, immediately after the study period (immediate test) and one week later (delayed test). For the immediate test, we observed no significant association between grid-like codes in the left entorhinal cortex and recognition memory performance ($d$-prime; $N = 20$, $r_{Pearson} = 0.01$, 95% CI = $[-0.437, 0.448]$, $p_{two\text{-}tailed} = 0.98$; Fig. 3D). A similar picture emerged for the delayed test ($d$-prime; $N = 20$, $r_{Pearson} = -0.04$, 95% CI = $[-0.4714, 0.4127]$, $p_{two\text{-}tailed} = 0.88$; see Supplementary Results S8 for an analysis of grid-like codes in relation to memory durability). Once again, we suspect that this might be due to differences in task design between the two studies (in the Vienna study, participants viewed fewer scenes and were not required to solve an additional task between study and test periods).

While $d$-prime is a well-established and sensitive measure of recognition memory performance[32], it includes false alarms to novel stimuli that were not part of the study period. To better align the behavioral metric with our neural data (specifically reflecting scene encoding), we correlated saccade-based grid-like codes with individual hit rates (restricted to scenes that were actually viewed during the study periods). Results were virtually identical (Donders: $N = 29$, $r_{Pearson} = -0.49$, 95% CI = $[-0.726, -0.150]$, $p_{two\text{-}tailed} = 0.0007$; Vienna: $N = 20$, $r_{Pearson} = 0.001$, 95% CI = $[-0.442, 0.443]$, $p_{two\text{-}tailed} = 0.998$; Supplementary Fig. S10), highlighting the consistency between the two behavioral measures. Overall, the results of the Donders data set suggest a potential link between lower saccade-based grid-like codes and successful memory formation.

## Saccade-based entorhinal grid-like codes are specific to successful memory encoding

To examine potential memory-related differences in grid-like codes during the study period, we quantified grid-like codes separately for subsequently remembered and forgotten scenes (hits vs. misses). We reasoned that if saccade-based entorhinal grid-like codes were involved in successful memory encoding, they should be present during hits rather than misses. Consistent with our behavioral findings regarding $d$-prime, we observed significantly more hits than misses among participants of both studies (mean ± SEM; Donders study: $N = 32$; $141 ± 6$ hits, 70.5%, $59 ± 6$ misses, 29.5%; Vienna study: $N = 46$; $86 ± 1.5$ hits, 89%, $10 ± 1.5$ misses, 11%), and more hits in the Vienna than in the Donders study (Welch two-sample $t$-test, $t(49.14) = 5.95$, $d = 1.47$, 95% CI $[0.126, 0.255]$, $p_{two\text{-}tailed} < 0.0001$). Identical to our main analysis, we focused on the left entorhinal cortex and split the data into equal halves. To ensure a reliable estimation of individual grid orientations, the estimation step was based on saccade trajectories from all trials (both hits and misses) in the first data half. This was a necessary choice due to the

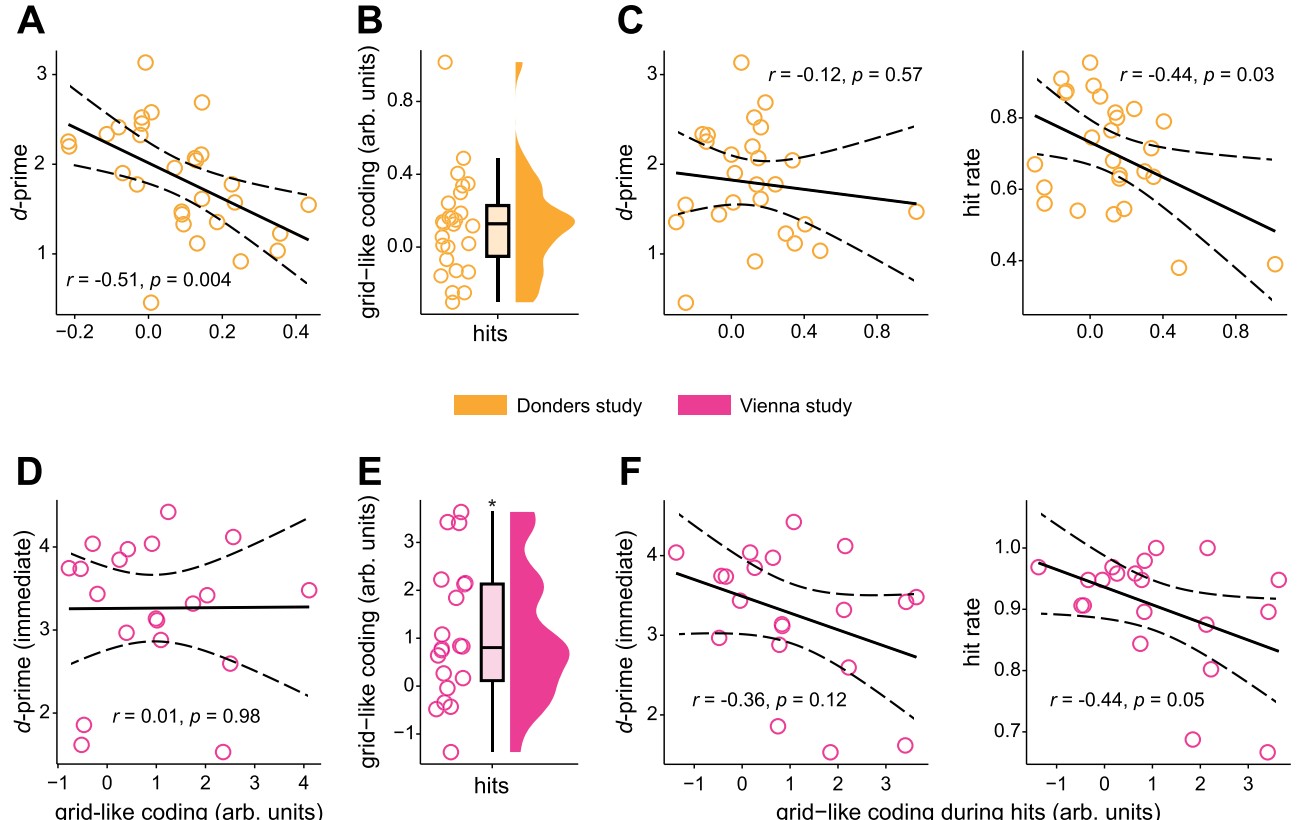

**Fig. 3 | Relationship between saccade-based grid-like codes and memory performance.** Donders study (in orange), **A** The scatter plot displays the Pearson correlation between the magnitude of grid-like codes (arb. units = arbitrary units) and individual recognition memory performance (*d*-prime; two-tailed, $N = 29$, $p = 0.004$). **B** Data points show individual grid magnitudes in the left entorhinal cortex during study trials that were subsequently remembered (6-fold, hits, $N = 26$, Wilcoxon test, one-tailed, $p = 0.023$), and boxplots show the median (upper and lower borders mark the interquartile range, whiskers show minimum and maximum non-outlier values). **C** The scatter plots show Pearson correlations (two-tailed, $N = 26$) between grid-like codes during hits and recognition memory performance (*d*-prime and hit rate in left and right panels, respectively). Vienna study (in pink), **D** The scatter plot displays the Pearson correlation between the magnitude of grid-like codes and individual recognition memory performance (*d*-prime; two-tailed, $N = 20$, $p = 0.98$). **E** Data points show individual grid magnitudes in the left entorhinal cortex during hits (6-fold, $N = 20$, Wilcoxon test, one-tailed, $p = 0.002$), and boxplots show the median (upper and lower borders mark the interquartile range, whiskers show minimum and maximum non-outlier values). **F** The scatter plots show Pearson correlations between grid-like codes during hits and recognition memory performance (*d*-prime and hit rate, two-tailed, $N = 20$). The solid line indicates the regression fit, and the dashed lines mark the confidence interval (95% CI). *Significant at $p < 0.05$. Source data are provided as a Source Data file.

relatively low number of miss trials, particularly in the Vienna study, which would preclude the reliable estimation of grid orientations from miss trials alone. We then estimated individual grid orientations using a General Linear Model (GLM1, same model structure as in the main analysis). Using the second data half, we tested the grid orientations' fit on saccades during hit trials, thereby quantifying the magnitude of grid-like codes during hits (GLM2). The same procedure was repeated for miss trials.

As expected, in the Donders study, grid-like codes in the left entorhinal cortex were significantly increased during hits ($p_{one\text{-}tailed} = 0.023$, Fig. 3B), but not during misses ($p_{one\text{-}tailed} = 0.26$; Supplementary Table S1; note that grid-like codes were not significantly different between hits vs. misses, paired-sample *t*-test, $N = 26$, $t(25) = 0.47$, $d = 0.09$, 95% CI = [−0.23, 0.37], $p_{two\text{-}tailed} = 0.64$). We observed the same results pattern in the Vienna study, revealing that grid-like codes in the left entorhinal cortex were significantly increased during hits ($p_{one\text{-}tailed} = 0.002$, Fig. 3E) but not during misses ($p_{one\text{-}tailed} = 0.75$; Supplementary Table S1; again grid-like codes were not significantly different between hits vs. misses, $N = 9$, Wilcoxon signed-rank test, $V = 36$, $p = 0.13$, $d = 0.39$). Thus, across two independent data sets, we found significantly increased saccade-based grid-like codes in the entorhinal cortex that appeared specific to successful memory encoding.

## Saccade-based entorhinal grid-like codes during successful memory encoding are lower at better recognition memory

To examine the brain-behavior relationship specifically for successful memory encoding, we assessed whether the magnitude of grid-like codes during hits was associated with individual variations in recognition memory performance (*d*-prime). In both studies, grid-like codes tended to be higher in participants with lower *d*-prime, but the results were not significant (all $p > 0.05$; Fig. 3C, F, left panels, Supplementary Table S2). As above, we also interrogated the correlation between grid-like codes and hit rate. We observed a significantly negative association in the Donders study ($N = 26$, $r_{Pearson} = −0.44$, CI = [−0.70, −0.06], $p_{two\text{-}tailed} = 0.026$, Fig. 3C, right panel) and a similar (albeit non-significant trend) in the Vienna study ($N = 20$, $r_{Pearson} = −0.44$, CI = [−0.74, 0.01], $p_{two\text{-}tailed} = 0.055$; Fig. 3F, right panel, Supplementary Table S3). No significant correlations were found between grid-like codes during miss trials and behavioral performance (Supplementary Tables S2, S3).

## Saccade-based entorhinal grid-like codes are time-locked to neural activity in the frontal eye fields

Saccade generation and coordination are associated with neural activity in a set of brain regions that appear engaged in visual processing and oculomotion, including the visual cortex and the frontal eye fields (FEF)[35,36]. Consequently, we reasoned that saccade-based

**Fig. 4 | Saccade-based grid-like codes are time-locked to frontal eye field activation.** In the Donders study, neural activity in the frontal eye fields co-varied with the magnitude of saccade-based grid-like codes in the entorhinal cortex ($N = 32$, one-sample $t$-test). Results are shown at $p < 0.05$ FWE-corrected at cluster-level (cluster-defining threshold of $p < 0.001$, cluster extent = 116 voxels). Source data are provided as a Source Data file.

entorhinal grid-like codes would be coupled to activation changes in this wider network of brain regions. To reveal whether BOLD activation varied as a function of saccade-based entorhinal grid-like codes, we adopted a whole-brain approach. We examined the data from the Donders study and, for each participant, averaged the magnitude of entorhinal grid-like codes across all saccades within a single scene trial. We then modeled all scene trials during the study period and included the magnitude of trial-wise grid-like codes as a parametric modulator in a group-based analysis (see Methods section).

Results from the whole-brain analysis showed that, as participants viewed scene images, increased saccade-based grid-like codes were coupled to increased activation specifically in the left FEF. In other words, if participants made saccades that were aligned with their individual entorhinal grid orientation (i.e., resulting in larger grid-like codes for that trial), BOLD activation in the left FEF was high as well ($N = 32$, one-sample $t$-test, $p < 0.05$ FWE-corrected at cluster level using a cluster-defining threshold of $p < 0.001$, cluster size = 116 voxels; $x = 55$, $y = 22$, $z = 31$; Fig. 4; please note that we only performed this analysis for the Donders study, as the partial field-of-view in the Vienna study did not allow for reliable FEF coverage). FEF activation was neither significantly tied to the average saccade numbers during encoding ($p > 0.05$) nor related to recognition memory performance across participants ($d$-prime, $p > 0.05$).

## Discussion

In the present study, we investigated whether saccade-based grid-like codes in the human entorhinal cortex play a role in human memory formation. Leveraging data from two independent data sets, we consistently identified saccade-based grid-like codes in the left entorhinal cortex while participants studied scene images. Interestingly, in the Donders study, grid signals appeared lower at better individual recognition memory, and higher grid signals were associated with increased neural activation in the frontal eye fields (FEF). Together, these findings highlight saccade-based entorhinal grid-like codes as a potential player in human memory formation and reveal their link to the FEF that are crucial for eye movement coordination.

We hypothesized that saccades during scene viewing were coupled to grid-like codes in the entorhinal cortex and that the grid-related signals would be linked to individual variations in recognition memory performance. In line with this prediction, we found significantly increased entorhinal grid-like codes related to saccades in two independent studies. This effect was specific to subsequently remembered scenes (hits) across both data sets (but note that the small number of miss trials, particularly in the Vienna study, might have limited analysis reliability). This is consistent with reports of grid cells in the entorhinal cortex of non-human primates that were shown to respond to multiple gaze locations and saccade directions during free visual exploration[9,10]. Similarly, grid-like codes in the human entorhinal cortex were discovered while individuals performed a visual

search task[6], a visual tracking task[7], or while freely viewing scene images[13]. By encoding saccade directions, grid-like signals may represent relational information between the different elements of a visual scene[1], potentially providing us with a mental framework for the organization of visual memory reminiscent of the cognitive map for physical space[18,37]. Mapping visual space to guide navigation should be highly adaptive for all animals that use vision as their primary sense for exploration, including humans. While relational processing has been proposed as a mechanism by which visual grid cells may support memory[1], this relationship was not clearly evident in our data. However, our studies were not specifically designed to test relational processing, and our stimulus set may have allowed for scene recognition based on visual properties, such as color, perspective, or global features. We thus encourage future work to interrogate the relationship between relational processing and saccade-based entorhinal grid-like codes during memory formation with dedicated paradigms.

Crucially, we found that saccade-based grid-like codes were lower the better participants discriminated old from novel scene images in a subsequent recognition memory task (we did not detect this relationship in the Vienna data set, possibly due to the fact that these participants encoded substantially fewer scene images). The negative brain-behavior relationship was confirmed when focusing exclusively on hits: lower saccade-based entorhinal grid-like codes were tied to better memory performance (hit rate) across individuals and data sets. Overall, our findings reinforce the notion that grid-like codes, which are typically discussed in the context of spatial navigation, also play a role in the representation of visual space and are negatively associated with subsequent memory. To the best of our knowledge, only one previous paper described a relationship between visual grid cell activity and visual memory in non-human primates[9]. The authors showed that grid cells located in the anterior entorhinal cortex displayed neural adaption upon stimulus repetition (i.e., decreased activation as a surrogate marker of memory)[19,20]. Other work on the relationship between (non-visual) grid-like codes and behavior yielded mixed results: Some studies have linked lower[17] or higher grid-like codes[11,15,16] to better spatial memory performance, while others reported no significant brain-behavior relationship[7,38].

To better understand the negative brain-behavior relationship in our data, we speculated that participants who were better at recognizing the scene images may have used different strategies to complete the task (intentionally or not). Rather than predominantly relying on visuospatial encoding, they may have drawn upon prior knowledge more extensively. For instance, when viewing a beach scene, they might have recalled a recent beach vacation or their mental "schema" of a typical beach scene with highly similar visual features. Such integration of prior knowledge could have facilitated scene encoding by guiding eye movements to prominent scene features, thereby enhancing participants' recognition memory. This is consistent with the idea that schema-congruent information is encoded more efficiently and relies less on the medial temporal lobes[39,40]. In contrast, participants who relied exclusively on visuospatial encoding may have shown lower recognition memory performance while displaying increased medial temporal lobe activity and higher grid-like codes for visual space. Unfortunately, we did not qualitatively assess participants' strategy use during encoding, nor did we account for individual experience or expertise with the specific scene stimuli, which could have helped to clarify the role of grid-like codes in memory formation.

Alternatively, individuals with good recognition memory might display less neural activation during memory formation, thus, encoding "more efficiently"[41]. For instance, increased memory performance due to memory training was associated with reduced neural activation in brain areas relevant to mental navigation, including the posterior hippocampus and retrosplenial cortex[34]. In macaques, long-term practice was associated with reduced glucose uptake while maintaining neural activity levels[42]. We previously showed that individuals who

exhibited less entorhinal grid-like codes and entorhinal-cortical connectivity when observing a demonstrator moving through virtual space performed better when later retracing the demonstrator's path[17]. Nau and colleagues found that fMRI-based directional coding in the medial temporal lobe was weaker in participants with better memory performance in a virtual spatial navigation task as quantified by a smaller drop-error when trying to replace objects to their locations[33]. Note, however, that the correlation between grid-like codes and overall BOLD signal was not significant in our data set (Supplementary Results S4, Supplementary Fig. S4). Similarly, high-performing participants may have explored the scenes more selectively, suppressing saccades toward visual features that were less informative for subsequent recall[43,44] and resulting in lower grid-like codes (but this stands in contrast to our observation that participants who made more saccades recognized the scenes better; see the Supplementary Discussion for a more detailed account of grid-like codes in relation to saccade characteristics and visual exploration patterns). Overall, the observed negative brain-behavior relationship challenges the idea that grid-like codes universally support memory formation. Rather than enhancing memory per se, grid-like codes may sometimes interfere with memory, or their contribution may depend on specific conditions. Bicanski and Burgess argue that grid-like codes mainly contribute to relational memory for familiar content, suggesting they may be less involved in directing gaze when encoding previously unknown scenes[1]. Depending on prior knowledge and task demands, we propose that increased grid-like codes may hinder or support memory formation, which is an interesting aspect warranting future interrogation.

We next hypothesized that saccade-based grid-like codes in the entorhinal cortex were tied to neural activity in visuo-oculomotor regions that are known to be involved in saccade generation and coordination, such as the visual cortex and the FEF[26]. Here, we built on previous work proposing that memory-guided saccades may result from coordinated activity between medial temporal and visuo-oculomotor regions, supported by entorhinal grid-like activity[1,18]. Indeed, we observed that increased grid-like codes in the entorhinal cortex were related to an increase in FEF activation. In other words, when participants made saccades that were aligned with their individual grid orientation, activation in the FEF was high as well, possibly reflecting an interaction between oculomotor processes and grid-like activity. Interestingly, FEF activation was unrelated to overall saccade numbers and, unlike grid-like codes, showed no clear association with memory performance. This may indicate that different mechanisms are at play and that the relationship between the entorhinal cortex and oculomotor regions needs further investigation.

The entorhinal cortex and the FEF are closely connected and are embedded in a set of regions spanning visual[24] and medial temporal areas[45,46]. Disynaptic pathways between these areas provide the ideal infrastructure to interface between regions for memory and oculomotion[47]. Ryan and colleagues modeled the functional dynamics between these regions and demonstrated that medial temporal lobe lesions disrupted signal transmission to different oculomotor areas, including the FEF[48]. This is in line with findings from patients with Alzheimer's Disease (AD) who show the first signs of neurodegenerative changes in the entorhinal cortex[49]. Importantly, AD patients display altered eye movement patterns, with saccades lacking accuracy and speed when directed toward, or when suppressing movement toward, a predefined target[50–53]. This suggests that the entorhinal cortex provides relevant visuospatial input into the FEF, thereby informing the planning and execution of subsequent saccades. In turn, saccades shape the activity in the medial temporal lobe[54,55]. Dynamic causal modeling revealed that free viewing during mental scene construction (as opposed to a restricted viewing condition) enhanced excitatory functional connections from the medial temporal lobe to the FEF, indicating the influence of saccadic activity on the interaction

between these regions[56]. Transcranial magnetic stimulation of the FEF was further shown to disrupt spatial working memory[26]. Considering these findings, saccade-based entorhinal grid-like codes may not only encode the spatial relationships between visual features but also interact with the FEF to guide the computation of future saccade directions and to support perception and memory[1]. In doing so, grid-like signals (which were, unexpectedly, negatively associated with memory) may help coordinate the interplay between medial temporal and visuo-oculomotor regions involved in memory formation.

Even though grid-like codes have been consistently reported in prior and in our present work, there is an active debate about the extent to which fMRI-based grid-like codes reflect the underlying cellular activity[12]. Single-cell recordings in humans have identified grid cells in the entorhinal cortex during virtual navigation[57], and entorhinal grid-like codes have been linked to saccadic eye movements[6,7]. MEG and intracranial electroencephalography (EEG) recordings revealed grid-like modulation of visual space in the human anterior medial temporal lobe[13]. However, direct evidence connecting grid cell recordings in humans to saccades during visual exploration is currently missing. Bicanski and Burgess proposed a computational model in which grid cells encode trajectories between salient stimulus features of a visual scene, offering a possible explanation for their role in memory encoding[1]. Future studies involving partial occlusions (e.g., Wynn et al.[58]) or an analysis of eye movement reinstatement[59] (where gaze during retrieval is returned to locations that were viewed during encoding) could help explain whether relational processing mediates the relationship between memory formation and grid-like codes. Another challenge is posed by the generally slower temporal resolution of the fMRI-based BOLD signal, rendering it unlikely that measured fluctuations in brain activity can be tied to single saccades (i.e., fast-occurring events with millisecond-duration). While we employed a short repetition time (TR) to maximize the temporal resolution in the Donders study (657 ms), measurements in the Vienna study were based on a longer TR (2029 ms). This may account for the generally weaker effects observed in that data set (i.e., one TR likely included several saccades, potentially yielding lower grid-like codes due to averaging across multiple saccades that were (mis-)aligned with the individual grid orientation). Nevertheless, our results are backed up by previous studies that assessed saccade-based grid-like codes with fMRI[6,7], as well as by our numerous control analyses that corroborate the validity and stability of the findings.

To conclude, we identified grid-like codes in the entorhinal cortex that were time-locked to saccades. Across individuals in the Donders study, the saccade-based grid-like codes during scene viewing were specific to later remembered trials and were lower at better subsequent recognition memory, suggesting that grid signals contribute to memory formation. Moreover, the magnitude of grid-like codes was coupled to increases in neural activation within the FEF, a brain region involved in saccade generation and coordination. Our findings show that saccade-based grid-like codes in the entorhinal cortex play a role in human memory formation (which unexpectedly appeared to be negative), highlighting interregional coordination of neural activity that is time-locked to the internal map of visual space.

## Methods
### Data set obtained at the Donders Institute ("Donders study")
**Study setup.** This study belonged to a larger project examining the effects of eye movements on memory processing (performed at the Donders Institute for Brain, Cognition and Behaviour, Nijmegen, The Netherlands). In two separate sessions, participants underwent MEG (not reported here, but see Staudigl et al.[60]) and fMRI (see also Wagner et al.[61]) while monitoring their eye movements, and while performing a recognition memory task. The order of fMRI/MEG sessions was balanced across participants and involved parallel task versions to avoid training effects.

**Participant sample.** Forty-eight participants volunteered for this study. Sixteen participants were excluded due to not completing the study (7 individuals), excessive motion (4 individuals), technical problems during the data recording (3 individuals), a low number of identified saccades during the fMRI session (1 individual, < 30 detected saccades per condition), or low recognition memory performance during the MRI session (1 individual, false alarms > correct rejections). The final sample thus comprised 32 participants (23 females, age range 18–30 years, mean age = 23 years, 32 right-handed). Biological sex was assessed by self-report. Sex differences were not the focus of this study, and sex-based analyses were not performed. All individuals were healthy and did not report any history of neurological and/or psychiatric disorders, had normal or corrected-to-normal vision, and provided written informed consent before the start of the experiment. Participants received monetary compensation or course credit. The study was reviewed and approved by the local ethics committee (Commissie Mensgebonden Onderzoek, region Arnhem-Nijmegen, The Netherlands; reference number CMO-2014/288).

**Recognition memory task.** During the study period, participants were instructed to memorize 200 scene images (100 indoor, 100 outdoor). Images were resized to a dimension of 640 × 480 pixels and were presented on a black background. Each scene was shown for 4 s during which participants could freely view the image. To ensure attention to each scene, participants were asked to judge whether the image depicted an indoor or outdoor scenario via button press during the subsequent fixation period (2125, 4125, or 7125 ms, 80/80/40 distribution across the 200 trials, pseudo-randomized), after which the next scene appeared. The order of scenes was pseudorandomized, with no more than four scenes of the same type (indoor/outdoor) shown consecutively. The study period was followed by a distractor task (i.e., solving simple mathematical problems, 1 min) and a rest period (3 min).

During the test period, participants viewed all scene images that were shown during the previous study period, intermixed with 100 novel scene images (half of them indoor/outdoor; i.e., a total of 300 scene images were presented). Some of the novel scene images were similar to the previously studied ones (e.g., comparison between a new and a previously studied forest scene) or could be markedly different (e.g., comparison between a forest and a beach scene). Hence, recognition judgments were based on scenes with shared visual elements rather than on entirely separate scene categories. The assignment of scene images to study or test periods was counterbalanced across participants. Scenes were presented for 4 s each and were followed by a 6-point rating scale that required participants to indicate whether they recognized the scene as "old" or "new" (self-paced; the scale ranged from (1) "very sure old" to (6) "very sure new"), and a fixation period until the next trial started (2125, 4125, or 7125 ms, 80/80/40 distribution across the 300 trials, pseudo-randomized). The test period was divided into 2 blocks separated by a short break. After completing the task, participants were asked to fixate on different locations on the screen to evaluate eye tracker accuracy (5 min), followed by the structural scan.

**Recognition memory performance (d-prime).** Trials were grouped into four bins based on individual performance during the recognition memory test: (1) scenes that were correctly judged as "old" (i.e., hits, collapsing across confidence ratings 1–3, mean ± standard error of the mean (SEM): 140.9 ± 5.6 trials); (2) scenes that were correctly judged as "new" (i.e., correct rejections, collapsing across confidence ratings 4–6, 85.8 ± 1.8 trials); (3) scenes that were incorrectly judged as "old" (i.e., false alarms, collapsing across confidence ratings 1–3, 14.2 ± 1.8 trials); (4) scenes that were incorrectly judged as "new" (i.e., misses, collapsing across confidence ratings 4–6, 59.1 ± 5.6 trials). None of the participants displayed any actually missed trials without button presses. Individual hit and false alarm rates were $z$-scored, and recognition memory performance ($d$-prime) was calculated as [$z$(hits) – $z$(false alarms)].

**Eye tracking data acquisition, analysis, and saccade detection.** To capture saccadic eye movements, we recorded horizontal and vertical eye gaze and pupil size, using a video-based infrared eye tracker (Eye-Link 1000 Plus, SR Research, Ontario, Canada). Eye tracking was performed at a viewing distance of 86.6 cm. The screen had a resolution of 1280 × 960 pixels and measured 36.9 cm in width and 27.7 cm in height.

Before recording, raw eye movement data was mapped onto screen coordinates by means of a calibration procedure. Participants sequentially fixated on nine fixation points on the screen, arranged in a 3 × 3 grid. This was followed by a validation procedure during which the nine fixation points were presented once more while the differences between the current and previously obtained gaze fixations (from the calibration period) were measured. The calibration settings were accepted if these differences were < 1° of visual angle, and the eye tracker recording was started. A detailed assessment of eye tracking data quality, including saccade and fixation metrics is provided in the Supplement (Supplementary Methods S1, Supplementary Figs. S8, S9).

Eye tracking data was processed using Fieldtrip (https://www.fieldtriptoolbox.org) through a two-step procedure consisting of automatic saccade detection followed by visual inspection of the data. Saccadic eye movements were identified by transforming vertical and horizontal eye movements into velocities, whereby velocities exceeding a threshold of 6 × the standard deviation ($SD$) of the velocity distribution and with a duration of > 12 ms were defined as saccades[62]. Saccade onsets during trials of the study period (i.e., during the presentation of scene images) were defined as events-of-interest. Only saccades that followed a minimum fixation period of 25 ms were included. Saccades that were followed or preceded by blinks (+/−100 ms) were excluded (blinks were defined as large deflections in pupil diameter: mean ± 5 standard deviations; eye tracking data in the vicinity of blinks is unreliable due to saturation effects). To identify noise or blinks that could have been misattributed as saccades, trials with more than 25% of missing eye tracker data were discarded. We detected a total of 48,510 saccades in the eye tracking data ($N = 32$; average number of saccades per participant, mean ± SEM: 1515.94 ± 80.95 saccades).

**MRI data acquisition.** Imaging data were collected at the Donders Institute for Brain, Cognition and Behaviour (Nijmegen, The Netherlands), using a 3T Prisma Fit scanner (Siemens, Erlangen, Germany) equipped with a 32-channel head coil. We acquired on average 2456 (± 5.3) T2*-weighted blood oxygen level-dependent (BOLD) images during the study period of the recognition memory task, using the following echo-planar imaging (EPI) sequence: repetition time (TR) = 657 ms, echo time (TE) = 30.8 ms, multi-band acceleration factor = 8, 72 axial slices, interleaved acquisition, field of view (FoV) = 174 × 174 mm, 72 × 72 matrix, flip angle = 53°, slice thickness = 2.4 mm, no slice gap, voxel size = 2.4 mm isotropic. The structural image was acquired using a standard magnetization-prepared rapid gradient-echo (MPRAGE) sequence with the following parameters: TR = 2300 ms, TE = 3.03 ms, FoV = 256 × 256 mm, flip angle = 8°, voxel size = 1 mm isotropic.

**MRI data preprocessing.** The fMRI data were processed with SPM8 in combination with MATLAB (The Mathworks, Natick, MA, USA). The first 12 volumes were excluded to allow for T1-equilibration. The remaining volumes (of both the study and test periods) were realigned to the mean image. The structural scan was co-registered to the mean functional image and was segmented into gray matter, white matter, and cerebrospinal fluid using the "New Segmentation" algorithm. All images (functional and structural) were then spatially normalized to

the Montreal Neurological Institute (MNI) EPI template using Diffeomorphic Anatomical Registration Through Exponentiated Lie Algebra (DARTEL)[63], and functional images were further smoothed with a 3D Gaussian kernel (6 mm full-width at half-maximum, FWHM).

**Region-of-interest (ROI) definition.** For the analysis of grid-like codes, left and right posterior medial entorhinal cortex masks were based on Maass et al.[22]. Masks were binarized and co-registered to the mean functional image of one participant (Maass, bilateral entorhinal cortex: 25 voxels, left entorhinal cortex: 14 voxels, right entorhinal cortex: 18 voxels). To validate the quality of the co-registration, the overlap between each mask and the corresponding (co-registered) structural and mean functional image was visually assessed for each participant.

To test for potential grid-like codes in control regions, we defined additional ROIs that are known to be involved in memory, visuo-spatial processing, and oculomotor control, but for which no significant grid-like codes have been detected. This included the hippocampus, anterior thalamus, FEF, and visual cortex. The hippocampus and visual cortex were defined based on bilateral anatomical masks of the Automatic Anatomical Labeling (AAL) atlas[64] (hippocampus = 1148 voxels, visual cortex = 1860 voxels). To delineate the anterior thalamus, we used the stereotactic mean anatomical atlas provided by Krauth and colleagues[65] (© University of Zurich and ETH Zurich, Axel Krauth, Rémi Blanc, Alejandra Poveda, Daniel Jeanmonod, Anne Morel, Gábor Székely), which is based on histological, cytoarchitectural features defined ex vivo[66]. We specified the anterior thalamus by combining the bilateral anterior dorsal, -medial, and -ventral nucleus masks (59 voxels). The FEF were defined by contrasting memory-related activity during scene encoding across all participants (later remembered > later forgotten). The resulting cluster peak coordinate ($x = 43$, $y = 7$, $z = 29$) was surrounded by a 10 mm sphere and was mirrored to create a bilateral ROI (320 voxels). See Supplementary Results S2, Supplementary Fig. S2A.

**Analysis of grid-like codes.** Grid-like codes were analyzed using the openly available Grid Code Analysis Toolbox[67] (GridCAT, software version 1.0.4, https://www.nitrc.org/projects/gridcat, which is based on the procedures developed by Doeller et al.[11].

Saccades during the study period of the recognition memory task were defined as events-of-interest. We then leveraged the General Linear Model (GLM) to model the BOLD response time-locked to saccade onsets. All saccades were estimated with stick functions (duration = 0 s) and were convolved with the SPM default canonical hemodynamic response function (HRF). To account for noise due to head movement, we included the six realignment parameters, their first derivatives, and the squared first derivatives in the design matrix. A high-pass filter with a cutoff at 128 s was applied.

Analysis of grid-like codes progressed in two steps pertaining to estimating and testing individual grid orientations. We partitioned the data into two equally-sized data halves (i.e., corresponding to two separate regressors that contained the saccades of the estimation or test data sets, respectively). During step 1 (GLM 1), saccade-related activity of the estimation data set (i.e., the first regressor) was modulated by the respective saccade direction. This was calculated as saccade angle ($\alpha_t$) relative to a predefined reference point and was modeled using two parametric modulators, $\sin(\alpha_t*6)$ and $\cos(\alpha_t*6)$, that converted directional information into 60° space, reflecting the hypothesized 6-fold rotational symmetry in the fMRI signal (presumably due to the firing patterns of underlying grid cells). The voxel-wise beta estimates, $\beta_1$ and $\beta_2$, of the two parametric modulators were then extracted, and the mean grid orientation within the respective ROI was calculated using arctan[mean($\beta_1$)/mean($\beta_2$)]/6 (i.e., converting directional information back into 360° space).

During step 2, the estimated grid orientations were then tested in a second GLM (GLM 2) that was virtually identical to the abovementioned model, but with the exception that the saccades within the first regressor (i.e., the estimation data set) were unmodulated, while the saccades in the second regressor (i.e., the test data set) were parametrically modulated by the difference between the respective saccade angle ($\alpha_t$) and the individual ROI-based grid orientation ($\varphi$) using $\cos[6*(\alpha_t-\varphi)]$. In other words, a smaller difference between $\alpha_t$ and $\varphi$ should result in increased grid-like codes since the saccade direction is aligned with the individual grid orientation. The beta values from the parametric modulator were then extracted for all voxels within the ROI and were averaged to produce the mean amount of grid-like codes.

ROI-based grid-like code data were analyzed using a set of Wilcoxon-tests in R (software version 4.3.0; https://www.r-project.org; R stats version 3.6.2). We hypothesized that significant grid-like codes in the entorhinal cortex should be associated with a 6-fold rotational symmetry of the fMRI signal in the entorhinal cortex. This is based on the assumption that participants cross more grid cell firing fields as they perform saccades aligned with the underlying grid axes. The choice of the statistical test thus reflected an a priori expectation, which is why we adopted an α-level of 0.05 (one-tailed). Effect sizes were calculated as Cohen's d. Additionally, we applied Bonferroni-correction to account for multiple comparisons (1 entorhinal cortex ROI and 4 control ROIs), using a threshold of $\alpha_{Bonferroni} = 0.05/5$ ROIs = 0.01 or (6-fold symmetry and 4 control symmetries), using a threshold of $\alpha_{Bonferroni} = 0.05/5$ ROIs = 0.01, respectively. Grid-like code values exceeding the median value ± 3 × the median absolute deviation (MAD) were excluded from the analyses. We chose this method because the mean and standard deviation are particularly sensitive to outliers whereas the median is not[68].

**Whole-brain activation modulated by saccade-based entorhinal grid-like codes.** We performed additional analyses to test whether the activity of entorhinal grid-like codes modulated voxel-wise changes in whole-brain activation. The magnitude of grid-like codes of each saccade was taken from the results of GLM 2 (this GLM had tested the previously estimated grid orientations in the second half of the data). To obtain grid-like codes for the first half of the data, we repeated the analysis but reversed the partitioning of the estimation/test data sets (i.e., we estimated grid orientations on the second data half and tested them on the first data half). Saccade-based grid-like codes were then extracted from the parametric modulation regressor (i.e., relying on the difference between each saccade's translational direction and the mean grid orientation of the participant, whereby a smaller difference should be associated with a stronger grid-like signal within the entorhinal cortex). We then averaged the magnitude of grid-like codes of all saccades within a trial, producing a trial-wise value for grid-like codes.

Next, to be able to perform a group-based analysis, we used the normalized, standard-space data and created a separate GLM (GLM 3). This model contained a single task regressor that captured all scene trials that were presented during the study period (modeled with a boxcar function, duration 4 s). This regressor was parametrically modulated with trial-wise grid-like codes. As above, GLM 3 included the six realignment parameters, their first derivatives, and the squared first derivatives into the design matrix. A high-pass filter with a cutoff at 128 s was applied. We then contrasted the parametric modulation regressors that captured the trial-wise fluctuations in entorhinal grid-like codes against baseline (entorhinal grid-like codes during scene > implicit baseline) and tested for group effects by submitting the individual contrast images to a one-sample t-test. Significance was assessed using cluster-inference with a cluster-defining threshold of $p < 0.001$ and a cluster-probability of $p < 0.05$ family-wise error (FWE) corrected for multiple comparisons. The corrected cluster size (i.e., the spatial extent of a cluster that is required in order to be labeled as significant) was calculated using the SPM extension "CorrClusTh.m"

and the Newton-Raphson search method (script provided by Thomas Nichols, University of Warwick, United Kingdom, and Marko Wilke, University of Tübingen, Germany; http://www2.warwick.ac.uk/fac/sci/statistics/staff/academic-research/nichols/scripts/spm/).

**Data set obtained at the University of Vienna ("Vienna study")**
**Study setup.** To validate our results, we repeated the analyses in an independent data set involving different participants who took part in two separate sessions (the study was performed at the University of Vienna, Austria). First, participants completed a recognition memory task (study period and immediate test) during fMRI scanning while their eye movements were recorded. Second, their recognition memory was tested once more in the behavioral laboratory after one week (delayed test).

**Participant sample.** Fifty participants were invited to partake in the study. Four participants were excluded due to technical problems during the eye tracking recording, leaving a sample of 46 individuals. For the analysis of grid-like codes, 26 more participants were excluded due to a low number of identified saccades during the fMRI session (2 individuals < 30 detected saccades per condition) or due to signal drop-outs in the entorhinal cortex region (24 individuals < 14 voxels in ROI mask) to match the number of voxels in ROI masks of the Donders study (as we have done previously[17]), resulting in a sample of 20 participants (15 females, age range 18–29 years, mean age = 21.75 years, 18 right-handed). Biological sex was determined by self-report. Sex differences were not the focus of the study and were therefore not analyzed. All participants were healthy, did not report any history of neurological and/or psychiatric disorders, had normal, or corrected-to-normal vision, and provided written informed consent prior to participation. Participants received monetary compensation or course credit. The study was reviewed and approved by the local ethics committee of the University of Vienna (reference number 00538).

**Recognition memory task.** The recognition memory task consisted of two study-test cycles (i.e., $study_1$, $test_1$, $study_2$, $test_2$). During each study period, participants were instructed to memorize 48 scene and 48 face images (24 indoor/outdoor scenes, 24 female/male faces; scenes derived from the same stimulus set used in the Donders study above). Images were presented in the dimensions of $500 \times 500$ pixels on a gray background. An image was shown for 3 s, during which participants could freely view the image and was followed by a fixation cross (inter-trial-interval 2–7 s, mean 5 s). The order of images was pseudo-randomized with the restriction that no more than three images of the same scene/face category were presented in succession while ensuring an equal number of scene/face images was displayed in every quartile of each study period. Across both study-test cycles, participants studied 192 images.

After each study period, participants completed a test period (i.e., the immediate test) where the 96 previously viewed ("old") images were pseudo-randomly interleaved with 48 novel ("new") images, with the constraint that no more than three images of the same image category (scene or face) or the same memory condition ("old" or "new") were presented in succession. Previously viewed and novel images could share visual elements (e.g., comparison between two forest scenes), while some were markedly different (e.g., comparison between a forest and a beach). As above, each image was presented for 3 s, followed by a 4-point rating scale that prompted participants to indicate whether they recognized the image as "old" or "new", ranging from "very sure old" to "very sure new" (duration 2 s). The next trial started after an inter-trial-interval during which a fixation cross was presented on the computer screen (duration 2–7 s, mean 5 s).

During the delayed test after one week, participants were shown all 192 images that were studied during the initial study periods (i.e., all "old" stimuli), interleaved with 96 novel, unseen images. Image presentation was again pseudo-randomized such that no more than three images of the same image category (scene or face) or memory

condition ("old" or "new") would appear in succession. In the following, we will focus on analyzing scene images only to enable a more direct comparison between Donders and Vienna studies.

**Recognition memory performance (d-prime).** Depending on individual performance during the immediate or delayed recognition memory task, scene trials were marked as (1) images that were correctly judged as "old" (i.e., hits, collapsing across confidence ratings 1–2; mean ± standard error of the mean (SEM): immediate test, 86.9 ± 2.00 trials, delayed test, 84.8 ± 2.59 trials); (2) scenes that were correctly judged as "new" (i.e., correct rejections, collapsing across confidence ratings 3–4; immediate test, 9.72 ± 2.11 trials, delayed test, 11.4 ± 2.62 trials); (3) scenes that were mistakenly recognized as "old" (i.e., false alarms, collapsing across confidence ratings 1–2; immediate test, 3.64 ± 0.85 trials, delayed test, 5.12 ± 1.32 trials); (4) scenes that were mistakenly rejected as "new" (i.e., misses, collapsing across ratings 3–4; immediate test, 45.5 ± 0.77 trials, delayed test, 43.1 ± 1.22 trials). Individual hit and false alarm rates were $z$-scored separately for the immediate and delayed test. To avoid an indeterminate $d'$ (this problem arises with hit or false alarm rates of 0 or 1), we used the log-linear approach, where 0.5 is added to both the number of hits and false alarms, while 1 is added to the number of signal and noise trials, before calculations are performed[69]. To compensate for an unequal number of signal and noise trials in this study's design (signal trials = 96 old images, noise trials = 48 new images at test), we added proportional values to the number of hits and false alarms (i.e., 0.7 to the number of hits and 0.3 to the number of false alarms; 2 × 0.7 to the number of signal trials, 2 × 0.3 to the number of noise trials). Recognition memory performance was then quantified using $d$-prime, calculated as the difference between these adjusted hit and false alarm rates [$z$(hits) – $z$(false alarms)]. Analysis of recognition memory performance ($d$-prime) was carried out in MATLAB (The Mathworks, Natick, MA, USA, R2020b, dprime_simple.m, version 1.1.0.0 by Karin Cox) and R (versions, base, version 4.3.0, dplyr, version 1.1.3).

**Eye tracking data acquisition, analysis, and saccade detection.** Saccades were tracked by recording horizontal and vertical eye gaze and pupil size with a video-based infrared eye tracker (EyeLink 1000 Plus, SR Research, Ontario, Canada), performed at a viewing distance of 102 cm. The monitor measured 69.84 cm in width and 39.88 cm in height. Screen resolution varied across participants. A resolution of 1920 × 1080 pixels was used for participants 1–3, 14–15, 17–22, and 36–37, while a resolution of 1280 × 1024 pixels was used for participants 4–13, 16, 23, 25–28, 30–35, 38–47, 49, and 50. To map raw eye movement data onto screen coordinates, we implemented a calibration-validation procedure (as described for the Donders study). A detailed assessment of eye tracking data quality is provided in the Supplement (Supplementary Methods S1, Supplementary Figs. S8, S9). Eye tracking data were processed using Fieldtrip (https://www.fieldtriptoolbox.org).

Saccades were excluded if they were preceded or followed either by blinks (± 300 ms) or other saccades (± 100 ms). Blinks were defined as pupil dilations deviating more than one standard deviation from the mean pupil diameter. We counted a total of 19,818 saccades in the eye tracking data ($N = 46$; average number of saccades per participant, mean ± SEM: 430.8 ± 15.03 saccades).

**MRI data acquisition.** Imaging data were collected at the Neuroimaging Center of the University of Vienna, using a 3T Skyra MR-Scanner (Siemens, Erlangen, Germany) equipped with a 32-channel head coil. On average, we acquired 396.77 (± 7.24 SD) T2*-weighted blood oxygenation level-dependent (BOLD) images during each of the two study periods and 732.54 (± 9.10 SD) BOLD images during the two immediate test periods of the recognition memory task. We used the following partial-volume echo-planar imaging (EPI) sequence: TR = 2029 ms;

TE = 30 ms; number of slices = 30 axial slices; slice order = interleaved acquisition; FoV = 216 mm; flip angle = 90°; slice thickness = 3 mm; in-plane resolution = 2 × 2 mm, using parallel imaging with GRAPPA acceleration factor of 2. Slice orientation was parallel to the line connecting the anterior and posterior commissure (AC-PC alignment), with a 10° rotational shift upwards. The T1-weighted structural image was acquired using a standard magnetization-prepared rapid gradient-echo (MPRAGE) sequence with the following parameters: TR = 2300 ms; TE = 2.43 ms; FoV = 240 mm; flip angle = 8°; voxel size = 0.8 mm isotropic. We additionally acquired a T2-weighted structural image used to delineate the entorhinal cortex. A turbo-spin-echo (TSE) Sampling Perfection with Application optimized Contrasts was applied using a different flip angle Evolution (SPACE) sequence with the following parameters: TR = 3.2 s; TE = 564 ms; FoV = 256 mm, voxel size = 0.8 mm isotropic, slices were oriented perpendicular to the long axis of the hippocampus.

Due to the local proximity to air-filled cavities, entorhinal cortices are susceptible to image distortions. To ameliorate this effect, we collected 30 images with the same functional sequence but with a reversed phase-encoding direction (thus, stretching potential image distortions into the opposite direction). Additionally, we acquired 10 whole-brain EPI images to facilitate the co-registration of anatomical entorhinal cortex masks to the partial-volume EPI images with the following parameters: TR = 2.832 s, TE = 30 ms, number of slices = 42 axial slices, slice order = interleaved acquisition, FoV = 216 mm, flip angle = 90°, slice thickness = 3 mm, in-place resolution = 2 × 2 mm, using parallel imaging with a GRAPPA acceleration factor of 2. As above, slices were oriented parallel to the AC-PC line with a 10° rotational shift upwards.

**MRI data preprocessing.** The fMRI data were processed using SPM12 (https://www.fil.ion.ucl.ac.uk/spm/) in combination with MATLAB (The Mathworks, Natick, MA, USA, R2020b). Structural and functional scans were manually AC-PC corrected. The first six functional volumes were then excluded to allow for T1-equilibration. The remaining volumes were slice-time-corrected to the middle slice and spatially realigned to the mean functional image (across both study-test cycles). FSL's "topup" command (FMRIB Software Library; https://fsl.fmrib.ox.ac.uk/fsl/fslwiki/topup)[70] was applied to correct potential image distortions. Specifically, the mean functional image was calculated based on the 30 functional volumes (with the reversed phase-encoding direction) and was used to estimate and correct susceptibility-induced distortions. Since grid-like codes were analyzed in subject-native space, we refrained from normalizing the data. Functional images were smoothed with a 3D Gaussian kernel (5 mm FWHM).

For the whole-brain group analyses, the distortion-corrected data was normalized into standard space. The structural scan was co-registered to the mean functional image (across both study-test cycles) and was segmented into gray matter, white matter, and cerebrospinal fluid using the "New Segmentation" algorithm. All images (functional and structural) were spatially normalized to the Montreal Neurological Institute (MNI) EPI template (MNI-152) using Diffeomorphic Anatomical Registration Through Exponentiated Lie Algebra (DARTEL)[63], and functional images were smoothed with a 3D Gaussian kernel (5 mm FWHM).

**ROI definition.** Left and right entorhinal cortex masks were segmented using the Automatic Segmentation of Hippocampal Subfields algorithm (ASHS, software version 1.0.0, https://sites.google.com/site/hipposubfields/)[71] based on each participant's T1- and T2-weighted, high-resolution structural image. Masks were binarized and transformed into the subject-native space of the (partial-volume) functional images. To facilitate co-registration (which can be hampered by the partial-volume field-of-view), we progressed in several steps: First, participants' T2-weighted structural scan (along with the segmented left and right entorhinal cortex masks) was co-registered to align with

the mean functional image (based on the 10 whole-brain functional images we acquired). Second, the mean (whole-brain) functional image (along with the co-registered T2 image and the entorhinal cortex masks) was co-registered to the mean (partial-volume) functional image. The overlap between each entorhinal cortex mask and the corresponding (co-registered) structural and functional data was visually inspected for each participant.

Due to its location close to the lateral ventricle, the entorhinal cortex can be associated with a lower signal-to-noise ratio. To bypass this issue, only voxels that exceeded a signal-to-noise threshold of 0.8 were examined, mainly leading to the exclusion of voxels along the anterior-medial entorhinal cortex border. Participants with less than 14 voxels in the (left or right) entorhinal cortex mask were excluded from the analyses. Consequently, in alignment with the ROI from the Donders study, analyses were focused on the posterior-medial entorhinal cortex and were based on a final sample of 20 participants (mean ± SEM; left entorhinal cortex, 19.85 ± 1.09 voxels, right entorhinal cortex, 20.35 ± 1.19 voxels).

**Analysis of grid-like codes.** Grid-like codes during the study periods of the recognition memory task were analyzed identically to above. Analyses were focused on saccades during scene presentations only (saccades during face images were collapsed in a regressor-of-no-interest and were not modulated by their respective saccade direction angle). Each study period was partitioned into equal halves of estimation/test data sets and both study periods were combined into one GLM.

**Whole-brain activation modulated by saccade-based entorhinal grid-like codes.** In the Vienna study, the partial field of view (FoV) during fMRI was designed to optimize signal quality in the entorhinal cortex and surrounding medial temporal lobe regions, but did not reliably cover the FEF. To assess FEF coverage, we used Nilearn's compute_epi_mask function to generate a subject-specific field-of-view mask from each participant's mean normalized functional image. We overlaid this mask onto the normalized group-average T1-weighted image, together with the binarized FEF cluster, to visually inspect whether the FEF were included in the acquired functional volume for each subject. This revealed that in 7 out of the 20 participants who were included in the grid-like code analysis, the FEF were not fully covered. Since this would have left us with a sample of $N = 13$, which does not allow for a reliable group analysis[72], we did not perform this analysis for the Vienna data set.

**Quantification and statistical analysis**
Statistical analysis was carried out in MATLAB (The Mathworks, Natick, MA, USA, R2020b) and R (software version 4.3.0; https://www.r-project.org) using a set of correlations and Wilcoxon-tests (R stats 3.6.2). Unless stated otherwise, effect sizes were tested using Cohen's $d$, and an α-level of 0.05 (two-sided) was adopted.

**Reporting summary**
Further information on research design is available in the Nature Portfolio Reporting Summary linked to this article.

## Data availability
Donders study: Raw, anonymized data will be made publicly available upon completion of orthogonal projects that rely on the same data set. Until then, raw data are available upon request to the corresponding authors (isabella.wagner@univie.ac.at, luise.philine.graichen@univie.ac.at) in accordance with the requirements of the institute, the funding body, and the institutional ethics board. Vienna study: Raw, anonymized fMRI and eye tracking data are available upon request to the corresponding authors (isabella.wagner@univie.ac.at, luise.philine.graichen@univie.ac.at). At present, participant informed consent does not allow for depositing the full data set.

Source data to reproduce figures and tables of both studies (behavioral performance, ROI-based grid-like code results, and (un-)thresholded statistical whole-brain fMRI maps) are openly available at the Open Science Framework (https://osf.io/vp7t3/). Source data are provided with this paper.

## Code availability

All analysis is based on openly available software or custom code, which can be accessed at the Open Science Framework (https://osf.io/vp7t3/).

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

## Acknowledgements

This research was funded in part by the Austrian Science Fund (FWF) [10.55776/P34775], awarded to I.C.W. For open access purposes, the author has applied a CC BY public copyright license to any author-accepted manuscript version arising from this submission. T.S. was funded by the European Union's Horizon 2020 research and innovation programme (https://ec.europa.eu/programmes/horizon2020/; grant number 661373) and by the European Research Council (https://erc.europa.eu/, Starting Grant 802681).

## Author contributions

Conceptualization: L.P.G. and I.C.W.; Methodology: I.C.W., O.J. and C.F.D; Software: I.C.W.; Validation: L.P.G. and I.C.W.; Formal Analysis: L.P.G. and I.C.W.; Investigation: L.P.G., M.S.L., L.K. and T.S.; Resources: I.C.W., C.L. and T.S.; Data Curation: L.P.G., I.C.W. and T.S.; Writing – Original Draft: L.P.G.; Writing – Reviewing & Editing: all authors; Visualization: L.P.G.; Supervision: I.C.W. and T.S.; Project Administration: I.C.W.; Funding Acquisition: I.C.W. and T.S.

## Competing interests

The authors declare no competing interests.
