## [Peer Review file · Nature Communications]

Entorhinal grid-like codes for visual space during memory formation

Corresponding Author: Ms Luise Graichen

Version 0:

Reviewer comments:

Reviewer #1

(Remarks to the Author)

This article presents evidence for grid like codes in human visual processing and potentially for their use in memory. However, I am not sure if the main claim is well supported (see major comment 1). Possibly the chosen stimulus set is ill-suited to test the claims. In addition, two thirds of the figures are essentially replications of prior findings/methods: more saccades lead to better recognition (fig1) and six fold symmetry as markers of grid coding (fig2). the former is a very old known finding, the latter, while newer, has been reported several times by now. Figure 3 is an interesting finding. However, overall it seems a few interesting analyses that could have been done were omitted (see major comment 2). Supplementary figures are mainly controls. As it stands there is limited novelty and not a lot of material here. The results section is fairly thin compared to other Nature Comms articles. To be fair, I do think these findings should be out there, but as is the paper seems not a good fit for nature comms.

Major comments:

1. The main claim is that the present findings are the first to show that saccade-based grid-like codes in the entorhinal cortex play a role in human memory formation. However the authors themselves state that given the negative correlation between behavior and brain measure a possible explanation is that participants who were better at recognizing the scene images may have used a different strategy to complete the task. This goes against their own claims. Yes, adaptation could be a factor, but the literature is split on this (see L300 and following), and if the grid coding supported memory we should the FEF correlation scale with the presumed use of grids. But the trend is reversed. FEF activity correlates positively with the grid signal, but negatively with d-prime (the latter correlating negatively with the grid signal). This goes counter to the claim of the use of grids in memory formation. L273: "larger saccade-based grid-like codes were coupled to increased activation specifically in the left frontal eye fields". If a different strategy (not grid related) was used, then this is not evidence for the use of grid like codes. If it is, then there must another explanation for the decrease in activity. It seems likely that if a different strategy was used, this could be due to the fact that the stimulus set seems sub-optimal to probe relational processing and thus the need for grid codes. Judging from Figure 1A it seems the stimuli can be distinguished easily based on global features without taking into account relations among different locations in a scene. E.g., the pool is easily distinguished from the forest. The same could hold for face. The authors should clarify if there were sufficiently similar foils in the recognition phase that would necessitate relational discrimination (e.g., 2 forest scenes, one old, one as foil). Moreover this should have been present at training in order to force the use of relational processing.

2. Regarding the presented results: Hexagonal modulation is not a new result by itself any more. Similarly, by itself, the finding that more saccades in the study phase lead to better recognition is again not novel, far from it. See for instance as far back as 1972:

LOFTUS, Geoffrey R. Eye fixations and recognition memory for pictures. *Cognitive psychology*, 1972, 3. Jg., Nr. 4, S. 525-551.

Related to comment 1, there are many analyses the authors could have done to probe the claim of relational processing further (see below). The analyses as are leave a lot of potentially interesting results on the table. Saccades should return to the locations fixated at study (within some margin), particularly when expecting relational processing. See for instance:

WYNN, Jordana S.; RYAN, Jennifer D.; BUCHSBAUM, Bradley R. Eye movements support behavioral pattern completion. *Proceedings of the National Academy of Sciences*, 2020, 117. Jg., Nr. 11, S. 6246-6254.

JOHANSSON, Roger, et al. Eye-movement replay supports episodic remembering. *Proceedings of the Royal Society B*, 2022, 289. Jg., Nr. 1977, S. 20220964.

A much more fine-grained analysis could be conducted here, though I am not sure the task and stimuli allow for it. Do participants fixate the same locations as during study (possibly in the same sequence) when successfully recognizing and not otherwise? That would be consistent with relational processing and hence the productive use of the grid code. Moreover, partial occlusions as in the Wynn paper are much more informative, because such test stimuli allow for the disentanglement of 1. genuine memory guidance (presumably via grid cells if we believe the model cited) and 2. salience attracting eye movements (presumably to the same regions at study and test). The latter could be the case in the present studies, both during training and test due to the nature of the stimuli, which would make it possible to solve the task without relational processing. There is no exploration these aspects and it seems the stimulus set was not chosen accordingly.

3. The result of lower saccade-based grid-like codes coupled to successful memory formation is only found in the discovery study - despite non-significant trends. That is, the verification study does not verify.

4. The most important contribution here seems to be the time locking to frontal eye fields. Larger saccade-based grid-like codes were coupled to increased activation specifically in the left frontal eye fields. This is consistent with grid codes informing eye-movements. How does this relate to decreased grid like signals for recognition (see comment 1 too)? Also, was this analysis not done for the validation study? Finally, while we can rationalize increased grid like signals on axis vs off-axis, shouldn't the FEF be equally active for all eye-movement vectors regardless of grid axis? The distribution of grid activity should not matter to the FEF, as long as that grid activity informs target location. It seems unreasonable to suggest FEF and other saccade related activity to code saccades in such a non-uniform way. Why should the FEF be more active for on axis vectors?

5. Some hypotheses in the present article were anticipated by the Buffalo perspective that is cited (L60/61) and these +additional hypotheses were formulated precisely in the Bicanski model, including coupling to the FEF. With the caveat that their model may or may not apply to faces, which are known to benefit from holistic processing. Regardless of this caveat, if prior work - that the authors already cite and hence are aware of - provided inspiration for the present hypotheses, then that should be stated accordingly.

Minor:

Very few minor comments as the manuscript is generally well written and mostly clear:

Why would the average d-prime be so significantly different in a second cohort that performs the same recognition task? It is mentioned later in the text, but the question pops into one's mind immediately when looking at figure 1. Maybe a sentence on this could be included in the main text or caption when this is first discussed.

How the two independent studies differed could be clarified earlier. It described in the Materials and methods, but an extra sentence in the main text will be helpful to avoid interrupting the reading flow.

"A possible explanation for the negative brain-behavior result is that participants who were better at recognizing the scene images may have used a different strategy to complete the task. Rather than relying only on visuospatial encoding, they may have drawn upon prior knowledge. For instance, when viewing a specific beach scene, they might have recalled a recent beach vacation with highly similar scene features. ..."

This suggestion of memory by proxy is a bit odd. Schemas may have a positive effect, e.g., in an office scene I might look for the computer always on the desk. Hence the schema can inform eye movements and this should apply at study and at test.

Reviewer #2

(Remarks to the Author)

The authors present findings from two independent studies regarding grid-like codes in the human entorhinal cortex using fMRI and a scene encoding and recognition memory task. Saccade-related grid-like responses were observed in the left entorhinal cortex during encoding, were time-locked to activation in the left frontal eye field, and were negatively associated with recognition memory (more grid-like codes were associated with lower recognition performance).

This well-written paper builds nicely on the existing literature from human and non-human animals, and the authors prior work regarding hexadirectional modulation of the fMRI signal, and directly links saccade-based grid-like codes in the entorhinal cortex to the frontal eye fields (a region of cognitive oculomotor control) and to memory performance. The authors are careful to note that these grid-like codes are not merely linked to the number of saccades, and do not occur everywhere in the brain; instead they appear to be specific to the entorhinal cortex. The authors also nicely replicate other, simpler, findings from the literature, specifically, as the relationship between the number of saccades and subsequent memory (although perhaps a reference could be added to note this connection to the broader literature).

One part of the paper that was less convincing is the notion that grid-like codes "contribute to memory formation" or, as noted in the Abstract, "supporting memory formation", when there is a negative relationship between grid codes and subsequent

recognition memory (although I appreciate that the authors spend some considerable effort discussing this perhaps puzzling finding). Could one not argue the opposite, that grid-like codes are detrimental to memory formation since higher saccade-based grid-like codes are associated with worse subsequent memory? With respect to possible explanations for the negative relationship, it's not obvious that recall or integration of a schema would necessarily result in a lower grid-cell code, especially if a schema contains information regarding a typical spatial layout of a given scene. Also, if the scenes are fairly typical, it is likely that they automatically invoke the use of schemas. The argument regarding 'efficiency' of encoding is not as convincing as it would have been if the number of saccades had been related to the extent of a grid-like response. I do not have better interpretations to offer the authors regarding the negative relationship, but the careful wording that the codes 'contribute to memory formation' or 'play a role in human memory formation' would seem to hide the nuance of this relationship and perhaps not be a wholly correct interpretation.

Reviewer #3

(Remarks to the Author)

Overall summary:

This manuscript investigates the role of saccade-based grid-like codes in the human entorhinal cortex during memory formation. By analyzing functional MRI (fMRI) and eye-tracking data from two independent studies, the authors report that saccades during scene viewing are associated with grid-like coding patterns in the entorhinal cortex. They further find a negative correlation between the magnitude of these grid-like codes and subsequent recognition memory performance, calculated as a summary statistic for each participant. Additionally, they observe temporal coupling between grid-like codes and activation in the frontal eye fields.

Noteworthy results and significance to the field:

The study presents several noteworthy findings. First, it identifies grid-like coding patterns in the entorhinal cortex associated with saccadic eye movements during scene viewing. Second, it reports a negative correlation between the strength of these grid-like codes and participants' recognition memory performance. Third, it observes temporal coupling between entorhinal grid-like codes and activation in the frontal eye fields, suggesting a functional link between eye movement control and spatial coding.

This paper addresses a critical questions in cognitive and systems neuroscience surrounding the role of grid-like coding in memory formation. It is one of the first studies to explore this relationship in humans, bridging the gap between spatial navigation research and memory processes. The combination of eye tracking with fMRI is technically challenging and requires advanced statistical analyses. The authors have conducted their experiments and analyses with rigor and transparency, potentially providing valuable insights into the neural mechanisms underlying memory formation.

Evaluation of the work:

While the study presents intriguing findings, some major concerns need to be addressed to validate the results and their interpretation.

First, there is an issue with how recognition memory performance was analyzed. The authors computed the d-prime (d') recognition memory performance metric as a single summary statistic for each participant. However, memory encoding and recognition are dynamic processes that can vary significantly across different scenes due to individual experiences and expertise with the stimuli. Grid-like codes should be analyzed on a per-scene basis rather than using an aggregate measure. Specifically, comparing scenes that were later recognized (hits) versus those that were not (misses) would provide a more precise assessment of how grid-like codes relate to memory encoding strength.

Second, regarding the categorization of trial outcomes, the task design yields four categories:

1. Hits: Scenes shown in the study phase and correctly identified as old in the test phase.
2. Misses: Scenes shown in the study phase but incorrectly judged as new in the test phase.
3. Correct Rejections: Scenes not shown in the study phase and correctly identified as new in the test phase.
4. False Alarms: Scenes not shown in the study phase but incorrectly judged as old in the test phase.

Since the BOLD signal analyses focus on the study (encoding) phase, incorporating false alarms—which pertain to stimuli not presented during encoding—into the recognition memory metric may confound the results. Recognition memory performance should be computed using only hits and misses, focusing on scenes that were actually encoded during the study phase. This approach aligns the behavioral data with the neural data analyzed from the study period.

Third, there is a need for clarification in the interpretation of the negative correlation between grid-like codes and memory performance. The finding suggests that stronger grid-like coding is associated with poorer memory recognition, which is counterintuitive and requires careful interpretation. An analysis comparing grid-like codes between strong encoding trials (hits) and weak encoding trials (misses) on a scene-by-scene basis would clarify this relationship. If grid codes are indeed stronger for scenes that are later forgotten, this would support the authors' interpretation and provide a more nuanced understanding of the data.

Fourth, the validation of the eye-tracking data is essential. Accurate eye-tracking data are crucial for linking saccadic movements to neural activity. Eye tracking within an fMRI environment poses additional challenges due to equipment constraints and participant movement. The manuscript should include visualizations of basic eye movement statistics, such as saccade distributions, fixation durations, and calibration accuracy. This information would validate the quality of the eye-tracking data and assure readers of the reliability of the findings.

Methodological soundness:

The authors have employed advanced fMRI analyses and integrated eye-tracking data, demonstrating technical expertise. They have been transparent with methodological details, but additional information on data preprocessing steps, especially for eye-tracking data, would enhance reproducibility. The statistical methods appear appropriate, but re-analysis based on the recommendations above is necessary to strengthen the conclusions.

Conclusions and claims:

The current analysis partially supports the conclusions, but the methodological concerns raised need to be addressed. Re-analyzing the data on a per-scene basis and refining the recognition memory metric are essential steps to confirm the findings.

Summary of recommendations:

I recommend that the authors re-analyze the recognition memory data by computing performance using hits and misses only and perform scene-by-scene analyses to correlate grid-like codes with subsequent memory outcomes. Enhancing the presentation of eye-tracking data by including figures or tables showing saccade amplitudes, directions, and fixation patterns, as well as providing details on eye-tracking calibration procedures and error rates, would greatly improve the manuscript. Clarifying the interpretation of findings, discussing potential reasons for the negative correlation between grid-like codes and memory performance, and exploring alternative explanations or confounding factors that may influence the results are also important.

The manuscript addresses an important topic with the potential to make a significant contribution to the field. However, the concerns regarding the analysis and interpretation of the data need to be addressed. By implementing the recommended revisions, the authors can strengthen their findings and provide clearer insights into the role of grid-like coding in memory formation.

Version 1:

Reviewer comments:

Reviewer #1

(Remarks to the Author)
Comments on the manuscript

I thank the authors for the revision.

The fact that saccade-based grid-like codes were specific to hits is a very interesting addition. Still the inverse correlation is puzzling, but possibly related to less time needed for well remembered items (see below). Overall the authors made a good effort to address all comments. I remain a bit put off by the use of non-significant results to “confirm” interpretation or assess trends. E.g., a $r_{\text{Pearson}} = -0.04$ with $p = 0.4$ is probably meaningless and should not be used to suggest trends (L256). L305 (a similar (albeit non-significant trend) in the Vienna study ($r = -.44$, $p = .056$;) on the other hand is a fair point. That said, I think the core findings are very interesting. The addition of hits vs misses improves the paper substantially. I also welcome the renaming of the studies, which wasn't appropriate before (the verification study did not verify). My main remaining concern is that the stimulus set does not require relational processing, but I accept that the authors did not intend to test it (see some comments below). I think with a few small correction this paper can be published.

A few additional comments:

Abstract:

this is a nitpick but the sentence “Unexpectedly, saccade-based grid-like codes were associated with recognition memory, such that grid-like codes were lower the better participants performed in subsequently recognizing the scene images.” initially reads as if the association between grid codes and memory is unexpected, but it is the direction of the correlation that is unexpected. It is probably better to invert the order. E.g. “Unexpectedly, lower grid codes were associated with ...”. But this is up to authors as this is a very minor point.

Introduction:

Fig1A: nice to see alternative, similar images. Though I would maintain relational processing is not (or less) needed for such images.

Results:

L117: “the average number of saccades during scenes that were later remembered (7.95 ± 0.40) was significantly higher than during scenes that were later forgotten”

L174: “Results confirmed that the magnitude of grid-like codes was not related to saccade numbers”
But the magnitude of grid-like codes inversely correlates with d-prime?

L270 and following: this is nice to see, and suggestive of at least some involvement of visual grid codes, despite the imperfect conditions to test of relational processing within images.

Fig3: in panel C it takes a while to find the “hit rate” label vs “d-prime” label due to the common plot title (“hits”) and the density of the panels.

L344: “the average number of fixations and returns to previously viewed locations (i.e., the revisitations)”
If this is supposed to be informative of memory, revisiting the same image location between study and test should be more informative of memory. Within a trial (as revisitations are defined in Kragel), is also interesting, but more of a short term effect. If revisiting the same locations at test and study correlated correct trials and with 6-fold mod, that would speak to memory involvement. Though of course pure bottom up visual processing could also attract saccades to the same locations at study and test.

Discussion:

L401: “While relational processing has been proposed as a mechanism by which visual grid cells may support memory (Bicanski & Burgess, 2019), we could not confirm this relationship...”
I think this also requires a brief, explicit mention of the stimulus set in the present study, which did not necessarily require relational processing. Also, is the correlation of grid signals with hits (even if negative) in principle compatible with that model? (see also L451 comment below)

L432: see comment above about revisitations

L443: I don’t think that model speaks about schemas. Although if grid cells generalize across scenes that may help.

L451 and following: the efficiency argument is an interesting candidate. The authors could also mention that more effective suppression of alternative saccade targets (for well-remembered items) may explain lower activations.

L467 ...: I guess rather than less effective, it would depend on whether or not the task taxes relational processing at encoding AND recall. Relational encoding may of course also be incidental at first, and relied upon later when needed. I would suspect this to be highly task-dependent.

L481: “Unlike grid-like codes, however, frontal eye field activity showed no clear association with memory performance.” Is this missing in the FEF results section?

Comments on the reply to referees (no line numbers there)

The authors state: “To reiterate, we found that fMRI BOLD activity in the frontal eye fields (FEF) was increased when saccade-based grid-like codes in the entorhinal cortex were increased as well. This suggests that the activity of saccade-based entorhinal grid-like codes was time-locked to activity changes in oculomotor regions as individuals visually explored new content.”

“We therefore politely disagree with the reviewer” I politely accept the correction. The new Discussion text is neutral enough here. I am puzzled though, if d-prime correlates negatively with grid signals, and grid signals correlate positively with FEF, how can d-prime correlate positively with the FEF? I was going to say this suggests the need for a joint statistical treatment. However, the authors state: “This was, contrary to the reviewer’s comment, not the case ($r_{\text{Pearson}} = .05$, $p_{\text{two-tailed}} = .80$).” I want to try to discourage the authors from arguing based on a 0.05 correlation that is not significant.

On relational processing. I see, it was my misunderstanding then that I assumed relational processing and not just establishing whether grid-like codes play some un-defined role in memory formation. However, targeting a possible mechanism: are there other suggestions besides relational processing? What should engage the network of interest more? Or is relational processing still our best guess for entorhinal grid codes? In either case, I very much welcome the revision of Figure 1 to show more similar stimuli, though one could argue they are still easily discernible based on color, perspective, and global features. The discussion should reflect this a bit more clearly. The point I was making in my original review is that if entorhinal cortex performs relational processing in general, and specific to vision possibly in the field of view via visual grid cells, then these stimuli are suboptimal, which of course does not necessarily prevent grid signals, as the Buffalo study with covert attention shows (Wilming et al). This is important to note.

“Additional analysis of revisitations:

Nevertheless, we performed the additional analyses with regard to whether individuals produced saccades that returned to a previously fixated location (revisitations, please see above for a detailed outline of the methodology). In essence, we did not find a significant correlation between the average number of revisitations per trial and recognition memory performance across participants ...”

See similar comment above. If we are looking for long term memory driven effects, revisitations should occur between study and test. Within a trial is less informative. Within a trial a salient part of an image might just as well attract attention repeatedly.

Regarding partial occlusions, that was an example, I was not suggesting the authors perform additional experiments. But the authors nicely cover this in the Discussion now.

A clarification regarding the FEF. I do not suggest that "FEF are equally active for all eye-movement vectors". Of course there are other determinants of eye-movements. However, for those eye-movements primarily instructed by FEF+grids, FEF activity should not depend on the direction of the saccade relative to the grid orientation. This is an odd suggestion because there is no a priori obligation for useful saccade targets to fall on grid axes.

The hits vs misses analysis is interesting and a nice addition.

The explanation for lack of FEF analysis in the second study is satisfactory

Overall, I thank the referees for the great effort in addressing my comments.

Reviewer #2

(Remarks to the Author)

I appreciate the authors' comprehensive responses to my previous concerns; the resulting manuscript has greater transparency and clarity, and is, consequently, much improved.

RESPONSE TO REVIEWERS

Reviewer #1

This article presents evidence for grid like codes in human visual processing and potentially for their use in memory. However, I am not sure if the main claim is well supported (see major comment 1). Possibly the chosen stimulus set is ill-suited to test the claims. In addition, two thirds of the figures are essentially replications of prior findings/methods: more saccades lead to better recognition (fig1) and six fold symmetry as markers of grid coding (fig2). the former is a very old known finding, the latter, while newer, has been reported several times by now. Figure 3 is an interesting finding. However, overall it seems a few interesting analyses that could have been done were omitted (see major comment 2). Supplementary figures are mainly controls. As it stands there is limited novelty and not a lot of material here. The results section is fairly thin compared to other Nature Comms articles. To be fair, I do think these findings should be out there, but as is the paper seems not a good fit for nature comms.

Author reply:

We thank the reviewer for the time that they invested in thoroughly evaluating our work and for the constructive comments to improve it.

Major comments:

Reviewer 1, comment 1

The main claim is that the present findings are the first to show that saccade-based grid-like codes in the entorhinal cortex play a role in human memory formation. However the authors themselves state that given the negative correlation between behavior and brain measure a possible explanation is that participants who were better at recognizing the scene images may have used a different strategy to complete the task. This goes against their own claims. Yes, adaptation could be a factor, but the literature is split on this (see L300 and following), and if the grid coding supported memory we should the FEF correlation scale with the presumed use of grids. But the trend is reversed. FEF activity correlates positively with the grid signal, but negatively with d-prime (the latter correlating negatively with the grid signal). This goes counter to the claim of the use of grids in memory formation. L273: "larger saccade-based grid-like codes were coupled to increased activation specifically in the left frontal eye fields". If a different strategy (not grid related) was used, then this is not evidence for the use of grid like codes. If it is, then there must another explanation for the decrease in activity. It seems likely that if a different strategy was used, this could be due to the fact that the stimulus set seems sub-optimal to probe relational processing and thus the need for grid codes. Judging from Figure 1A it seems the stimuli can be distinguished easily based on global features without taking into account relations among different locations in a scene. E.g., the pool is easily distinguished from the forrest. The same could hold for face. The authors should clarify if there were sufficiently similar foils in the recognition phase that would necessitate relational discrimination (e.g., 2 forrest scenes, one old, one as foil). Moreover this should have been present at training in order to force the use of relational processing.

The reviewer raises several points. We highlighted the individual comments once more to provide a targeted reply:

Reviewer comment: "The main claim is that the present findings are the first to show that saccade-based grid-like codes in the entorhinal cortex play a role in human memory formation. However the authors themselves state that given the negative correlation between behavior and brain measure a possible explanation is that participants who were better at recognizing the scene images may have used a different strategy to complete the task. This goes against their own claims."

Saccade characteristics vs. "strategy use":

Our goal was to provide an explanation for the finding of lower entorhinal grid-like codes being associated with better recognition memory performance. To clarify, it was not our intention to imply that high-performing individuals intentionally employed a specific "strategy" to encode the images, but we wanted to explore whether the negative brain-behavior relationship was associated with differences in eye movements between individuals. We agree with the reviewer that our phrasing of "strategy use" was suboptimal. We now updated our manuscript thoroughly, providing a clear rationale for our reasoning (please see below for the new text passages).

To briefly recapitulate, we reasoned that individuals with better memory performance might have produced saccades with shorter durations during the encoding of the scenes, resulting in lower grid-like coding. This does not negate our hypothesis that grid-like codes are relevant to subsequent recognition memory performance but highlights that the magnitude of grid-like codes may be associated with specific eye

movement characteristics (analogous to path length for grid-like codes during spatial navigation; Wagner et al., 2023).

Our original analysis suggested that individuals with better recognition memory performance did not make shorter saccades than low-performing individuals (correlational analysis between average saccade durations and d' -prime across participants in the Donders study¹, $N = 29$, $r_{\text{Pearson}} = -.07$, $p_{\text{two-tailed}} = .72$). We now performed additional work to provide a more detailed analysis of saccade characteristics. We computed the average saccade frequency (average number of saccades per trial), velocity (average speed of saccades across all trials, in degrees of visual angle per second), and amplitude (average distance of saccades across all trials, in degrees of visual angle) per individual and correlated these measures with the saccade-based grid-like codes obtained from our main analysis (Donders study, bilateral entorhinal cortex, $N = 29$). Results showed non-significant but consistent trends toward higher grid-like codes at lower average saccade frequency, saccade velocity, and saccade amplitude per trial, suggesting that higher saccade-based grid-like codes might be associated with fewer, slower, and shorter eye movements across participants (Figure R-1).

While our initial reasoning was focused on eye movements rather than “strategy use”, it is possible that individuals (intentionally or not) employed specific strategies to encode the scene images. For instance, individuals may have relied on higher-order visual details such as color and the spatial relationships between scene elements (e.g., focusing on leaf color and the spacing between trees in a forest scene) or on semantic representations (e.g., categorizing the scene as a forest in the fall) to support visual memory. This is in line with evidence showing that visual short-term memory contains higher-order visual and semantic information (Liu et al., 2020). Since we did not qualitatively assess whether participants used specific strategies during encoding, we can only speculate about the various strategies they used. We now mention this shortcoming in our updated text (please find the adapted text passages below).

Fig. R-1.: Trend toward higher saccade-based grid-like codes at less frequent, slower, and shorter saccades.

The scatter plots show Pearson correlations between the magnitude of saccade-based entorhinal grid-like codes (Donders data set, $N = 29$) and saccade characteristics such as the mean frequency per trial, and the mean velocity and amplitude across trials ($dva = \text{degree of visual angle}$, $N = 29$, 95% confidence interval CI indicated by the dashed line). Results are not significant (all $p > .05$) but indicate a consistent trend toward a negative relationship.

Additional analysis of visual exploration:

Since our abovementioned analysis of the basic saccade characteristics (frequency, velocity, amplitude) revealed consistent trends toward higher saccade-based grid-like codes being associated with fewer, slower, and shorter eye movements across participants, we were excited to follow this up with a more in-depth analysis of visual exploration. Specifically, we analyzed the average number of fixations and the average number of returns to previously fixated locations per trial (so-called “revisitations”). This analysis aligns with subsequent comments of the same reviewer (comments 3 & 4).

Eye movements play a crucial role in memory formation, as they allow us to actively sample information from the environment for later memory recall. Higher numbers of saccades and fixations have been linked to better recognition memory across participants (Fehlmann et al., 2020; Loftus, 1972; which is a finding that we partly replicated in our work). Similarly, revisitations were linked to better recognition memory (Kragel et al., 2021). It is speculated that revisitations support a thorough encoding process by repeatedly shifting

¹ Please note that we now refer to the “Donders study” (previously: “discovery study”) in line with comment 5 by reviewer 1.

attention to specific elements of a given scene, but whether revisitations relate to saccade-based entorhinal grid-like codes remains unclear.

Methodology: We defined fixations as periods between saccades and blinks, as identified by the EyeLink system. Saccade detection was based on velocity and acceleration thresholds, which were automatically adjusted for each participant based on the signal-to-noise ratio during calibration. The mean threshold values were 78.2°/s for velocity and 241.9°/s² for acceleration. To ensure consistent and meaningful fixation detection across participants, we excluded fixations with a duration shorter than 100 ms (Hannula, 2010). In a next step, we quantified the number of revisitations per trial. We applied the “CalcRets” function from the “PyEyeSim” package (<https://github.com/jozsarato/PyEyeSim>), which identified fixations that returned to a previously visited location within a threshold of 1° of visual angle. To ensure that revisitations reflected actual return movements rather than incidental proximity to a previously fixated location, the current fixation was required to be closer to the revisited location than the immediately preceding fixation.

First, we examined the relationship between visual exploration (fixations, revisitations) and recognition memory performance across participants (analysis was performed on the Donders data set, $N = 32$; note that sample size varied between analyses due to analysis-specific inclusion criteria; analyses of memory and eye tracking data relied on the full available sample ($N = 32$), while for the analysis of grid-like codes participants with extreme values (± 3 median absolute deviation, MAD) were excluded, yielding $N = 29$). Correlations between the average number of fixations and revisitations per trial and recognition memory performance (d -prime) were not significant (Figure R-2, A; we also present our original finding showing a significantly positive association between the average saccade frequency per trial and d -prime across participants) but indicated a positive relationship between visual exploration and memory performance. Although theoretically linked to saccade frequency, fixation frequency did not show the same significant relationship with d -prime. This may be due to the 100-ms fixation threshold or blink-related artifacts. Second, we examined the relationship between visual exploration and the magnitude of saccade-based entorhinal grid-like codes (analysis was performed on the Donders data set, $N = 29$). Correlational analyses were not significant (all $p > 0.05$) but were suggestive of a negative relationship between the average saccade frequency per trial (Figure R-2, B), in line with the results pertaining to saccade characteristics above. Overall, these findings reflect inter-individual variability in viewing behavior, which may have contributed to differences in recognition memory performance and entorhinal grid-like coding.

Fig. R-2.: Individual differences in visual exploration in relation to recognition memory performance (d -prime) and saccade-based entorhinal grid-like coding.

The scatter plots display (A) Pearson correlations between visual exploration patterns (saccades, fixations, revisitations) and recognition memory performance (d -prime; two-tailed, $N = 32$, 95% confidence interval CI indicated by the dashed line). Higher mean saccade frequency per trial was associated with higher d -prime. Other results were not significant (all $p > 0.05$) but indicated a consistent trend toward a positive relationship. (B) Pearson correlations between visual exploration patterns and saccade-based entorhinal grid-like coding (two-tailed, $N = 29$, 95% CI indicated by the dashed line). Results were not significant (all $p > 0.05$) but pointed to a negative relationship between saccade frequency and grid-like coding.

Action taken:

We clarified our argument on “strategy use” and included the additional analyses regarding individual differences in saccade characteristics (frequency, velocity, amplitude; **Supplementary Fig. S8**) and visual exploration (saccades, fixations, revisitations; **Supplementary Fig. S9**), and their relationships with recognition memory performance and saccade-based entorhinal grid-like codes into the updated manuscript.

Results (page 10-11, line 320-352):

“No significant relationship between saccade characteristics and entorhinal grid-like codes

To assess whether the observed negative relationship between saccade-based entorhinal grid-like codes and recognition memory performance could be attributed to individual differences in saccade characteristics (e.g., participants with better recognition memory might have been able to memorize the scene images by producing saccades with shorter durations, which may have resulted in lower grid-like codes), we extracted the average saccade frequency (the average number of saccades per trial), velocity (the average speed of saccades across all trials in degrees of visual angle per second), and amplitude (the average distance of saccades across all trials in degrees of visual angle) per individual. We then correlated these measures with the saccade-based grid-like codes obtained from our main analysis (Donders study, bilateral entorhinal cortex, $N = 29$). Results did not confirm a significant relationship between grid-like codes and average saccade frequency, velocity, or amplitude (Supplementary information, Figure S8). This suggests that our finding of increased saccade-based grid-like codes appeared unaffected by individual differences in saccade characteristics.

No significant relationship between visual exploration and entorhinal grid-like codes

Next, we went on to conduct a more detailed analysis of visual exploration. Eye movements play a crucial role in memory formation, as they allow us to actively sample information from the environment for later memory recall. Higher numbers of saccades and fixations have been linked to better recognition memory across participants (Fehlmann et al., 2020; Loftus, 1972). Similarly, returns to previously viewed locations (so-called “revisitations”) have been linked to better recognition memory and may support the encoding process by redirecting attention to specific elements of a given scene (Kragel et al., 2021). We reasoned that grid-like codes might be particularly linked to revisitations, supporting the encoding of translational vectors between salient stimulus features, akin to relational processing (Bicanski & Burgess, 2019; Killian & Buffalo, 2018).

To test whether these visual exploration patterns might have contributed to our results, we quantified the average number of fixations and returns to previously viewed locations (i.e., the revisitations) per trial (see Supplementary information for details). We first examined whether the average fixation and revisitation frequencies were related to recognition memory performance (d -prime) across participants (Donders data set, $N = 32$), but this was not the case (Supplementary information, Figure S9A; but note that results indicated a positive trend). We next examined whether the average fixation and revisitation frequencies were associated with the magnitude of saccade-based entorhinal grid-like codes across participants (Donders data set, $N = 29$) but, again, results were not significant (all $p > 0.05$). Overall, these findings show that potential differences in visual exploration between individuals did not significantly impact our results.”

Discussion (page 13-14, line 422-450):

“To better understand the negative brain-behavior relationship observed in our data, we speculated that memory success was associated with differences in eye movements during encoding. For instance, participants with better recognition memory might have been able to memorize the scene images by producing saccades with shorter durations, which may have resulted in lower grid-like codes. While this was not the case, we found consistent trends of higher grid-like codes being associated with lower average saccade frequencies, velocities, and amplitudes (Supplementary information, Figure S8). We encourage future studies to test to what extent grid-like codes are tied to saccade characteristics. Moreover, memory performance might also be influenced by the number of fixations or revisitations. The latter were previously linked to better recognition memory performance and may support thorough encoding by redirecting attention to specific elements of a scene (Kragel et al., 2021). Once more, we could not detect a significant relationship between fixations, revisitations, recognition memory performance, and grid-like codes (Supplementary information, Figure S9A; individuals who made more fixations during encoding showed marginally lower grid-like codes, Supplementary information, Figure S9B). Overall, these findings reflect some inter-individual variability in visual exploration, but these did not significantly contribute to the observed results.

Another possible explanation for the negative brain-behavior result is that participants who were better at recognizing the scene images may have (intentionally or not) used different strategies to complete the task. Rather than relying predominantly on visuospatial encoding, they may have more extensively drawn upon prior knowledge. For instance, when viewing a specific beach scene, they might have recalled a recent beach vacation with highly similar scene features. Such integration of prior knowledge, or memory “schemas”, could have facilitated scene encoding by guiding eye movements to prominent scene features, thereby enhancing participants’ recognition memory (Bicanski & Burgess, 2019). This is consistent with the idea that schema-congruent information is encoded more efficiently and engages the medial prefrontal cortex more than the medial temporal lobes (Bonasia et al., 2018; Van Kesteren et al., 2013). In contrast, participants who relied exclusively on visuospatial encoding may have shown lower recognition memory performance while displaying increased medial temporal lobe activity and higher grid-like codes for visual space. Unfortunately, we did not qualitatively assess participants’ strategy use during encoding, nor did we account for individual experience or expertise with the specific scene stimuli, which could have helped to clarify the role of grid-like codes in memory formation.”

Reviewer comment: “Yes, adaptation could be a factor, but the literature is split on this (see L300 and following), and if the grid coding supported memory we should the FEF correlation scale with the presumed use of grids. But the trend is reversed. FEF activity correlates positively with the grid signal, but negatively with d -prime (the latter correlating negatively with the grid signal). This goes counter to the claim of the use

of grids in memory formation. L273: "larger saccade-based grid-like codes were coupled to increased activation specifically in the left frontal eye fields". If a different strategy (not grid related) was used, then this is not evidence for the use of grid like codes. If it is, then there must another explanation for the decrease in activity."

To reiterate, we found that fMRI BOLD activity in the frontal eye fields (FEF) was increased when saccade-based grid-like codes in the entorhinal cortex were increased as well. This suggests that the activity of saccade-based entorhinal grid-like codes was time-locked to activity changes in oculomotor regions as individuals visually explored new content. Since grid-like codes were negatively associated with d -prime ($r_{\text{Pearson}} = -.51$, $p_{\text{two-tailed}} = .004$; Figure 3A, left panel), we explored whether FEF activity (extracted from the significant cluster) would show a negative association with d -prime as well:

This was, contrary to the reviewer's comment, not the case ($r_{\text{Pearson}} = .05$, $p_{\text{two-tailed}} = .80$).

We therefore politely disagree with the reviewer that these findings stand in contrast to our hypothesis of grid-like codes being related to memory formation. The fact that FEF activity does not show the same relationship with d -prime as entorhinal grid-like codes merely indicates that different mechanisms may be at play and that the relationship between entorhinal cortex and oculomotor regions needs further investigation. For example, one could speculate that high engagement of the FEF is needed to guide saccades in a fashion that supports grid-like coding, hence the positive correlation. Engagement of the FEF might thus be particularly necessary in participants who show comparably poorer memory performance. However, it is also plausible to assume that the FEF guide saccades in participants with better memory performance who do not show high grid-like coding. This could cancel a potential negative relationship with d -prime. It has been speculated that the FEF are involved in generating grid-like codes for visual space (Bicanski & Burgess, 2019), but their precise involvement in guiding saccades for a particular coding scheme – grid-like coding or otherwise – is not known and remains to be further elucidated. Our results thus provide first data on this relationship. We updated our manuscript to provide a more detailed discussion regarding this point.

Discussion (page 14, line 471-483):

"We next hypothesized that saccade-based grid-like codes in the entorhinal cortex were tied to neural activity in visuo-oculomotor regions that are known to be involved in saccade generation and coordination, such as the visual cortex or frontal eye fields (Prime et al., 2010). Here, we built on previous work proposing that memory-guided saccades may result from coordinated activity between medial temporal and visuo-oculomotor regions, supported by entorhinal grid-like activity (Bicanski & Burgess, 2019; Killian & Buffalo, 2018). Indeed, we observed that increased grid-like codes in the entorhinal cortex were related to an increased activation in the frontal eye fields. In other words, if participants made saccades that were aligned with their individual grid orientation, activation in the frontal eye fields was high as well, possibly reflecting an interaction between oculomotor processes and grid-like activity. Interestingly, frontal eye field activation also tended to be higher in participants who made fewer saccades overall. Unlike grid-like codes, however, frontal eye field activity showed no clear association with memory performance. This may indicate that different mechanisms are at play and that the relationship between the entorhinal cortex and oculomotor regions needs further investigation."

Reviewer comment: *"It seems likely that if a different strategy was used, this could be due to the fact that the stimulus set seems sub-optimal to probe relational processing and thus the need for grid codes. Judging from Figure 1A it seems the stimuli can be distinguished easily based on global features without taking into account relations among different locations in a scene. E.g., the pool is easily distinguished from the Forrest. The same could hold for face. The authors should clarify if there were sufficiently similar foils in the recognition phase that would necessitate relational discrimination (e.g., 2 Forrest scenes, one old, one as foil). Moreover this should have been present at training in order to force the use of relational processing."*

Relational processing:

First, we would like to clarify that this study did not aim to probe relational processing, that is, the integration of between-feature relations to inform the encoding and retrieval of visual stimuli (Bicanski & Burgess, 2019). While we agree with the reviewer that relational processing likely serves as the underlying mechanism linking grid-like coding and memory formation, our aim was to establish whether saccade-based entorhinal grid-like codes play a role in memory formation. We apologize if the reviewer caught a different impression, and we thoroughly revised our manuscript to avoid any further misunderstandings (please find the new text passages below).

Stimulus set:

Second, both data sets were based on the same stimulus set that included 300 scene images, drawn from multiple semantic categories, each represented by several images. Participants encoded 200/96 (Donders/Vienna) of these scene images during the study period and were asked to dissociate them from 100/48 (Donders/Vienna) new scenes during the test period. The new scenes did not belong to entirely

different categories but contained similar as well as dissimilar scenes (e.g., comparisons were made between an old forest scene and a new forest scene or between a forest and a pool scene). Hence, recognition judgments were based on scenes with shared visual elements. We adapted Figure 1 in the main manuscript to make this clear (Figure R-3) and updated the manuscript (new text passages below). Please note that face stimuli were not included in the analysis since they were only presented in the Vienna study², but not in the Donders study.

Fig. R-3: Stimulus set and task design of the recognition memory task.

(A) Examples of scenes that participants viewed during the study period (“old” scenes) and that were shown during the test period, together with novel scenes (“old” and “new” scenes). Note that scenes presented during study and test periods varied in similarity, with some differing more clearly, like a forest and a beach scene, or a sports hall and a pool. (B) Example of how the scene images were integrated into the recognition memory task. During the test period, the first image would be correctly classified as “old”, whereas the second image would be correctly classified as “new”. The figure panels were incorporated into the updated Figure 1 in the main manuscript.

Action taken:

We clarified that relational processing was not the primary aim of our study and revised the manuscript accordingly. Additionally, we adapted Figure 1 to better illustrate the visual similarities between “old” and “new” scene images presented in the recognition memory task.

Discussion (page 13, line 401-405):

“While relational processing has been proposed as a mechanism by which visual grid cells may support memory (Bicanski & Burgess, 2019), we could not confirm this relationship. However, since our studies were not specifically designed to test this mechanism, we encourage future work to interrogate the relationship between relational processing and saccade-based entorhinal grid-like codes during memory formation.”

Donders study

Methods (page 16, line 571-576):

“During the test period, participants viewed all scene images that were shown during the previous study period, intermixed with 100 novel scene images (half of them indoor/outdoor; i.e., a total of 300 scene images were presented). Some of the novel scene images were similar to the previously studied ones (e.g., comparison between a new and a previously studied forest scene) or could be markedly different (e.g., comparison between a forest and a beach scene). Hence, recognition judgments were based on scenes with shared visual elements rather than on entirely separate scene categories.”

Vienna study

Methods (page 21, line 757-762):

“After each study period, participants completed a test period (i.e., the immediate test) where the 96 previously viewed (“old”) images were pseudo-randomly interleaved with 48 novel (“new”) images, with the constraint that no more than three images of the same image category (scene or face) or the same memory condition (“old” or “new”) were presented in succession. Previously

² Please note that we now refer to the “Vienna study” (previously: “validation study”) in line with comment 5 by reviewer 1.

Rebuttal letter, Graichen et al.

viewed and novel images could share visual elements (e.g., comparison between two forest scenes), while some were markedly different (e.g., comparison between a forest and a beach)."

Reviewer 1, comment 2

Regarding the presented results: Hexagonal modulation is not a new result by itself any more. Similarly, by itself, the finding that more saccades in the study phase lead to better recognition is again not novel, far from it. See for instance as far back as 1972:

LOFTUS, Geoffrey R. Eye fixations and recognition memory for pictures. *Cognitive psychology*, 1972, 3. Jg., Nr. 4, S. 525-551.

Hexagonal modulation:

By no means did we intend to highlight hexagonal modulation as a new result *per se*. We would like to emphasize, however, that our results of saccade-based entorhinal grid-like codes during human memory formation are entirely novel and have not been reported previously. We adapted our revised manuscript to make this clear.

Results (page 8, line 222-236):

"Our main goal was to clarify the role of saccade-based grid-like codes in memory formation. While hexadirectional modulation of the fMRI signal in the entorhinal cortex is a well-established finding (e.g., Doeller et al., 2010) and has been observed in relation to saccade activity during free visual search (Julian et al., 2018; Nau, Navarro Schröder, et al., 2018), a direct link to human memory formation is missing. Previous studies reported mixed results regarding the relationship between entorhinal grid-like codes and behavior. In short, increased grid-like codes were associated with better spatial navigation performance of human participants in an object-location memory task (i.e., lower drop error when placing an object at its correct location; Doeller et al., 2010; Kunz et al., 2015; Stangl et al., 2018). When observing a virtual demonstrator, increased grid-like codes were associated with lower navigation performance (Wagner et al., 2023), and a similarly negative relationship was found for directional coding in the human medial temporal lobe (Nau et al., 2020). Saccade-based grid-like codes were shown to be positively associated with self-reported navigation ability (Julian et al., 2018). In non-human primates, only some of the detected visual grid cells showed neural adaption (i.e., decreased activation) upon stimulus repetition (Killian et al., 2012; Meister & Buffalo, 2016), leaving it open whether saccade-based entorhinal grid-like codes are related to human memory formation as well."

Saccade-memory relationship:

As the reviewer states correctly, the finding of more saccades during the study period being associated with better recognition memory has been reported previously. We reported this result as it serves as an important first step that provides the basis for our investigation of saccade-based grid-like codes and their relationship with memory performance. We made sure to provide our reasoning and clearly linked it to the relevant prior works.

Results (page 4-5, line 112-124):

"To provide the basis for our subsequent investigation of the relationship between saccade-based grid-like codes and recognition memory performance, we first assessed the link between eye movements and memory formation. Consistent with previous literature on the role of eye movements in memory encoding (Fehlmann et al., 2020; Loftus, 1972; Meister & Buffalo, 2016; Olsen et al., 2016), the average number of saccades during scenes that were later remembered (7.95 ± 0.40) was significantly higher than during scenes that were later forgotten (6.64 ± 0.43 ; paired-sample t-test, $N = 32$; $t(31) = 8.262$, Cohen's $d = 1.46$, 95% confidence interval (CI) = [0.99, 1.63], ptwo-tailed < 0.0001). In line with these results, we observed a positive correlation between individual recognition memory performance (d-prime, 1.82 ± 0.11 ; Figure 1C, left panel) and the number of saccades made when studying later remembered scenes (rPearson = 0.51, 95% CI = [0.20, 0.73], ptwo-tailed = 0.003; Figure 1D). Thus, participants who used more saccades to explore later remembered scenes performed better in distinguishing old from novel material during the subsequent recognition memory test."

Reviewer 1, comment 3

Related to comment 1, there are many analyses the authors could have done to probe the claim of relational processing further (see below). The analyses as are leave a lot of potentially interesting results on the table. Saccades should return to the locations fixated at study (within some margin), particularly when expecting relational processing. See for instance:

WYNN, Jordana S.; RYAN, Jennifer D.; BUCHSBAUM, Bradley R. Eye movements support behavioral pattern completion. *Proceedings of the National Academy of Sciences*, 2020, 117. Jg., Nr. 11, S. 6246-6254.

JOHANSSON, Roger, et al. Eye-movement replay supports episodic remembering. *Proceedings of the Royal Society B*, 2022, 289. Jg., Nr. 1977, S. 20220964.

Relational processing:

We agree with the reviewer that there are many analyses that could be performed to probe relational processing, but we would like to highlight that this was not the initial focus of our work. We thoroughly revised our manuscript to clarify this and to avoid further misunderstandings (please find the text passages below).

Additional analysis of revisitations:

Nevertheless, we performed additional analyses with regards to whether individuals produced saccades that returned to a previously fixated location (so-called “revisitations”). Revisitations were shown to support recognition memory by repeatedly shifting attention to specific elements of a given scene, which may reflect relational processing (Kragel et al., 2021). To assess whether participants revisited scene locations during scene encoding, we identified within-trial fixations that landed on the same location (within 1° of visual angle) more than once. The precise methodological details resonate with our reply to comment 1 by the same reviewer.

Methodological details: Fixations were defined as periods between saccades and blinks, as identified by the EyeLink system. Saccade detection was based on velocity and acceleration thresholds, which were automatically adjusted for each participant based on the signal-to-noise ratio during calibration. The mean threshold values were 78.2°/s for velocity and 241.9°/s² for acceleration. To ensure consistent and meaningful fixation detection across participants, we excluded fixations with a duration shorter than 100 ms (Hannula, 2010). In a next step, we quantified the number of revisitations per trial. We applied the “CalcRets” function from the “PyEyeSim” package (<https://github.com/jozsarato/PyEyeSim>), which identified fixations that returned to a previously visited location within a threshold of 1° of visual angle. To ensure that revisitations reflected actual return movements rather than incidental proximity to a previously fixated location, the current fixation was required to be closer to the revisited location than the immediately preceding fixation.

We correlated the average number of revisitations per trial with individual recognition memory performance (*d*-prime) across participants (Figure R-2A). While the correlation did not reach significance, the result showed a trend toward a positive relationship between revisitations and *d*-prime, suggesting that participants may have revisited image locations in support of later memory. This result may differ from previous findings due to differences in task instructions, which prompted scene recognition rather than explicit spatial recall.

The reviewer raises an interesting point regarding the return to locations (during the test period) that were fixated during the study period. A higher number of revisitations during encoding was linked to gaze reinstatement during retrieval across individuals, indicative of spatiotemporal memory (Kragel et al., 2021). We refrained from performing this analysis because we did not find a significant relationship between revisitations and *d*-prime in the first place, and because this aspect (albeit interesting) stretches beyond the focus of the current manuscript.

Action taken:

We revised our manuscript to clarify the main focus of our work. Additionally, we analyzed revisitations and their relationship to recognition memory performance and integrated the results into the manuscript.

Discussion (page 13, line 401-405):

“While relational processing has been proposed as a mechanism by which visual grid cells may support memory (Bicanski & Burgess, 2019), we could not confirm this relationship. However, since our studies were not specifically designed to test this mechanism, we encourage future work to interrogate the relationship between relational processing and saccade-based entorhinal grid-like codes during memory formation.”

Results (page 11, line 334-352):

“Next, we went on to conduct a more detailed analysis of visual exploration. Eye movements play a crucial role in memory formation, as they allow us to actively sample information from the environment for later memory recall. Higher numbers of saccades and fixations have been linked to better recognition memory across participants (Fehlmann et al., 2020; Loftus, 1972). Similarly, returns to previously viewed locations (so-called “revisitations”) have been linked to better recognition memory and may support the encoding process by redirecting attention to specific elements of a given scene (Kragel et al., 2021). We reasoned that grid-like codes might be particularly linked to revisitations, supporting the encoding of translational vectors between salient stimulus features, akin to relational processing (Bicanski & Burgess, 2019; Killian & Buffalo, 2018).

To test whether these visual exploration patterns might have contributed to our results, we quantified the average number of fixations and returns to previously viewed locations (i.e., the revisitations) per trial (see Supplementary information for details). We first examined whether the average fixation and revisitation frequencies were related to recognition memory performance (*d*-prime) across participants (Donders data set, *N* = 32), but this was not the case (Supplementary information, Figure S9A; but note that results indicated a positive trend). We next examined whether the average fixation

Rebuttal letter, Graichen et al.

and revisitatio frequencies were associated with the magnitude of saccade-based entorhinal grid-like codes across participants (Donders data set, $N = 29$) but, again, results were not significant (all $p > 0.05$). Overall, these findings show that potential differences in visual exploration between individuals did not significantly impact our results.”

Discussion (page 13, line 428-436):

“Moreover, memory performance might also be influenced by the number of fixations or revisitations. The latter were previously linked to better recognition memory performance and may support thorough encoding by redirecting attention to specific elements of a scene (Kragel et al., 2021). Once more, we could not detect a significant relationship between fixations, revisitations, recognition memory performance, and grid-like codes (Supplementary information, Figure S9A; individuals who made more fixations during encoding showed marginally lower grid-like codes, Supplementary information, Figure S9B). Overall, these findings reflect some inter-individual variability in visual exploration, but these did not significantly contribute to the observed results.”

Reviewer 1, comment 4

A much more fine-grained analysis could be conducted here, though I am not sure the task and stimuli allow for it. Do participants fixate the same locations as during study (possibly in the same sequence) when successfully recognizing and not otherwise? That would be consistent with relational processing and hence the productive use of the grid code. Moreover, partial occlusions as in the Wynn paper are much more informative, because such test stimuli allow for the disentanglement of 1. genuine memory guidance (presumably via grid cells if we believe the model cited) and 2. salience attracting eye movements (presumably to the same regions at study and test). The latter could be the case in the present studies, both during training and test due to the nature of the stimuli, which would make it possible to solve the task without relational processing. There is no exploration these aspects and it seems the stimulus set was not chosen accordingly.

Relational processing:

The reviewer touches on a similar topic as in comment 3, which is why our reply resonates with the previous response. We would like to highlight that probing relational processing was not the focus of our work. We thoroughly revised our manuscript to clarify this and avoid any further misunderstandings (please find the text passages below).

Additional analysis of revisitations:

Nevertheless, we performed the additional analyses with regard to whether individuals produced saccades that returned to a previously fixated location (revisitations, please see above for a detailed outline of the methodology). In essence, we did not find a significant correlation between the average number of revisitations per trial and recognition memory performance across participants (Figure R-2A). This result may differ from previous findings due to differences in task instructions, which prompted scene recognition rather than explicit spatial recall. We included these results in our updated manuscript and integrated the interpretation into the discussion (please see text passages below).

Partial occlusions:

Wynn et al. (2020) tracked participants' eye movements during a recognition memory task and showed that gaze behavior was reinstated at retrieval, even when parts of the scenes were occluded, indicating that gaze reinstatement is guided by memory. We agree that partial occlusions might yield additional information on how eye movements guide memory (e.g., as to whether memory impacts eye gaze returns to partially occluded image parts) and salience (e.g., as to whether eye gaze changes when salient image parts are occluded). However, partial occlusions also add additional processes to a memory test like visual search (when searching for the occluded target during the test), that may interfere with the main research question of the paper, which was to investigate the relationship between grid-like coding in the encoding phase and memory performance as assessed in a recognition memory test.

Future studies involving partial occlusions (e.g., Wynn et al., 2020) or analyses of eye movement reinstatement (Johansson et al., 2022) could help explain whether relational processing mediates the relationship between memory formation and grid-like codes. We have now incorporated this aspect in our updated discussion section.

Discussion (page 13, line 401-405):

“While relational processing has been proposed as a mechanism by which visual grid cells may support memory (Bicanski & Burgess, 2019), we could not confirm this relationship. However, since our studies were not specifically designed to test this mechanism, we encourage future work to interrogate the relationship between relational processing and saccade-based entorhinal grid-like codes during memory formation.”

Results (page 11, line 334-352):

“Next, we went on to conduct a more detailed analysis of visual exploration. Eye movements play a crucial role in memory formation, as they allow us to actively sample information from the environment for later memory recall. Higher numbers of saccades and fixations have been linked to better recognition memory across participants (Fehlmann et al., 2020; Loftus, 1972). Similarly, returns to previously viewed locations (so-called “revisitations”) have been linked to better recognition memory and may support the encoding process by redirecting attention to specific elements of a given scene (Kragel et al., 2021). We reasoned that grid-like codes might be particularly linked to revisitations, supporting the encoding of translational vectors between salient stimulus features, akin to relational processing (Bicanski & Burgess, 2019; Killian & Buffalo, 2018).

To test whether these visual exploration patterns might have contributed to our results, we quantified the average number of fixations and returns to previously viewed locations (i.e., the revisitations) per trial (see Supplementary information for details). We first examined whether the average fixation and revisitation frequencies were related to recognition memory performance (d-prime) across participants (Donders data set, N = 32), but this was not the case (Supplementary information, Figure S9A; but note that results indicated a positive trend). We next examined whether the average fixation and revisitation frequencies were associated with the magnitude of saccade-based entorhinal grid-like codes across participants (Donders data set, N = 29) but, again, results were not significant (all $p > 0.05$). Overall, these findings show that potential differences in visual exploration between individuals did not significantly impact our results.”

Discussion (page 13, line 428-436):

“Moreover, memory performance might also be influenced by the number of fixations or revisitations. The latter were previously linked to better recognition memory performance and may support thorough encoding by redirecting attention to specific elements of a scene (Kragel et al., 2021). Once more, we could not detect a significant relationship between fixations, revisitations, recognition memory performance, and grid-like codes (Supplementary information, Figure S9A; individuals who made more fixations during encoding showed marginally lower grid-like codes, Supplementary information, Figure S9B). Overall, these findings reflect some inter-individual variability in visual exploration, but these did not significantly contribute to the observed results.”

Discussion (page 15, line 514-519):

“Bicanski and Burgess proposed a computational model in which grid cells encode trajectories between salient stimulus features of a visual scene, offering a possible explanation for their role in memory encoding (Bicanski & Burgess, 2019). Future studies involving partial occlusions (e.g., Wynn et al., 2020) or an analysis of eye movement reinstatement (Johansson et al., 2022) could help explain whether relational processing mediates the relationship between memory formation and grid-like codes.”

Reviewer 1, comment 5

The result of lower saccade-based grid-like codes coupled to successful memory formation is only found in the discovery study - despite non-significant trends. That is, the verification study does not verify.

We thank the reviewer for raising this important point, and we agree that the terminology was suboptimal. We initially chose this term to emphasize that our main finding of saccade-based entorhinal grid-like coding was observed in two independent data sets and tried to highlight any discrepancies between the two studies in the text. To avoid further confusion, we have now changed the terminology and instead refer to the “Donders study” (previously: “discovery study”) and the “Vienna study” (previously: “validation study”). We adapted this throughout (text parts not shown here).

Reviewer 1, comment 6

The most important contribution here seems to be the time locking to frontal eye fields. Larger saccade-based grid-like codes were coupled to increased activation specifically in the left frontal eye fields. This is consistent with grid codes informing eye-movements. How does this relate to decreased grid like signals for recognition (see comment 1 too)? Also, was this analysis not done for the validation study? Finally, while we can rationalize increased grid like signals on axis vs off-axis, shouldn't the FEF be equally active for all eye-movement vectors regardless of grid axis? The distribution of grid activity should not matter to the FEF, as long as that grid activity informs target location. It seems unreasonable to suggest FEF and other saccade related activity to code saccades in such a non-uniform way. Why should the FEF be more active for on axis vectors?

We thank the reviewer for raising these points. Below, we present the individual comments along with our stepwise responses.

Reviewer comment: *The most important contribution here seems to be the time locking to frontal eye fields. Larger saccade-based grid-like codes were coupled to increased activation specifically in the left frontal eye fields. This is consistent with grid codes informing eye-movements. How does this relate to decreased grid-like signals for recognition (see comment 1 too)?*

Relationship between FEF activity and recognition memory performance:

To recapitulate, we found increased activation in the left frontal eye fields (FEF) in trials where saccade-based entorhinal grid-like codes were increased as well. Additionally, we found a negative relationship between grid-like codes and recognition memory performance (d -prime), but the relationship between FEF activity and d -prime was not significant ($r_{\text{Pearson}} = .05$, $p_{\text{two-tailed}} = .80$). This may suggest that the frontal eye fields and the entorhinal cortex play distinct roles in memory formation.

We do not agree that one can assume that the FEF are equally active for all eye-movement vectors. While the FEF are an important hub in the oculomotor system, regions like the parietal eye fields, superior colliculus, primary visual cortex, basal ganglia, supplementary eye fields, and others are also important in planning and executing eye movements. When it comes to planning saccades, the FEF have been highlighted in prior research (Bruce & Goldberg, 1985; Schall, 1991; Vernet et al., 2014), but whether they are equally engaged or modulated by encoding strategies, is, to the best of our knowledge, not known.

For example, one could speculate that high engagement of the FEF is needed to guide saccades in a fashion that supports grid-like coding, hence the positive correlation. In other words, it could be speculated that the entorhinal grid system actively engages the FEF to enforce saccade vectors aligned to the grid axes. Engagement of the FEF might thus be particularly necessary in participants who show comparably poorer memory performance. However, it is also plausible to assume that the FEF guide saccades in participants with better memory performance who do not show high grid-like coding. In this scenario, the FEF would not be engaged by a system other than the entorhinal grid system. This could cancel a potential negative relationship with d -prime. It has been speculated that the FEF are involved in generating grid-like codes for visual space (Bicanski & Burgess, 2019), but their precise involvement in guiding saccades for a particular coding scheme – grid-like coding or otherwise – is not known and remains to be further elucidated. Our results thus provide first data on this relationship.

To further explore this, and in line with comment 2 by reviewer 3, we also tested the relationship between FEF activity and individual hit rates. While d -prime accounts for both hits and false alarms, the hit rate reflects only how many of the previously studied scene images were later recognized. Results were again not significant but revealed a negative trend ($r_{\text{Pearson}} = -.22$, $p_{\text{two-tailed}} = .22$), suggesting that lower FEF activity may be associated with a higher hit rate (i.e., better performance) across participants. These findings indicate that, although FEF activity co-occurs with saccade-based entorhinal grid-like codes during memory encoding, its role in memory formation remains unclear.

Grid-like codes and eye movement characteristics:

We next examined whether individual differences in eye movement behavior during encoding may account for the negative relationship between grid-like coding and recognition memory. Specifically, we tested whether grid-like coding was associated with different saccade characteristics (frequency, velocity, amplitude) or patterns of visual exploration (fixations and returns to previously visited locations, so-called “revisitations”). Across participants, higher saccade-based grid-like codes tended to coincide with fewer and shorter saccades (grid-like coding vs. saccade frequency: $r_{\text{Pearson}} = -.17$, $p = .39$; velocity: $r_{\text{Pearson}} = -.22$, $p = .26$; amplitude: $r_{\text{Pearson}} = -.24$, $p = .20$, Figure R-1), while the relationship between grid-like codes and different patterns of visual exploration indicated no particular trend (grid-like coding vs. average fixations per trial: $r_{\text{Pearson}} = -.06$, $p = .77$; average revisitations per trial: $r_{\text{Pearson}} = .06$, $p = .78$, Figure R-2B). Overall, these findings suggest that individual differences in viewing behavior may relate to both recognition memory performance and the magnitude of grid-like coding.

Grid-like codes during hits vs. misses:

To further disentangle the relationship between grid-like codes and memory performance, we repeated our main analysis of saccade-based entorhinal grid-like coding during the study period, when participants visually explored the scene images. Analysis was performed separately for later remembered and later forgotten material, allowing us to obtain a grid-like coding value per stimulus category (hits, misses) for each participant (see reviewer 3, comment 1 for a more detailed description of the procedure). Results showed significant grid-like coding during hits ($N = 26$, mean \pm SEM = $.119 \pm .054$, Wilcoxon-test, $V = 237.5$, $p_{\text{one-tailed}} = .023$, Cohen's $d = .43$, Wilcoxon's $r = .40$), but not during misses ($N = 26$, mean \pm SEM = $.050 \pm .126$, Wilcoxon-test, $V = 201$, $p_{\text{one-tailed}} = .26$, Cohen's $d = .08$, Wilcoxon's $r = .13$). Correlating grid-like codes during hits with recognition memory performance (d -prime) revealed a negative, albeit non-significant, trend ($N = 26$, $r_{\text{Pearson}} = -.12$, 95% CI = $[-.483, .282]$, $p_{\text{two-tailed}} = .57$; Figure R-6). In response to a request from reviewer 3, we additionally examined the correlation between grid-like codes during hits and hit rate (reflecting the proportion of correctly recognized scenes as opposed to d -prime, which accounts for hits and

false alarms). This revealed a significant negative relationship, with higher grid-like codes associated with lower hit rates across participants ($N = 26$, $r_{\text{Pearson}} = -.44$, 95% CI = $[-.704, -.059]$, $p_{\text{two-tailed}} = .03$). While these results do not explain the underlying cause of the negative grid-behavior relationship, they further corroborate the stability of our findings.

Alternative explanations:

Possible explanations for the negative grid-behavior relationship include that participants might differ with respect to visuo-spatial, relational processing during encoding (Bicanski & Burgess, 2019). Others could have drawn upon prior knowledge connecting a given scene with a previously visited place (such as connecting the forest scene with a view from a recent hike), which might boost memory (Maguire et al., 1999). Another possibility is that participants who performed well in the recognition memory task displayed less neural activation during memory formation and encoded visual content more efficiently (Neubauer & Fink, 2009). In line with this interpretation, improved memory performance following method of loci training was associated with reduced neural activation in brain regions that are involved in visuo-spatial processing and mental navigation (Wagner et al., 2021). In macaques, long-term training was linked to decreased glucose uptake despite stable neural activity levels (Picard et al., 2013). Unfortunately, we did not qualitatively assess whether participants used specific strategies during encoding. We now mention this shortcoming in our updated text (please find the adapted text passages below).

Action taken:

While the negative brain-behavior relationship could be caused by multiple aspects, we explored a potential association with individual differences in saccade characteristics and visual exploration. We included these new results in the manuscript and updated our discussion:

Discussion (page 14, line 471-483):

"We next hypothesized that saccade-based grid-like codes in the entorhinal cortex were tied to neural activity in visuo-oculomotor regions that are known to be involved in saccade generation and coordination, such as the visual cortex or frontal eye fields (Prime et al., 2010). Here, we built on previous work proposing that memory-guided saccades may result from coordinated activity between medial temporal and visuo-oculomotor regions, supported by entorhinal grid-like activity (Bicanski & Burgess, 2019; Killian & Buffalo, 2018). Indeed, we observed that increased grid-like codes in the entorhinal cortex were related to an increased activation in the frontal eye fields. In other words, if participants made saccades that were aligned with their individual grid orientation, activation in the frontal eye fields was high as well, possibly reflecting an interaction between oculomotor processes and grid-like activity. Interestingly, frontal eye field activation also tended to be higher in participants who made fewer saccades overall. Unlike grid-like codes, however, frontal eye field activity showed no clear association with memory performance. This may indicate that different mechanisms are at play and that the relationship between the entorhinal cortex and oculomotor regions needs further investigation."

Discussion (page 13, line 422-436):

"To better understand the negative brain-behavior relationship observed in our data, we speculated that memory success was associated with differences in eye movements during encoding. For instance, participants with better recognition memory might have been able to memorize the scene images by producing saccades with shorter durations, which may have resulted in lower grid-like codes. While this was not the case, we found consistent trends of higher grid-like codes being associated with lower average saccade frequencies, velocities, and amplitudes (Supplementary information, Figure S8). We encourage future studies to test to what extent grid-like codes are tied to saccade characteristics. Moreover, memory performance might also be influenced by the number of fixations or revisitations. The latter were previously linked to better recognition memory performance and may support thorough encoding by redirecting attention to specific elements of a scene (Kragel et al., 2021). Once more, we could not detect a significant relationship between fixations, revisitations, recognition memory performance, and grid-like codes (Supplementary information, Figure S9A; individuals who made more fixations during encoding showed marginally lower grid-like codes, Supplementary information, Figure S9B). Overall, these findings reflect some inter-individual variability in visual exploration, but these did not significantly contribute to the observed results."

Discussion (page 13, line 406-414):

"Crucially, we found that saccade-based grid-like codes were lower the better participants were able to correctly discriminate old from novel scene images in a subsequent recognition memory task (we did not detect this relationship in the Vienna data set, possibly due to the fact that these participants encoded substantially fewer scene images). The negative brain-behavior relationship was confirmed when focusing exclusively on hits: lower saccade-based entorhinal grid-like codes were tied to better memory performance (hit rate) across individuals and across both data sets. Overall, our findings reinforce the notion that grid-like codes, which are typically discussed in the context of spatial navigation, also play a role in the representation of visual space, and that they are negatively associated with subsequent memory."

Discussion (page 13-14, line 437-450):

"Another possible explanation for the negative brain-behavior result is that participants who were better at recognizing the scene images may have (intentionally or not) used different strategies to complete the task. Rather than relying predominantly on visuospatial encoding, they may have more extensively drawn upon prior knowledge. For instance, when viewing a specific beach scene, they might have recalled a recent beach vacation with highly similar scene features. Such integration of prior knowledge, or

memory “schemas”, could have facilitated scene encoding by guiding eye movements to prominent scene features, thereby enhancing participants’ recognition memory (Bicanski & Burgess, 2019). This is consistent with the idea that schema-congruent information is encoded more efficiently and engages the medial prefrontal cortex more than the medial temporal lobes (Bonasia et al., 2018; Van Kesteren et al., 2013). In contrast, participants who relied exclusively on visuospatial encoding may have shown lower recognition memory performance while displaying increased medial temporal lobe activity and higher grid-like codes for visual space. Unfortunately, we did not qualitatively assess participants’ strategy use during encoding, nor did we account for individual experience or expertise with the specific scene stimuli, which could have helped to clarify the role of grid-like codes in memory formation.”

Reviewer comment: Also, was this analysis not done for the validation study?

We did not perform this analysis for the Vienna data set, since the partial field of view (FoV) during fMRI was designed to optimize signal quality in the entorhinal cortex and surrounding medial temporal lobe regions, but it did not reliably include the frontal eye fields (FEF). To assess FEF coverage, we used *Nilearn’s compute_epi_mask* function to generate a subject-specific field-of-view mask from each participant’s mean normalized functional image. We then overlaid this mask onto the normalized group-average T1-weighted image, together with the binarized FEF cluster, to visually inspect whether the FEF region was included in the acquired functional volume for each subject. This revealed that in 7 out of the 20 participants who were included in the grid-like code analysis, the FEF were not fully covered (see Figure R-4 below). This would have left us with a sample of $N = 13$, which does not allow for a reliable group analysis (Poldrack et al., 2017).

Fig. R-4.: Field-of-view coverage of the frontal eye fields.

The figure shows the estimated field of view (FoV) for all 20 participants in the Vienna data set, overlaid onto the normalized group-average T1-weighted image. The frontal eye field (FEF) cluster is displayed in white, together with each subject’s FoV mask in pink. The figure illustrates that in 7 out of 20 participants (S10, S19, S26, S30, S31, S41, S50), the FEF were not sufficiently covered by the FoV.

Action taken:

We added a brief explanation to the Methods section, clarifying that this analysis was not performed for the Vienna data set.

Methods (page 23-24, line 877-887):

“Whole-brain activation modulated by saccade-based entorhinal grid-like codes

In the Vienna study, the partial field of view (FoV) during fMRI was designed to optimize signal quality in the entorhinal cortex and surrounding medial temporal lobe regions, but did not reliably cover the frontal eye fields. To assess frontal eye field coverage, we used *Nilearn’s compute_epi_mask* function to generate a subject-specific field-of-view mask from each participant’s mean normalized functional image. We overlaid this mask onto the normalized group-average T1-weighted image, together with the binarized frontal eye field cluster, to visually inspect whether the frontal eye fields were included in the acquired functional volume for each subject. This revealed that in 7 out of the 20 participants who were included in the grid-like code analysis, the frontal eye

fields were not fully covered. Since this would have left us with a sample of $N = 13$, which does not allow for a reliable group analysis (Poldrack et al., 2017), we did not perform this analysis for the Vienna data set.”

Reviewer comment: Finally, while we can rationalize increased grid like signals on axis vs off-axis, shouldn't the FEF be equally active for all eye-movement vectors regardless of grid axis? The distribution of grid activity should not matter to the FEF, as long as that grid activity informs target location. It seems unreasonable to suggest FEF and other saccade related activity to code saccades in such a non-uniform way. Why should the FEF be more active for on axis vectors?

We do not agree that one can assume the frontal eye fields (FEF) are equally active for all eye-movement vectors. While the FEF play a central role in the oculomotor system, other regions, such as the parietal eye fields, superior colliculus, primary visual cortex, basal ganglia, and supplementary eye fields, are also critically involved in planning and executing eye movements. It could for example be speculated that the entorhinal grid system actively engages the FEF (and potentially other oculomotor regions) to enforce saccade vectors aligned to the grid axis. Although the FEF have been prominently linked to saccade planning in previous work (Bruce & Goldberg, 1985; Schall, 1991; Vernet et al., 2014), it remains, to the best of our knowledge, unknown whether they are equally engaged or modulated by encoding strategies.

To further examine the relationship between FEF activity and saccades, we extracted the participant-specific, average BOLD activation level from the significant FEF cluster and correlated it with the average number of saccades made during the study period. Results were not significant ($p > .05$) but suggested a tendency toward higher FEF activation being associated with lower saccade numbers across participants (Figure R-5).

Fig. R-5.: Frontal eye field activity vs. saccades.

The scatter plot displays the Pearson correlation between frontal eye field (FEF) BOLD activity extracted from the significant FEF cluster and the mean saccade frequency per trial. Results were not significant ($p > .05$) but pointed toward higher FEF activation at lower saccade numbers across participants.

Action taken:

We performed an additional analysis examining whether FEF activation reflects overall saccade frequency. We did not include this in the manuscript, however, we are happy to do so, if the reviewer insists.

Reviewer 1, comment 7

Some hypotheses in the present article were anticipated by the Buffalo perspective that is cited (L60/61) and these +additional hypotheses were formulated precisely in the Bicanski model, including coupling to the FEF. With the caveat that their model may or may not apply to faces, which are known to benefit from holistic processing. Regardless of this caveat, if prior work - that the authors already cite and hence are aware of - provided inspiration for the present hypotheses, then that should be stated accordingly.

We apologize for being unclear and now explicitly state that our hypotheses are partly informed by the work of Killian and Buffalo (2018), and Bicanski and Burgess (2019). These authors propose that grid-like codes may encode translational vectors between salient stimulus features, thereby constructing a mental map of visual information. Through coordinated activity between medial temporal and visuo-oculomotor regions, grid signals may help to generate memory-guided saccades that support the encoding of visually presented content.

Introduction (page 2, line 59-61):

“Similarly, visual grid cells may provide us with a mental map to organize the spatial relationships of visually presented content. By encoding translational vectors between salient stimulus features, they may guide memory formation (Bicanski & Burgess, 2019; Killian & Buffalo, 2018).”

Rebuttal letter, Graichen et al.

Discussion (page 14, line 471-476):

"We next hypothesized that saccade-based grid-like codes in the entorhinal cortex were tied to neural activity in visuo-oculomotor regions that are known to be involved in saccade generation and coordination, such as the visual cortex or frontal eye fields (Prime et al., 2010). Here, we built on previous work proposing that memory-guided saccades may result from coordinated activity between medial temporal and visuo-oculomotor regions, supported by entorhinal grid-like activity (Bicanski & Burgess, 2019; Killian & Buffalo, 2018)."

Minor:

Very few minor comments as the manuscript is generally well written and mostly clear:

Reviewer 1, comment 8

Why would the average d-prime be so significantly different in a second cohort that performs the same recognition task? It is mentioned later in the text, but the question pops into one's mind immediately when looking at figure 1. Maybe a sentence on this could be included in the main text or caption when this is first discussed.

We added a sentence in the figure caption.

Caption Figure 1C:

"The performance discrepancy between Donders and Vienna studies likely arose from the Vienna study including fewer scenes, no additional task during encoding, and no distractor task between study and test (s. Methods)."

Reviewer 1, comment 9

How the two independent studies differed could be clarified earlier. It described in the Materials and methods, but an extra sentence in the main text will be helpful to avoid interrupting the reading flow.

We updated our manuscript to clarify the differences in study setup between the two studies already earlier in the text.

Introduction (page 3, line 83-85):

"Immediately thereafter, participants completed a recognition memory task and discriminated previously studied ("old") from novel scenes (Figure 1B). The Vienna study also included a delayed test session in the behavioral laboratory after one week."

Reviewer 1, comment 10

"A possible explanation for the negative brain-behavior result is that participants who were better at recognizing the scene images may have used a different strategy to complete the task. Rather than relying only on visuospatial encoding, they may have drawn upon prior knowledge. For instance, when viewing a specific beach scene, they might have recalled a recent beach vacation with highly similar scene features. ..."

This suggestion of memory by proxy is a bit odd. Schemas may have a positive effect, e.g., in an office scene I might look for the computer always on the desk. Hence the schema can inform eye movements and this should apply at study and at test.

We agree with the reviewer that memory schemas can inform eye movements and that this should be the case during study and test periods (Huang et al., 2023). To clarify, rather than suggesting "memory by proxy", our intention was to highlight that some participants may have integrated the visuospatial cues of a given scene with prior knowledge (e.g., the forest scene sparked the memory of a recent hike). This may have been connected to specific gaze behavior, such as schema-guided eye movements (Huang et al., 2023), resulting in lower grid-like codes but better recognition memory performance. In contrast, participants who relied primarily on visuospatial encoding (but did not integrate with prior knowledge) may have produced different eye movement patterns, resulting in higher grid-like codes but lower recognition memory performance.

This account is consistent with work showing that recognition memory for complex scenes relies more on categorical (knowledge-based) representations rather than on purely sensory or visuospatial encoding, such that the absence of prior knowledge may reduce memory performance (Morales-Torres et al., 2024). Moreover, previous work showed that memory encoding engages different brain systems depending on the congruency of novel information with prior knowledge. Incongruent events are typically associated with increased medial temporal lobe activity, whereas congruent events tend to engage the medial prefrontal cortex more strongly (Bonasia et al., 2018; Van Kesteren et al., 2012). Participants who drew on prior knowledge during encoding may have perceived the scene content as more congruent with an existing "schema", leading to reduced visuospatial encoding and medial temporal lobe engagement and, consequently, lower grid-like codes.

Lastly, we would like to emphasize that we are only able to speculate about the role of prior knowledge in saccade-based grid-like codes, as we did not assess whether participants used specific strategies during encoding. We now mention this limitation in our updated text (please find the adapted text passages below).

Action taken:

We updated the discussion to clarify our argument.

Discussion (page 13-14, line 437-450):

“Another possible explanation for the negative brain-behavior result is that participants who were better at recognizing the scene images may have (intentionally or not) used different strategies to complete the task. Rather than relying predominantly on visuospatial encoding, they may have more extensively drawn upon prior knowledge. For instance, when viewing a specific beach scene, they might have recalled a recent beach vacation with highly similar scene features. Such integration of prior knowledge, or memory “schemas”, could have facilitated scene encoding by guiding eye movements to prominent scene features, thereby enhancing participants’ recognition memory (Bicanski & Burgess, 2019). This is consistent with the idea that schema-congruent information is encoded more efficiently and engages the medial prefrontal cortex more than the medial temporal lobes (Bonasia et al., 2018; Van Kesteren et al., 2013). In contrast, participants who relied exclusively on visuospatial encoding may have shown lower recognition memory performance while displaying increased medial temporal lobe activity and higher grid-like codes for visual space. Unfortunately, we did not qualitatively assess participants’ strategy use during encoding, nor did we account for individual experience or expertise with the specific scene stimuli, which could have helped to clarify the role of grid-like codes in memory formation.”

Reviewer #2

The authors present findings from two independent studies regarding grid-like codes in the human entorhinal cortex using fMRI and a scene encoding and recognition memory task. Saccade-related grid-like responses were observed in the left entorhinal cortex during encoding, were time-locked to activation in the left frontal eye field, and were negatively associated with recognition memory (more grid-like codes were associated with lower recognition performance).

Reviewer 2, comment 1

This well-written paper builds nicely on the existing literature from human and non-human animals, and the authors prior work regarding hexadirectional modulation of the fMRI signal, and directly links saccade-based grid-like codes in the entorhinal cortex to the frontal eye fields (a region of cognitive oculomotor control) and to memory performance. The authors are careful to note that these grid-like codes are not merely linked to the number of saccades, and do not occur everywhere in the brain; instead they appear to be specific to the entorhinal cortex. The authors also nicely replicate other, simpler, findings from the literature, specifically, as the relationship between the number of saccades and subsequent memory (although perhaps a reference could be added to note this connection to the broader literature).

Author reply:

We thank the reviewer for the time that they invested in thoroughly evaluating our work and for the constructive comments to improve it.

Following their suggestion, we added several key references that put our findings on the relationship between the number of saccades and subsequent memory in the context of prior work. Specifically, Loftus (1972) found that the more fixations participants made during picture viewing, the better their subsequent recognition memory was. Meister & Buffalo (2016) reported functional and structural connectivity between the medial temporal lobe and oculomotor regions, emphasizing the close interactions between the two systems. Olsen et al. (2016) showed that increased visual sampling in support of subsequent memory depends on hippocampal integrity. Lastly, Fehlmann et al. (2020) demonstrated that a higher frequency of fixations during visual exploration improves memory recall. We integrated these works into our updated manuscript.

Results (page 4-5, line 112-124):

“To provide the basis for our subsequent investigation of the relationship between saccade-based grid-like codes and recognition memory performance, we first assessed the link between eye movements and memory formation. Consistent with previous literature on the role of eye movements in memory encoding (Fehlmann et al., 2020; Loftus, 1972; Meister & Buffalo, 2016; Olsen et al., 2016), the average number of saccades during scenes that were later remembered (7.95 ± 0.40) was significantly higher than during scenes that were later forgotten (6.64 ± 0.43 ; paired-sample t-test, $N = 32$; $t(31) = 8.262$, Cohen’s $d = 1.46$, 95% confidence interval (CI) = [0.99, 1.63], ptwo-tailed < 0.0001). In line with these results, we observed a positive correlation between

individual recognition memory performance (d -prime, 1.82 ± 0.11 ; Figure 1C, left panel) and the number of saccades made when studying later remembered scenes ($r_{\text{Pearson}} = 0.51$, 95% CI = [0.20, 0.73], $p_{\text{two-tailed}} = 0.003$; Figure 1D). Thus, participants who used more saccades to explore later remembered scenes performed better in distinguishing old from novel material during the subsequent recognition memory test."

Reviewer 2, comment 2

One part of the paper that was less convincing is the notion that grid-like codes "contribute to memory formation" or, as noted in the Abstract, "supporting memory formation", when there is a negative relationship between grid codes and subsequent recognition memory (although I appreciate that the authors spend some considerable effort discussing this perhaps puzzling finding). Could one not argue the opposite, that grid-like codes are detrimental to memory formation since higher saccade-based grid-like codes are associated with worse subsequent memory? With respect to possible explanations for the negative relationship, it's not obvious that recall or integration of a schema would necessarily result in a lower grid-cell code, especially if a schema contains information regarding a typical spatial layout of a given scene. Also, if the scenes are fairly typical, it is likely that they automatically invoke the use of schemas. The argument regarding 'efficiency' of encoding is not as convincing as it would have been if the number of saccades had been related to the extent of a grid-like response. I do not have better interpretations to offer the authors regarding the negative relationship, but the careful wording that the codes 'contribute to memory formation' or 'play a role in human memory formation' would seem to hide the nuance of this relationship and perhaps not be a wholly correct interpretation.

We highlight the reviewer's individual comments below once more, followed by our detailed responses:

Reviewer comment: "One part of the paper that was less convincing is the notion that grid-like codes "contribute to memory formation" or, as noted in the Abstract, "supporting memory formation", when there is a negative relationship between grid codes and subsequent recognition memory (although I appreciate that the authors spend some considerable effort discussing this perhaps puzzling finding). Could one not argue the opposite, that grid-like codes are detrimental to memory formation since higher saccade-based grid-like codes are associated with worse subsequent memory?"

We thank the reviewer for raising this important point and agree that the observed negative relationship between entorhinal grid-like codes and subsequent recognition memory performance calls for a more cautious interpretation. We thoroughly revised our manuscript throughout and adjusted the phrasings in the abstract and discussion (please find the text passages below).

To explore alternative explanations for the negative brain-behavior relationship, we conducted several additional analyses:

Differences in saccade characteristics:

First, we reasoned that individuals with better memory performance might have produced saccades with shorter durations, resulting in lower grid-like coding. This does not negate our hypothesis that grid-like codes are relevant to recognition memory performance but highlights that the magnitude of grid-like codes may be associated with specific eye movement characteristics (analogous to path length for grid-like codes during spatial navigation; Wagner et al., 2023). Our original analysis suggested that individuals with better recognition memory performance did not make shorter saccades than low-performing individuals (correlational analysis between average saccade durations and d -prime across participants in the Donders study, $N = 29$, $r_{\text{Pearson}} = -.07$, $p_{\text{two-tailed}} = .72$). We now performed additional work to provide a more detailed analysis of saccade characteristics. We extracted the average saccade frequency (average number of saccades per trial), velocity (average speed of saccades across all trials, in degrees of visual angle per second), and amplitude (average distance of saccades across all trials, in degrees of visual angle) per individual and correlated these measures with the saccade-based grid-like codes obtained from our main analysis (Donders study, bilateral entorhinal cortex, $N = 29$). Results showed consistent trends toward higher grid-like codes at lower average saccade frequency, saccade velocity, and saccade amplitude per trial, suggesting that higher saccade-based grid-like codes might be associated with fewer, slower, and shorter eye movements across participants (Figure R-1).

Differences in patterns of visual exploration:

Since our abovementioned analysis of the basic saccade characteristics (frequency, velocity, amplitude) pointed to consistent trends toward higher saccade-based grid-like codes being associated with fewer and shorter eye movements across participants, we next examined the relationship between visual exploration (fixations, revisitations) and recognition memory performance across participants (analysis was performed on the Donders data set, $N = 32$). Correlations between the average number of fixations and revisitations per

trial and recognition memory performance (d -prime) were not significant (Figure R-2A; we also present our original finding showing a significantly positive association between the average number of saccades per trial and d -prime across participants) but indicated a positive relationship between visual exploration and memory performance. Although theoretically linked to saccade frequency, fixation frequency did not show the same significant relationship with d -prime. This may be due to the 100-ms fixation threshold or blink-related artifacts. We then examined the relationship between visual exploration and the magnitude of saccade-based entorhinal grid-like codes (analysis was performed on the Donders data set, $N = 29$). Correlational analyses were not significant (all $p > 0.05$) but were suggestive of a negative relationship with the average saccade frequency per trial (Figure R-2B), in line with the results pertaining to saccade characteristics above. Overall, these findings reflect inter-individual variability in viewing behavior, which may have contributed to differences in recognition memory performance and entorhinal grid-like coding.

Differences in grid-like coding during hits and misses:

To examine potential differences in grid-like codes during the study period as a function of memory, we quantified grid-like codes separately for subsequently remembered and forgotten scenes (hits vs. misses; see also reviewer 3, comment 1 for a more detailed account). In both studies, participants showed a higher number of hits than misses (mean \pm SEM; Donders study: $N = 32$; 141 ± 6 hits, 70.5 %, 59 ± 6 misses, 29.5 %; Vienna study: $N = 46$; 86 ± 1.5 hits, 89 %, 10 ± 1.5 misses, 11 %), and a higher number of hits in the Vienna compared to the Donders study ($t(49.14) = 5.95$, $p_{two-tailed} < .001$, 95% CI [.126, .255]).

Identical to our main analysis, we focused on the left entorhinal cortex and split the data into equal halves. We then estimated individual grid orientations based on all saccade trajectories in the first data half using a General Linear Model (GLM1, same model structure as in the main analysis). Next, we quantified the magnitude of grid-like codes for hit trials by testing the grid orientation fit on saccade trajectories from hit trials in the second half (GLM2). We then repeated this procedure using only miss trials to assess grid-like coding for forgotten scenes.

Results. Donders study: We found that the magnitude of grid-like codes in the left entorhinal cortex was significant during hit trials ($N = 26$, mean \pm SEM = $.119 \pm .054$, Wilcoxon-test, $V = 237.5$, $p_{one-tailed} = .023$, Cohen's $d = .43$, Wilcoxon's $r = .40$), but not significant during miss trials ($N = 26$, mean \pm SEM = $.050 \pm .126$, Wilcoxon-test, $V = 201$, $p_{one-tailed} = .26$, Cohen's $d = .08$, Wilcoxon's $r = .13$). Magnitudes of grid-like coding were not significantly different between hit vs. miss trials ($N = 26$, paired-sample t -test, $t(25) = .47$, $p_{two-tailed} = .64$, Cohen's $d = .14$, Wilcoxon's $r = .06$).

Results. Vienna study: We found significant grid-like codes in the left entorhinal cortex during hits ($N = 20$, mean \pm SEM = $1.074 \pm .315$, Wilcoxon-test, $V = 181$, $p_{one-tailed} = .0016$, Cohen's $d = .76$, Wilcoxon's $r = .63$), but not during misses ($N = 9$, mean \pm SEM = -3.302 ± 3.377 , Wilcoxon-test, $V = 17$, $p_{one-tailed} = .75$, Cohen's $d = -.23$, Wilcoxon's $r = .22$). Magnitudes of grid-like coding were not significantly different between hit vs. miss trials ($N = 9$, paired-sample t -test, $t(8) = 1.18$, $p_{two-tailed} = .27$, Wilcoxon signed-rank test, $V = 36$, $p = .13$).

Finally, we examined whether the magnitude of grid-like codes during hit trials correlated with recognition memory performance (d -prime or hit rate, which we additionally included in response to reviewer 3). In both studies, we discovered a trend toward higher grid-like codes at lower d -prime, but none of the results were significant (Donders: $N = 26$, $r_{Pearson} = -.12$, 95% CI = $[-.483, .282]$, $p_{two-tailed} = .57$; Vienna: $N = 20$, $r_{Pearson} = -.36$, 95% CI = $[-.690, 0.102]$, $p_{two-tailed} = .12$). Moreover, we observed a significant negative correlation with hit rate in the Donders study ($N = 26$, $r_{Pearson} = -.44$, 95% CI = $[-.704, -.059]$, $p_{two-tailed} = .03$) and a similar non-significant trend in the Vienna study ($N = 20$, $r_{Pearson} = -.44$, 95% CI = $[-.736, 0.009]$, $p_{two-tailed} = .055$). No significant correlations were found between grid-like coding during miss trials and behavioral performance (all $p > .05$). While these results do not explain the underlying cause of the negative grid-behavior relationship, they further corroborate the stability of our findings.

Action taken:

We rephrased our interpretation of the grid-behavior relationship in a more careful and nuanced way:

Abstract (page 2, line 33-36):

"Unexpectedly, saccade-based grid-like codes were associated with recognition memory, such that grid-like codes were lower the better participants performed in subsequently recognizing the scene images. Our findings suggest an entorhinal map of visual space that is timed with neural activity in oculomotor regions, and negatively associated with subsequent memory."

Discussion (page 15, line 504-506):

„In doing so, grid-like signals (which were, unexpectedly, negatively associated with memory) may help coordinate the interplay between medial temporal and visuo-oculomotor regions involved in memory formation.”

Discussion (page 15-16, line 537-540):

“Our findings are the first to show that saccade-based grid-like codes in the entorhinal cortex play a role in human memory formation (which unexpectedly appeared to be negative), highlighting interregional coordination of neural activity that is time-locked to the internal map of visual space.”

We added a paragraph to the discussion where we consider a possible detrimental effect of grid-like codes on memory.

Discussion (page 14, line 463-470):

“Overall, the observed negative brain-behavior relationship challenges the idea that grid-like codes universally support memory formation. Rather than enhancing memory per se, grid-like codes may sometimes interfere with memory, or their contribution may depend on specific conditions. Bicanski and Burgess (2019) argue that grid-like codes mainly contribute to relational memory for familiar content, suggesting they may be less effective in directing gaze when encoding previously unknown scenes. Depending on prior knowledge and task demands, we propose that increased grid-like codes may hinder or support memory formation, which is an interesting aspect warranting future interrogation.”

Reviewer comment: *“With respect to possible explanations for the negative relationship, it’s not obvious that recall or integration of a schema would necessarily result in a lower grid-cell code, especially if a schema contains information regarding a typical spatial layout of a given scene. Also, if the scenes are fairly typical, it is likely that they automatically invoke the use of schemas.”*

Our intention was to highlight that some participants may have integrated the visuospatial cues of a given scene with prior knowledge (e.g., the forest scene sparked the memory of a recent hike). This may have been connected to specific gaze behavior, such as schema-guided eye movements (Huang et al., 2023), resulting in lower grid-like codes but better recognition memory performance. In contrast, participants who relied primarily on visuospatial encoding (but did not integrate with prior knowledge) may have produced different eye movement patterns, resulting in higher grid-like codes but lower recognition memory performance.

This account is consistent with work showing that recognition memory for complex scenes relies more on categorical (knowledge-based) representations rather than on purely sensory or visuospatial encoding, such that the absence of prior knowledge may reduce memory performance (Morales-Torres et al., 2024). Moreover, previous work showed that memory encoding engages different brain systems depending on the congruency of novel information with prior knowledge. Incongruent events are typically associated with increased medial temporal lobe activity, whereas congruent events tend to engage the medial prefrontal cortex more strongly (Bonasia et al., 2018; Van Kesteren et al., 2012). Participants who drew on prior knowledge during encoding may have perceived the scene content as more congruent with an existing “schema”, leading to reduced visuospatial encoding and medial temporal lobe engagement and, consequently, lower grid-like codes.

Lastly, we would like to emphasize that we are only able to speculate about the role of prior knowledge in saccade-based grid-like codes, as we did not assess whether participants used specific strategies during encoding. We now mention this limitation in our updated text (please find the adapted text passages below).

Action taken:

We rephrased the argument accordingly:

Discussion (page 13-14, lines 437-450):

“Another possible explanation for the negative brain-behavior result is that participants who were better at recognizing the scene images may have (intentionally or not) used different strategies to complete the task. Rather than relying predominantly on visuospatial encoding, they may have more extensively drawn upon prior knowledge. For instance, when viewing a specific beach scene, they might have recalled a recent beach vacation with highly similar scene features. Such integration of prior knowledge, or memory “schemas”, could have facilitated scene encoding by guiding eye movements to prominent scene features, thereby enhancing participants’ recognition memory (Bicanski & Burgess, 2019). This is consistent with the idea that schema-congruent information is encoded more efficiently and engages the medial prefrontal cortex more than the medial temporal lobes (Bonasia et al., 2018; Van Kesteren et al., 2013). In contrast, participants who relied exclusively on visuospatial encoding may have shown lower recognition memory performance while displaying increased medial temporal lobe activity and higher grid-like codes for visual space. Unfortunately, we did not qualitatively assess participants’ strategy use during encoding, nor did we account for individual experience or expertise with the specific scene stimuli, which could have helped to clarify the role of grid-like codes in memory formation.”

Reviewer #3

Overall summary:

This manuscript investigates the role of saccade-based grid-like codes in the human entorhinal cortex during memory formation. By analyzing functional MRI (fMRI) and eye-tracking data from two independent studies, the authors report that saccades during scene viewing are associated with grid-like coding patterns in the entorhinal cortex. They further find a negative correlation between the magnitude of these grid-like codes and subsequent recognition memory performance, calculated as a summary statistic for each participant. Additionally, they observe temporal coupling between grid-like codes and activation in the frontal eye fields.

Noteworthy results and significance to the field:

The study presents several noteworthy findings. First, it identifies grid-like coding patterns in the entorhinal cortex associated with saccadic eye movements during scene viewing. Second, it reports a negative correlation between the strength of these grid-like codes and participants' recognition memory performance. Third, it observes temporal coupling between entorhinal grid-like codes and activation in the frontal eye fields, suggesting a functional link between eye movement control and spatial coding.

This paper addresses a critical questions in cognitive and systems neuroscience surrounding the role of grid-like coding in memory formation. It is one of the first studies to explore this relationship in humans, bridging the gap between spatial navigation research and memory processes. The combination of eye tracking with fMRI is technically challenging and requires advanced statistical analyses. The authors have conducted their experiments and analyses with rigor and transparency, potentially providing valuable insights into the neural mechanisms underlying memory formation.

Evaluation of the work:

While the study presents intriguing findings, some major concerns need to be addressed to validate the results and their interpretation.

Author reply:

We thank the reviewer for the time that they invested in thoroughly evaluating our work and for the constructive comments to improve it.

Reviewer 3, comment 1

First, there is an issue with how recognition memory performance was analyzed. The authors computed the d' (d') recognition memory performance metric as a single summary statistic for each participant. However, memory encoding and recognition are dynamic processes that can vary significantly across different scenes due to individual experiences and expertise with the stimuli. Grid-like codes should be analyzed on a per-scene basis rather than using an aggregate measure. Specifically, comparing scenes that were later recognized (hits) versus those that were not (misses) would provide a more precise assessment of how grid-like codes relate to memory encoding strength.

The reviewer raises an important point. We fully agree that encoding and recognition are dynamic processes that can vary across different stimuli (Kragel & Voss, 2022). Unfortunately, we did not collect data on individual experience or expertise with the specific scenes, and now mention this limitation in our updated manuscript (please see text passages below).

Analysis of grid-like codes on a per-scene basis:

We opted for d' as a well-established summary metric of recognition memory performance that allows comparison to prior work (Olsen et al., 2016). Most importantly, our choice was motivated by the specific methodological requirements for the analysis of grid-like codes (which also provides a summary metric). Grid orientations are estimated on one part of the data, and their fit is quantified on an unseen, independent test set. A reliable estimation of grid orientations requires a sufficient number of saccade trajectories that are evenly distributed across the different movement directions. This is typically achieved by partitioning the data into halves, allowing for the estimation of grid orientations and the quantification of grid-like coding using the held-out data part (Stangl et al., 2017). In our case, this yielded 24255/9909 saccades for estimation (Donders/Vienna study). An analysis of grid-like codes on a per-scene basis would require a substantially larger number of saccades per trial to be able to partition them into estimation vs. test sets. Given that we detected, on average, 7.58/4.69 saccades per trial in the Donders/Vienna study, a per-scene analysis would likely result in unstable grid orientation estimates.

Additional analysis of grid-like codes during hit vs. miss trials:

To examine potential differences in grid-like codes during the study period as a function of memory, we performed additional analyses. We quantified grid-like codes separately for subsequently remembered and forgotten scenes (hits vs. misses). In both studies, participants showed a higher number of hits than misses (mean \pm SEM; Donders study: $N = 32$; 141 ± 6 hits, 70.5 %, 59 ± 6 misses, 29.5 %; Vienna study: $N = 46$; 86 ± 1.5 hits, 89 %, 10 ± 1.5 misses, 11 %), and a higher number of hits in the Vienna compared to the Donders study ($t(49.14) = 5.95$, $p_{two-tailed} < .001$, 95% CI [.126, .255]).

Identical to our main analysis, we focused on the left entorhinal cortex and split the data into equal halves (18244/8961 saccades for estimation in Donders/Vienna studies). To ensure reliable estimation of individual grid orientations, the estimation step was based on saccade trajectories from both hit and miss trials in the first data half. This approach was necessary due to the relatively low number of miss trials, particularly in the Vienna study, which precluded reliable estimation from miss trials alone. We then estimated individual grid orientations using a General Linear Model (GLM1, same model structure as in the main analysis). In the second data half, we first tested the grid orientation fit on saccades from hit trials, thereby quantifying the magnitude of grid-like codes during hits (GLM2). We then repeated the same procedure for miss trials. Note that the small number of miss trials across participants may have limited the reliability of this analysis.

Results, Donders study: We found that the magnitude of grid-like codes in the left entorhinal cortex was significant during hit trials ($N = 26$, $p_{one-tailed} = .023$, Cohen's $d = .43$, Wilcoxon's $r = .40$; Table R-1; Figure R-6), but not significant during miss trials ($N = 26$, $p_{one-tailed} = .26$, Cohen's $d = .08$, Wilcoxon's $r = .13$). Magnitudes of grid-like coding were not significantly different between hit vs. miss trials ($N = 26$, paired-sample t -test, $t(25) = .47$, $p_{two-tailed} = .64$, Cohen's $d = .14$, Wilcoxon's $r = .06$).

Results, Vienna study: We found significant grid-like codes in the left entorhinal cortex during hits ($N = 20$, $p_{one-tailed} = .0016$, Cohen's $d = .76$, Wilcoxon's $r = .63$; Table R-1; Figure R-6), but not during misses ($N = 9$, $p_{one-tailed} = .75$, Cohen's $d = -.23$, Wilcoxon's $r = .22$). Magnitudes of grid-like coding were not significantly different between hit vs. miss trials ($N = 9$, paired-sample t -test, $t(8) = 1.18$, $p_{two-tailed} = .27$, Wilcoxon signed-rank test, $V = 36$, $p = .13$).

Finally, we examined whether the magnitude of grid-like codes during hit trials correlated with recognition memory performance (d -prime or hit rate, resonating with our reply to comment 2 by the same reviewer). In both studies, we discovered a trend toward higher grid-like codes (during hit trials) at lower d -prime, but none of the results were significant (Donders: $r = -.12$, $p = .57$; Vienna: $r = -.36$, $p = .12$; Table R-2, Figure R-6). In addition, we observed a significant negative correlation with hit rate in the Donders study ($r = -.44$, $p = .026$; Table R-3, Figure R-6) and a similar non-significant trend in the Vienna study ($r = -.44$, $p = .056$). No significant correlations were found between grid-like coding during miss trials and behavioral performance (Tables R-2 & R-3).

Summary of additional results:

These additional results revealed significant grid-like codes in the left entorhinal cortex during hit trials, but not during miss trials, in both data sets. While magnitudes of grid-like coding did not differ significantly between conditions, the negative association with recognition memory performance observed in the Donders study (higher grid-like codes associated with lower recognition memory performance) persisted when considering hit trials alone. This supports the robustness of the original finding.

The presence of grid-like coding during successful encoding is consistent with the idea that entorhinal spatial representations are engaged in memory formation. However, the lack of a positive association with recognition memory performance across individuals suggests that higher grid-like signals do not necessarily translate to improved memory. This finding may reflect individual variability in how spatial coding contributes to memory encoding, such as differences in encoding strategies across participants.

We thank the reviewer for this excellent suggestion to analyze grid-like codes separately for hit vs. miss trials, which enabled a more detailed investigation of the brain-behavior relationship.

Table 1
Grid-like coding during hits and misses (Wilcoxon tests)

Study	Condition	N	Mean ± SEM	Wilcoxon V	p one-tailed	Cohen's d	Wilcoxon's r
Donders	Hits	26	.119 ± .054	237.5	.023	.43	.40
Donders	Misses	26	.050 ± .126	201	.26	.08	.13
Vienna	Hits	20	1.074 ± .315	181	.0016	.76	.63
Vienna	Misses	9	-3.302 ± 3.377	17	.75	-.23	.22

Note. Grid-like coding is reported as mean ± SEM. Cohen's *d* and Wilcoxon's *r* reflect effect sizes. All *p*-values are one-tailed.

Table 2
Correlations between grid-like coding and *d*-prime

Study	Condition	N	r	95% CI	p two-tailed
Donders	Hits	26	-.12	[-.483, .282]	.57
Donders	Misses	26	-.24	[-.573, .163]	.24
Vienna	Hits	20	-.36	[-.690, 0.102]	.12
Vienna	Misses	9	.41	[-.352, .843]	.28

Note. Pearson's *r* values reflect the correlation between grid-like coding and recognition memory performance (*d*-prime). All *p*-values are two-tailed.

Table 3
Correlations between grid-like coding and hit rate

Study	Condition	N	r	95% CI	p two-tailed
Donders	Hits	26	-.44	[-.704, -.059]	.03
Donders	Misses	26	-.07	[-.444, .328]	.74
Vienna	Hits	20	-.44	[-.736, .009]	.055
Vienna	Misses	9	.12	[-.591, .726]	.76

Note. Pearson's *r* values reflect the correlation between grid-like coding and recognition memory performance (hit rate). All *p*-values are two-tailed.

Fig. R-6.: Saccade-based entorhinal grid-like coding for hits and misses.

A: Donders study (in orange), from left to right: Magnitude of grid-like coding in the left entorhinal cortex (a.u., arbitrary units) during trials that were subsequently remembered (hits) or forgotten (misses). Grid-like codes were significant during hits but not during misses. The scatter plots show Pearson correlations between grid-like codes during hits and recognition memory performance (*d*-prime and hit rate), with a negative but non-significant trend for *d*-prime and a significant negative association for hit rate. B: Vienna study (in pink), from left to right: Magnitude of grid-like codes in the left entorhinal cortex during hits and misses. Grid-like codes were significant during hits but not during misses. The scatter plots show Pearson correlations between grid-like coding during hits and recognition memory performance (*d*-prime and hit rate), with negative but non-significant trends for both measures.

Action taken:

We incorporated these additional results into the manuscript. Please see the relevant text passages below:

Results (page 9-10, line 270-307):

“Saccade-based entorhinal grid-like codes are specific to successful memory encoding

To examine potential differences in grid-like codes during the study period as a function of memory, we quantified grid-like codes separately for subsequently remembered and forgotten scenes (hits vs. misses). We reasoned that if saccade-based entorhinal grid-like codes were involved in successful memory encoding, we should be able to detect them during hits rather than misses. Consistent with our abovementioned behavioral findings regarding d-prime, participants of both studies showed a higher number of hits than misses (mean \pm SEM; Donders study: $N = 32$; 141 ± 6 hits, 70.5 %, 59 ± 6 misses, 29.5 %; Vienna study: $N = 46$; 86 ± 1.5 hits, 89 %, 10 ± 1.5 misses, 11 %), and a higher number of hits in the Vienna compared to the Donders study ($t(49.14) = 5.95$, $ptwo-tailed < .001$, 95% CI [0.126, 0.255]). Identical to our main analysis, we focused on the left entorhinal cortex and split the data into equal halves. To ensure a reliable estimation of individual grid orientations, the estimation step was based on saccade trajectories from all trials (thus, both hits and misses) in the first data half. This was a necessary choice due to the relatively low number of miss trials, particularly in the Vienna study, which would preclude the reliable estimation of grid orientations from miss trials alone. We then estimated individual grid orientations using a General Linear Model (GLM1, same model structure as in the main analysis). Using the second data half, we tested the grid orientations' fit on saccades during hit trials, thereby quantifying the magnitude of grid-like codes during hits (GLM2). The same procedure was repeated for miss trials.

As expected, in the Donders study, grid-like codes in the left entorhinal cortex were significantly increased during hits (one-tailed = .023, Figure 3B), but not during misses (one-tailed = .26; Supplementary information, Table S1; note that grid-like codes were not significantly different between hits vs. misses, $N = 26$, paired-sample t-test, $t(25) = .47$, $ptwo-tailed = .64$, Cohen's $d = .14$). We observed the same results pattern in the Vienna study, revealing that grid-like codes in the left entorhinal cortex were significantly increased during hits (one-tailed = .0016, Figure 3E) but not during misses (one-tailed = .75; Supplementary information, Table S1; again grid-like codes were not significantly different between hits vs. misses, $N = 9$, Wilcoxon signed-rank test, $V = 36$, $p = .13$). Thus, across two independent data sets, we found significantly increased saccade-based grid-like codes in the entorhinal cortex that appeared specific to successful memory encoding.

Saccade-based entorhinal grid-like codes during successful memory encoding are lower at better recognition memory

To examine the brain-behavior relationship more closely, we once more assessed whether the magnitude of grid-like codes during hits was associated with individual variations in recognition memory performance (d-prime). In both studies, we discovered a trend toward higher grid-like codes at lower d-prime, but the results were not significant (all $p > 0.05$; Figure 3C & 3F, left panels, Supplementary information, Table S2). As above, we then also interrogated the correlation between grid-like codes and hit rate. We observed a significantly negative association in the Donders study ($r = -.44$, $p = .026$, Figure 3C, right panel) and a similar (albeit non-significant trend) in the Vienna study ($r = -.44$, $p = .056$; Figure 3F, right panel, Supplementary information, Table S3). No significant correlations were found between grid-like codes during miss trials and behavioral performance (Supplementary Information, Tables S2 & S3).”

Discussion (page 12, line 386-391):

“We hypothesized that saccades during scene viewing were coupled to grid-like codes in the entorhinal cortex and that the grid-related signals would be linked to individual variations in recognition memory performance. In line with this prediction, we found significantly increased entorhinal grid-like codes related to saccades in two independent studies. This effect was specific to subsequently remembered scenes (hits) across both data sets (but note that the small number of miss trials, particularly in the Vienna study, might have limited analysis stability).”

Discussion (page 13, line 406-414):

“Crucially, we found that saccade-based grid-like codes were lower the better participants were able to correctly discriminate old from novel scene images in a subsequent recognition memory task (we did not detect this relationship in the Vienna data set, possibly due to the fact that these participants encoded substantially fewer scene images). The negative brain-behavior relationship was confirmed when focusing exclusively on hits: lower saccade-based entorhinal grid-like codes were tied to better memory performance (hit rate) across individuals and across both data sets. Overall, our findings reinforce the notion that grid-like codes, which are typically discussed in the context of spatial navigation, also play a role in the representation of visual space, and that they are negatively associated with subsequent memory.”

Discussion (page 14, line 448-450):

“Unfortunately, we did not qualitatively assess participants' strategy use during encoding, nor did we account for individual experience or expertise with the specific scene stimuli, which could have helped to clarify the role of grid-like codes in memory formation.”

Reviewer 3, comment 2

Second, regarding the categorization of trial outcomes, the task design yields four categories:

1. Hits: Scenes shown in the study phase and correctly identified as old in the test phase.
2. Misses: Scenes shown in the study phase but incorrectly judged as new in the test phase.
3. Correct Rejections: Scenes not shown in the study phase and correctly identified as new in the test phase.
4. False Alarms: Scenes not shown in the study phase but incorrectly judged as old in the test phase.

Since the BOLD signal analyses focus on the study (encoding) phase, incorporating false alarms—which pertain to stimuli not presented during encoding—into the recognition memory metric may confound the results. Recognition memory performance should be computed using only hits and misses, focusing on scenes that were actually encoded during the study phase. This approach aligns the behavioral data with the neural data analyzed from the study period.

We agree with the reviewer that assessing recognition memory performance using hits and misses only, based on those scenes participants viewed during memory encoding, may more directly capture the relationship between grid-like coding during memory formation and subsequent recognition memory performance.

Rationale for d -prime as recognition memory performance metric:

To explain our rationale, we chose to use d -prime as our primary measure of recognition memory performance as a well-established and widely used metric (Olsen et al., 2016). D -prime takes both false alarms and correct rejections into account [$d = z(\text{hit rate}) - z(\text{false alarm rate})$], allowing it to distinguish true recognition ability from potential response biases, such as a general tendency to respond “old” or “new” (Hautus et al., 2021; Swets, 1961). Although false alarms and correct rejections are linked to scenes that were not shown during encoding, these measures are not independent of the encoding process. In other words, the more thoroughly participants encoded the scenes, the better they were able to distinguish between old and new scenes later on. Thus, accurate recognition memory not only involves identifying previously encountered scenes as familiar but also relies on avoiding false recognition of novel scenes. Therefore, we think that d -prime offers a more sensitive measure of recognition memory performance than hit or miss rates, capturing recognition memory performance independent of response biases or guessing behavior.

Novel analysis based on hit rate:

Having said that, we agree with the reviewer’s comment regarding the alignment between behavioral and neural data and added new analyses to clarify this point. Specifically, we now report the correlation between d -prime and hit rate, as well as the correlations between saccade-based entorhinal grid-like codes and hit rate (Figure R-7; note that the grid-like coding results are based on the initial, main analysis that collapsed across all trials). The results illustrate the close relationship between the two behavioral metrics, yielding virtually identical results. The correlation between d -prime and hit rate was stronger in the Vienna study ($N = 20$, $r_{\text{Pearson}} = .88$, 95% CI = [.718, .952], $p_{\text{two-tailed}} < .001$) than in the Donders study ($N = 29$, $r_{\text{Pearson}} = .57$, 95% CI = [.260, .776], $p_{\text{two-tailed}} = .001$), potentially due to the higher recognition memory performance in the Vienna study (d -prime, mean \pm SEM: 3.24 ± 0.13 vs. 1.82 ± 0.11). In the Donders study, the correlation between grid-like coding and hit rate mirrored the correlation between grid-like coding and d -prime (bilateral entorhinal cortex, grid-like coding versus d -prime: $N = 29$, $r_{\text{Pearson}} = -.51$, 95% CI = [-.741, -.182], $p_{\text{two-tailed}} = .004$); grid-like coding versus hit-rate: $N = 29$, $r_{\text{Pearson}} = -.49$, 95% CI = [-.726, -.150], $p_{\text{two-tailed}} = .007$. The same was the case in the Vienna study (left entorhinal cortex, grid-like coding versus d -prime: $N = 20$, $r_{\text{Pearson}} = .01$, 95% CI = [-.437, .448], $p_{\text{two-tailed}} = .98$; grid-like coding versus hit rate: $N = 20$, $r_{\text{Pearson}} = .001$, 95% CI = [-.442, .443], $p_{\text{two-tailed}} = .998$).

Fig. R-7.: Comparison between *d*-prime and hit rate.

A: Donders study, the scatter plots show Pearson correlations between *d*-prime and hit rate (left panel, $N = 29$, $r = .57$, $p_{two-tailed} = .001$), grid-like coding (bilateral entorhinal cortex, a.u., arbitrary units) and *d*-prime (middle panel, $r = -.51$, $p_{two-tailed} = .004$), and grid-like coding and hit rate (right panel, $r = -.49$, $p_{two-tailed} = .007$). B: Vienna study, the scatter plots show Pearson correlations between hit rate and *d*-prime (left panel, $N = 20$, $r = .88$, $p_{two-tailed} < .001$), grid-like coding (left entorhinal cortex) and *d*-prime (middle panel, $r = .01$, $p_{two-tailed} = .98$), and grid-like coding and hit rate (right panel, $r = .001$, $p_{two-tailed} = .998$). The results illustrate the close relationship between the two behavioral metrics, yielding virtually identical results. The correlation between *d*-prime and hit rate was stronger in the Vienna study than in the Donders study.

Action taken:

Overall, the close correspondence between the correlations using *d*-prime vs. hit rate suggests that our findings do not depend on the specific memory measure used. Given the consistency of these results, and given our reasoning above, we would prefer to retain *d*-prime as our primary measure of recognition memory performance. However, following the reviewer’s suggestion, we now include these additional results in the manuscript to complement our approach (**Supplementary Fig. S10**).

Results (page 9, line 262-269):

“While *d*-prime is a well-established and sensitive measure of recognition memory performance (Olsen et al., 2016), it includes false alarms to novel stimuli that were not part of the study period. To better align the behavioral metric with our neural data (that specifically reflected scene encoding), we also correlated saccade-based grid-like codes with hit rate across participants (which is restricted to only those scenes that they actually viewed during the study periods). Results were virtually identical (Donders: $N = 29$, $r_{Pearson} = -0.49$, 95% CI = [-0.726, -0.150], $p_{two-tailed} = .0007$; Vienna: $N = 20$, $r_{Pearson} = .001$, 95% CI = [-.442, .443], $p_{two-tailed} = .998$; Supplementary information, Figure S10), highlighting the consistency between the two behavioral measures.”

Reviewer 3, comment 3

Third, there is a need for clarification in the interpretation of the negative correlation between grid-like codes and memory performance. The finding suggests that stronger grid-like coding is associated with poorer memory recognition, which is counterintuitive and requires careful interpretation. An analysis comparing grid-like codes between strong encoding trials (hits) and weak encoding trials (misses) on a scene-by-scene basis would clarify this relationship. If grid codes are indeed stronger for scenes that are later forgotten, this would support the authors’ interpretation and provide a more nuanced understanding of the data.

We thank the reviewer for this excellent point. While we agree that comparing grid-like codes scene-by-scene between hits and misses may help clarify the grid-behavior relationship, analyzing the magnitude of

grid-like codes on a per-scene basis would require a substantially larger number of saccades per trial to allow for stable grid orientation estimates. We thus opted for a different approach.

Additional analysis of grid-like codes during hit vs. miss trials:

As outlined above, we re-analyzed grid-like coding separately for subsequently remembered (hits) and forgotten scenes (misses). In both studies, participants showed a higher number of hits than misses (mean \pm SEM; Donders study: $N = 32$; 141 ± 6 hits, 70.5 %, 59 ± 6 misses, 29.5 %; Vienna study: $N = 46$; 86 ± 1.5 hits, 89 %, 10 ± 1.5 misses, 11 %), and a higher number of hits in the Vienna compared to the Donders study ($t(49.14) = 5.95$, $p_{two-tailed} < .001$, 95% CI [0.126, 0.255]).

Identical to our main analysis, we focused on the left entorhinal cortex and split the data into equal halves. We then estimated individual grid orientations based on all saccade trajectories in the first data half using a General Linear Model (GLM1, same model structure as in the main analysis). Next, we quantified the magnitude of grid-like codes for hit trials by testing the grid orientation fit on saccade trajectories from hit trials in the second half (GLM2). We then repeated this procedure using only miss trials to assess grid-like coding for forgotten scenes.

Results. Donders study: We found that the magnitude of grid-like codes in the left entorhinal cortex was significant during hit trials ($N = 26$, mean \pm SEM = $.119 \pm .054$, Wilcoxon-test, $V = 237.5$, $p_{one-tailed} = .023$, Cohen's $d = .43$, Wilcoxon's $r = .40$), but not significant during miss trials ($N = 26$, mean \pm SEM = $.050 \pm .126$, Wilcoxon-test, $V = 201$, $p_{one-tailed} = .26$, Cohen's $d = .08$, Wilcoxon's $r = .13$). Magnitudes of grid-like coding were not significantly different between hit vs. miss trials ($N = 26$, paired-sample t -test, $t(25) = .47$, $p_{two-tailed} = .64$, Cohen's $d = .14$, Wilcoxon's $r = .06$).

Results. Vienna study: We found significant grid-like codes in the left entorhinal cortex during hits ($N = 20$, mean \pm SEM = $1.074 \pm .315$, Wilcoxon-test, $V = 181$, $p_{one-tailed} = .0016$, Cohen's $d = .76$, Wilcoxon's $r = .63$), but not during misses ($N = 9$, mean \pm SEM = -3.302 ± 3.377 , Wilcoxon-test, $V = 17$, $p_{one-tailed} = .75$, Cohen's $d = -.23$, Wilcoxon's $r = .22$). Magnitudes of grid-like coding were not significantly different between hit vs. miss trials ($N = 9$, paired-sample t -test, $t(8) = 1.18$, $p_{two-tailed} = .27$, Wilcoxon signed-rank test, $V = 36$, $p = .13$).

Finally, we examined whether the magnitude of grid-like codes during hit trials correlated with recognition memory performance (d' -prime or hit rate, resonating with our reply to comment 2 by the same reviewer). In both studies, we discovered a trend toward higher grid-like codes at lower d' -prime, but none of the results were significant (Donders: $r = -.12$, $p = .57$; Vienna: $r = -.36$, $p = .12$; Table R-2, Figure R-6). In addition, we observed a significant negative correlation with hit rate in the Donders study ($r = -.44$, $p = .026$; Table R-3, Figure R-6) and a similar non-significant trend in the Vienna study ($r = -.44$, $p = .056$). No significant correlations were found between grid-like coding during miss trials and behavioral performance (Tables R-2 & R-3).

Interpretation of the brain-behavior relationship:

Altogether, these results provide more detailed insight into the relationship between grid-like coding and memory formation. While grid-like codes are engaged during successful memory encoding (hit trials), their magnitude appears to be negatively associated with memory success.

Action taken:

We re-analyzed grid-like coding for hit and miss trials separately and integrated the new results into the manuscript:

Discussion (page 12, line 386-391):

"We hypothesized that saccades during scene viewing were coupled to grid-like codes in the entorhinal cortex and that the grid-related signals would be linked to individual variations in recognition memory performance. In line with this prediction, we found significantly increased entorhinal grid-like codes related to saccades in two independent studies. This effect was specific to subsequently remembered scenes (hits) across both data sets (but note that the small number of miss trials, particularly in the Vienna study, might have limited analysis stability)."

Discussion (page 13, line 406-414):

"Crucially, we found that saccade-based grid-like codes were lower the better participants were able to correctly discriminate old from novel scene images in a subsequent recognition memory task (we did not detect this relationship in the Vienna data set, possibly due to the fact that these participants encoded substantially fewer scene images). The negative brain-behavior relationship was confirmed when focusing exclusively on hits: lower saccade-based entorhinal grid-like codes were tied to better memory performance (hit rate) across individuals and across both data sets. Overall, our findings reinforce the notion that grid-like codes, which are typically discussed in the context of spatial navigation, also play a role in the representation of visual space, and that they are negatively associated with subsequent memory."

Reviewer 3, comment 4

Fourth, the validation of the eye-tracking data is essential. Accurate eye-tracking data are crucial for linking saccadic movements to neural activity. Eye tracking within an fMRI environment poses additional challenges due to equipment constraints and participant movement. The manuscript should include visualizations of basic eye movement statistics, such as saccade distributions, fixation durations, and calibration accuracy. This information would validate the quality of the eye-tracking data and assure readers of the reliability of the findings.

To address this, we now provide novel analysis that highlights the quality of our eye tracking data. This includes distributions of saccade directions, saccade amplitude-velocity plots, fixation frequencies and durations (Figure R-8), as well as additional information on calibration accuracy (Figure R-9).

Distribution of saccade directions:

To examine potential biases in saccade directions, we visualized the angular distribution (in degrees) of all saccades that occurred during the study period. The distribution depicts the number of saccades in each direction (0° = rightward, 90° = upward, 180° = leftward, 270° = downward). In the Donders study, saccades were primarily oriented along the horizontal axis. In the Vienna study, a similar pattern is seen, albeit less pronounced, with more evenly distributed saccades across all cardinal directions. This slight deviation likely reflects differences in the stimulus presentation format. The rectangular scene images presented in the Donders study (640×480) may have encouraged more horizontal saccades compared to the square scene images in the Vienna study (500×500). Overall, participants' gaze behavior aligns with typical patterns observed during free viewing of visual scene stimuli (Foulsham et al., 2008).

Saccade amplitudes against velocities:

Next, we plotted saccade amplitudes against saccade velocities. In healthy individuals, these measures are typically positively correlated, reflecting the principle that larger saccades tend to be faster (Bahill et al., 1975; Gibaldi & Sabatini, 2021). Consistent with this, we observed a positive correlation across both studies, indicating reliable saccade measurements despite the challenges imposed by the fMRI environment (Donders study: $N = 32$, $r_{Pearson} = 0.891$, 95% CI = [0.787, 0.946], $p_{two-tailed} < .001$; Vienna study: $N = 46$, $r_{Pearson} = .532$, 95% CI = [.286, .712], $p_{two-tailed} < .001$).

Average fixation frequency and duration:

The main focus of our study was to investigate whether participants exhibited grid-like coding based on saccades that occurred during the study period and if this would subserve memory formation. Accordingly, our primary assessment of eye data quality focused on saccade measures. To also validate the quality of eye fixations, we returned to the identification of fixations as performed by the EyeLink system, where fixations were defined as periods between saccades and blinks. Fixations shorter than 100 ms were excluded to ensure that only meaningful fixation events were considered (Hannula, 2010). The average number of fixations per trial across subjects (Donders study: $N = 32$, mean \pm SEM, $4.50 \pm .51$; Vienna study: $N = 46$, $6.00 \pm .26$) and the average fixation durations per subject (Donders study: $N = 32$, mean \pm SEM, 184.99 ± 4.47 ms; Vienna study: $N = 46$, 210.78 ± 1.75 ms) were within the expected range for free viewing of visual scene stimuli (e.g., Nuthmann, 2017), further supporting the validity of our eye tracking data. Since fixations were identified using EyeLink's built-in parser, whereas saccades were detected separately using an independent method (s. Methods), the two measures may not correspond exactly.

Fig. R-8.: Quality assessment of eye movement data during scene encoding.

A: Polar histogram showing the distribution of saccade directions across all trials, reflecting typical patterns of visual exploration during scene encoding. B: Scatter plots show Pearson correlations between saccade amplitude (dva) and velocity (dva/ms) across participants. C: Mean fixations per trial across participants D: Mean fixation durations (ms) across participants. Data points show values per subject, boxplots show the median (upper and lower borders mark the interquartile range, whiskers show minimum and maximum non-outlier values), orange = Donders study, pink = Vienna study.

Calibration accuracy:

Finally, we examined calibration accuracy as an additional indicator of data quality. Each recording session began with a calibration-validation procedure according to the standard EyeLink protocol. To map raw eye movement data onto screen coordinates, participants first sequentially fixated on nine targets arranged in a 3 x 3 grid (calibration). In a second step, participants were asked to fixate the same targets once more until the differences between the current and the previously obtained gaze positions were < 1° of visual angle (validation). If these criteria were met, the eye tracker recording was started.

We now include a visualization of the mean calibration error (in degrees of visual angle, dva) for the Vienna study (Figure R-9, left panel). The average calibration error across all participants ($N = 46$, mean \pm SEM, $.596 \pm .028$ dva) confirmed that calibration accuracy remained below the 1° threshold. The two participants who exceeded this threshold were excluded from the grid-like code analysis and, therefore, did not influence the reported results (Figure R-9, right panel). Unfortunately, the calibration data for the Donders study is not available. As our analyses primarily relied on gaze direction rather than absolute gaze position, calibration accuracy was less critical for our main findings.

Fig. R-9.: Calibration accuracy.

Left panel: Distribution of mean calibration error (dva) across all participants in the Vienna study ($N = 46$: mean \pm SEM, 0.596 ± 0.028 dva), right panel: Distribution of mean calibration error (dva) for the subset of participants included in the grid-like coding analysis ($N = 20$: mean \pm SEM, 0.544 ± 0.034 dva). Values remained below the 1° threshold, confirming sufficient calibration accuracy.

Action taken:

We appreciate the reviewer's suggestion to include this information, as it helps to further demonstrate the quality of our eye tracking data. We included the additional analysis in the manuscript (**Supplementary Fig. S6 & S7**); please find the text passages below.

Supplement (page 3-4, line 87-133):

"Assessment of eye tracking data quality in the Donders and the Vienna study

To validate the quality of our eye tracking data, we assessed the distribution of saccade directions, the relationship between saccade amplitudes and velocities, fixation frequencies and durations, as well as calibration accuracy across subjects.

Distribution of saccade directions:

To examine potential biases in saccade directions, we visualized the angular distribution (in degrees) of all saccades that occurred during the study period (Supplementary Fig. S6). The distribution depicts the number of saccades in each direction (0° = rightward, 90° = upward, 180° = leftward, 270° = downward). In the Donders study, saccades were primarily oriented along the horizontal axis. In the Vienna study, a similar pattern is seen, albeit less pronounced, with more evenly distributed saccades across all cardinal directions. This slight deviation likely reflects differences in the stimulus presentation format. The rectangular scene images presented in the Donders study (640×480) may have encouraged more horizontal saccades compared to the square scene images in the Vienna study (500×500). Overall, participants' gaze behavior aligns with typical patterns observed during free viewing of visual scene stimuli (Foulsham et al., 2008).

Saccade amplitudes against velocities:

Next, we plotted saccade amplitudes against saccade velocities (Supplementary Fig. S6). In healthy individuals, these measures are typically positively correlated, reflecting the principle that larger saccades tend to be faster (Bahill et al., 1975; Gibaldi & Sabatini, 2021). Consistent with this, we observed a positive correlation across both studies, indicating that reliable saccade measurements despite the challenges imposed by the fMRI environment (Donders study: $N = 32$, $r_{\text{Pearson}} = 0.891$, 95% CI = [0.787, 0.946], ptwo-tailed < .001; Vienna study: $N = 46$, $r_{\text{Pearson}} = .532$, 95% CI = [.286, .712], ptwo-tailed < .001).

Average fixation frequency and duration:

The main focus of our study was to investigate whether participants exhibited grid-like coding based on saccades that occurred during the study period and if this would subserve memory formation. Accordingly, our primary assessment of eye data quality focused on saccade measures. To also validate the quality of eye fixations, we returned to the identification of fixations as performed by the EyeLink system, where fixations were defined as periods between saccades and blinks. Fixations shorter than 100 ms were excluded to ensure that only meaningful fixation events were considered (Hannula, 2010). The average number of fixations per trial across subjects (Donders study: $N = 32$, mean \pm SEM, $4.50 \pm .51$; Vienna study: $N = 46$, $6.00 \pm .26$) and the average fixation durations per subject (Donders study: $N = 32$, mean \pm SEM, 184.99 ± 4.47 ms; Vienna study: $N = 46$, 210.78 ± 1.75 ms) were within the expected range for free viewing of visual scene stimuli (e.g., Nuthmann, 2017), further supporting the validity of our eye tracking data (Supplementary Fig. S6).

Calibration accuracy:

Finally, we examined calibration accuracy as an additional indicator of data quality. Each recording session began with a calibration-validation procedure according to the standard EyeLink protocol. To map raw eye movement data onto screen coordinates, participants first sequentially fixated on nine targets arranged in a 3×3 grid (calibration). In a second step, participants were asked to fixate the same targets once more until the differences between the current and the previously obtained gaze positions were $< 1^\circ$ of visual angle (validation). If these criteria were met, the eye tracker recording was started.

The average calibration error across all participants in the Vienna study ($N = 46$, mean \pm SEM, 0.596 ± 0.028 degrees of visual angle, dva, Supplementary Fig. S7) confirmed that calibration accuracy remained below the 1° threshold. The two participants who exceeded this threshold were excluded from the grid-like code analysis and, therefore, did not influence the reported results (Supplementary Fig. S7, right panel). As our analyses primarily relied on gaze direction rather than absolute gaze position, calibration accuracy was less critical for our main findings."

Reviewer 3, comment 5

Methodological soundness:

The authors have employed advanced fMRI analyses and integrated eye-tracking data, demonstrating technical expertise. They have been transparent with methodological details, but additional information on data preprocessing steps, especially for eye-tracking data, would enhance reproducibility. The statistical methods appear appropriate, but re-analysis based on the recommendations above is necessary to strengthen the conclusions.

To ensure clarity and transparency, we carefully documented all steps involved in the preprocessing of the eye tracking data and have now further expanded the relevant methods section. Please find the relevant text parts below.

Donders study:

Methods (page 17, line 595-619):

"To capture saccadic eye movements, we recorded horizontal and vertical eye gaze and pupil size, using a video-based infrared eye tracker (EyeLink 1000 Plus, SR Research, Ontario, Canada). Eye tracking was performed at a viewing distance of 86.6 cm. The screen had a resolution of 1280 x 960 pixels and measured 36.9 cm in width and 27.7 cm in height.

Before recording, raw eye movement data was mapped onto screen coordinates by means of a calibration procedure. Participants sequentially fixated on nine fixation points on the screen, arranged in a 3 x 3 grid. This was followed by a validation procedure during which the nine fixation points were presented once more while the differences between the current and previously obtained gaze fixations (from the calibration period) were measured. The calibration settings were accepted if these differences were $< 1^\circ$ of visual angle, and the eye tracker recording was started. A detailed assessment of eye tracking data quality, including saccade and fixation metrics is provided in the Supplement (Supplementary information, Figures S6 & S7).

Eye tracking data was processed using Fieldtrip (<https://www.fieldtriptoolbox.org>) through a two-step procedure consisting of automatic saccade detection followed by visual inspection of the data. Saccadic eye movements were identified by transforming vertical and horizontal eye movements into velocities, whereby velocities exceeding a threshold of 6 x the standard deviation (SD) of the velocity distribution and with a duration of > 12 ms were defined as saccades (Engbert & Kliegl, 2003). Saccade onsets during trials of the study period (i.e., during the presentation of scene images) were defined as events-of-interest. Only saccades that followed a minimum fixation period of 25 ms were included. Saccades that were followed or preceded by blinks (± 100 ms) were excluded (blinks were defined as large deflections in pupil diameter: mean ± 5 standard deviations; eye tracking data in the vicinity of blinks is unreliable due to saturation effects). To identify noise or blinks that could have been misattributed as saccades, trials with more than 25% of missing eye tracker data were discarded. We detected a total of 48510 saccades in the eye tracking data ($N = 32$; average number of saccades per participant, mean \pm SEM: 1515.94 \pm 80.95 saccades)."

Vienna study:

Methods (page 22, line 795-803):

"Saccades were tracked by recording horizontal and vertical eye gaze and pupil size with a video-based infrared eye tracker (EyeLink 1000 Plus, SR Research, Ontario, Canada), performed at a viewing distance of 102 cm. The monitor measured 69.84 cm in width and 39.88 cm in height. Screen resolution varied across participants. A resolution of 1920 x 1080 pixels was used for participants 1–3, 14–15, 17–22, and 36–37, while a resolution of 1280 x 1024 pixels was used for participants 4–13, 16, 23, 25–28, 30–35, 38–47, 49, and 50. To map raw eye movement data onto screen coordinates, we implemented a calibration-validation procedure (as described for the Donders study). A detailed assessment of eye tracking data quality is provided in the Supplement (Supplementary information, Figures S6 & S7). Eye tracking data were processed using Fieldtrip (<https://www.fieldtriptoolbox.org>)."

Reviewer 3, comment 6

Conclusions and claims:

The current analysis partially supports the conclusions, but the methodological concerns raised need to be addressed. Re-analyzing the data on a per-scene basis and refining the recognition memory metric are essential steps to confirm the findings.

Summary of recommendations:

I recommend that the authors re-analyze the recognition memory data by computing performance using hits and misses only and perform scene-by-scene analyses to correlate grid-like codes with subsequent memory outcomes. Enhancing the presentation of eye-tracking data by including figures or tables showing saccade amplitudes, directions, and fixation patterns, as well as providing details on eye-tracking calibration procedures and error rates, would greatly improve the manuscript. Clarifying the interpretation of findings, discussing potential reasons for the negative correlation between grid-like codes and memory performance, and exploring alternative explanations or confounding factors that may influence the results are also important.

The manuscript addresses an important topic with the potential to make a significant contribution to the field. However, the concerns regarding the analysis and interpretation of the data need to be addressed. By implementing the recommended revisions, the authors can strengthen their findings and provide clearer insights into the role of grid-like coding in memory formation.

In response to the reviewer's suggestions, we performed the following analysis:

- We re-analyzed recognition memory performance based on scenes participants viewed during encoding (hits and misses) and tested the association with saccade-based entorhinal grid-like coding. This yielded virtually identical results to those obtained in our initial analysis.

Rebuttal letter, Graichen et al.

- To dissociate grid-like coding during successful versus less successful memory encoding, we repeated the grid-like code analysis separately for hits and misses. This revealed significant grid-like codes during hits, but not during misses. The negative brain-behavior relationship persisted (also when considering the hit rate).
- We now present the eye tracking data in full, showing saccade amplitudes, directions, fixation patterns, and information on the calibration procedures.
- Finally, we clarified our interpretation of the negative brain-behavior relationship and addressed multiple methodological aspects to form a concisely-written and transparent manuscript.

We would like to thank all reviewers once more for the thoughtful and constructive feedback, which has helped to substantially improve the manuscript!

References

- Bahill, A. T., Clark, M. R., & Stark, L. (1975). The main sequence, a tool for studying human eye movements. *Mathematical Biosciences*, *24*(3), 191–204. [https://doi.org/10.1016/0025-5564\(75\)90075-9](https://doi.org/10.1016/0025-5564(75)90075-9)
- Bicanski, A., & Burgess, N. (2019). A Computational Model of Visual Recognition Memory via Grid Cells. *Current Biology*, *29*(6), 979–990.e4. <https://doi.org/10.1016/J.CUB.2019.01.077>
- Bonasia, K., Sekeres, M. J., Gilboa, A., Grady, C. L., Winocur, G., & Moscovitch, M. (2018). Prior knowledge modulates the neural substrates of encoding and retrieving naturalistic events at short and long delays. *Neurobiology of Learning and Memory*, *153*, 26–39. <https://doi.org/10.1016/j.nlm.2018.02.017>
- Bruce, C. J., & Goldberg, M. E. (1985). Primate frontal eye fields. I. Single neurons discharging before saccades. *Journal of Neurophysiology*, *53*(3), 603–635. <https://doi.org/10.1152/jn.1985.53.3.603>
- Fehlmann, B., Coynel, D., Schickel, N., Milnik, A., Gschwind, L., Hofmann, P., Papassotiropoulos, A., & De Quervain, D. J.-F. (2020). Visual Exploration at Higher Fixation Frequency Increases Subsequent Memory Recall. *Cerebral Cortex Communications*, *1*(1), tgaa032. <https://doi.org/10.1093/texcom/tgaa032>
- Foulsham, T., Kingstone, A., & Underwood, G. (2008). Turning the world around: Patterns in saccade direction vary with picture orientation. *Vision Research*, *48*(17), 1777–1790. <https://doi.org/10.1016/j.visres.2008.05.018>
- Gibaldi, A., & Sabatini, S. P. (2021). The saccade main sequence revised: A fast and repeatable tool for oculomotor analysis. *Behavior Research Methods*, *53*(1), 167–187. <https://doi.org/10.3758/s13428-020-01388-2>
- Hannula, D. E. (2010). Worth a glance: Using eye movements to investigate the cognitive neuroscience of memory. *Frontiers in Human Neuroscience*, *4*. <https://doi.org/10.3389/fnhum.2010.00166>
- Hautus, M. J., Macmillan, N. A., & Creelman, C. D. (2021). *Detection Theory: A User's Guide* (3. Aufl.). Routledge. <https://doi.org/10.4324/9781003203636>
- Huang, J., Velarde, I., Ma, W. J., & Baldassano, C. (2023). Schema-based predictive eye movements support sequential memory encoding. *eLife*, *12*, e82599. <https://doi.org/10.7554/eLife.82599>
- Johansson, R., Nyström, M., Dewhurst, R., & Johansson, M. (2022). Eye-movement replay supports episodic remembering. *Proceedings of the Royal Society B*, *289*(1977), 20220964.
- Killian, N. J., & Buffalo, E. A. (2018). Grid cells map the visual world. *Nature Neuroscience*, *21*(2), 161–162. <https://doi.org/10.1038/s41593-017-0062-4>
- Kragel, J. E., Schuele, S., VanHaerents, S., Rosenow, J. M., & Voss, J. L. (2021). Rapid coordination of effective learning by the human hippocampus. *SCIENCE ADVANCES*.
- Kragel, J. E., & Voss, J. L. (2022). Looking for the neural basis of memory. *Trends in Cognitive Sciences*, *26*(1), 53–65. <https://doi.org/10.1016/j.tics.2021.10.010>
- Liu, J., Zhang, H., Yu, T., Ni, D., Ren, L., Yang, Q., Lu, B., Wang, D., Heinen, R., Axmacher, N., & Xue, G. (2020). Stable maintenance of multiple representational formats in human visual short-term memory. *Proceedings of the National Academy of Sciences*, *117*(51), 32329–32339. <https://doi.org/10.1073/pnas.2006752117>
- Loftus, G. R. (1972). Eye fixations and recognition memory for pictures. *Cognitive psychology*, *3*(4), 525–551.
- Maguire, E. A., Frith, C. D., & Morris, R. G. M. (1999). The functional neuroanatomy of comprehension and memory: The importance of prior knowledge. *Brain*, *122*(10), 1839–1850. <https://doi.org/10.1093/brain/122.10.1839>
- Morales-Torres, R., Wing, E. A., Deng, L., Davis, S. W., & Cabeza, R. (2024). Visual Recognition Memory of Scenes Is Driven by Categorical, Not Sensory, Visual Representations. *The Journal of Neuroscience*, *44*(21), e1479232024. <https://doi.org/10.1523/JNEUROSCI.1479-23.2024>
- Neubauer, A. C., & Fink, A. (2009). Intelligence and neural efficiency. *Neuroscience & Biobehavioral Reviews*, *33*(7), 1004–1023.
- Nuthmann, A. (2017). Fixation durations in scene viewing: Modeling the effects of local image features, oculomotor parameters, and task. *Psychonomic Bulletin & Review*, *24*(2), 370–392. <https://doi.org/10.3758/s13423-016-1124-4>
- Olsen, R. K., Sebanayagam, V., Lee, Y., Moscovitch, M., Grady, C. L., Rosenbaum, R. S., & Ryan, J. D. (2016). The relationship between eye movements and subsequent recognition: Evidence from individual differences and amnesia. *Cortex*, *85*, 182–193. <https://doi.org/10.1016/j.cortex.2016.10.007>
- Picard, N., Matsuzaka, Y., & Strick, P. L. (2013). Extended practice of a motor skill is associated with reduced metabolic activity in M1. *Nature Neuroscience*, *16*(9), 1340–1347. <https://doi.org/10.1038/nn.3477>
- Poldrack, R. A., Baker, C. I., Durnez, J., Gorgolewski, K. J., Matthews, P. M., Munafò, M. R., Nichols, T. E., Poline, J.-B., Vul, E., & Yarkoni, T. (2017). Scanning the horizon: Towards transparent and reproducible neuroimaging research. *Nature Reviews Neuroscience*, *18*(2), 115–126. <https://doi.org/10.1038/nrn.2016.167>
- Schall, J. D. (1991). Neuronal activity related to visually guided saccades in the frontal eye fields of rhesus monkeys: Comparison with supplementary eye fields. *Journal of Neurophysiology*, *66*(2), 559–579. <https://doi.org/10.1152/jn.1991.66.2.559>
- Stangl, M., Shine, J., & Wolbers, T. (2017). The GridCAT: A Toolbox for Automated Analysis of Human Grid Cell Codes in fMRI. *Frontiers in Neuroinformatics*, *11*(47).

Rebuttal letter, Graichen et al.

- Swets, J. A. (1961). Detection Theory and Psychophysics: A Review. *Psychometrika*, 26(1), 49–63. Cambridge Core. <https://doi.org/10.1007/BF02289684>
- Van Kesteren, M. T. R., Ruiter, D. J., Fernández, G., & Henson, R. N. (2012). How schema and novelty augment memory formation. *Trends in Neurosciences*, 35(4), 211–219. <https://doi.org/10.1016/j.tins.2012.02.001>
- Vernet, M., Quentin, R., Chanes, L., Mitsumasu, A., & Valero-Cabré, A. (2014). Frontal eye field, where art thou? Anatomy, function, and non-invasive manipulation of frontal regions involved in eye movements and associated cognitive operations. *Frontiers in Integrative Neuroscience*, 8. <https://doi.org/10.3389/fnint.2014.00066>
- Wagner, I. C., Graichen, L. P., Todorova, B., Lüttig, A., Omer, D. B., Stangl, M., & Lamm, C. (2023). Entorhinal grid-like codes and time-locked network dynamics track others navigating through space. *Nature communications*, 14(1), 231.
- Wagner, I. C., Konrad, B. N., Schuster, P., Weisig, S., Repantis, D., Ohla, K., Kühn, S., Fernández, G., Steiger, A., & Lamm, C. (2021). Durable memories and efficient neural coding through mnemonic training using the method of loci. *Science advances*, 7(10), eabc7606.
- Wynn, J. S., Ryan, J. D., & Buchsbaum, B. R. (2020). Eye movements support behavioral pattern completion. *Proceedings of the National Academy of Sciences*, 117(11), 6246–6254. <https://doi.org/10.1073/pnas.1917586117>

RESPONSE TO REVIEWERS

Reviewer #1

I thank the authors for the revision.

The fact that saccade-based grid-like codes were specific to hits is a very interesting addition. Still the inverse correlation is puzzling, but possibly related to less time needed for well remembered items (see below). Overall the authors made a good effort to address all comments. I remain a bit put off by the use of non-significant results to “confirm” interpretation or assess trends. E.g., a $r_{\text{Pearson}} = -0.04$ with $p = 0.4$ is probably meaningless and should not be used to suggest trends (L256). L305 (a similar (albeit non-significant trend) in the Vienna study ($r = -.44$, $p = .056$;) on the other hand is a fair point. That said, I think the core findings are very interesting. The addition of hits vs misses improves the paper substantially. I also welcome the renaming of the studies, which wasn’t appropriate before (the verification study did not verify). My main remaining concern is that the stimulus set does not require relational processing, but I accept that the authors did not intend to test it (see some comments below). I think with a few small corrections this paper can be published.

A few additional comments:

Reviewer 1, comment 1

Abstract:

this is a nitpick but the sentence “Unexpectedly, saccade-based grid-like codes were associated with recognition memory, such that grid-like codes were lower the better participants performed in subsequently recognizing the scene images.” initially reads as if the association between grid codes and memory is unexpected, but it is the direction of the correlation that is unexpected. It is probably better to invert the order. E.g. “Unexpectedly, lower grid codes were associated with ...”. But this is up to authors as this is a very minor point.

Author reply:

We have rephrased the sentence to improve clarity.

Abstract (page 2, line 33-34):

“Unexpectedly, lower saccade-based grid-like codes were associated with better subsequent recognition memory.”

Reviewer 1, comment 2

Introduction:

Fig1A: nice to see alternative, similar images. Though I would maintain relational processing is not (or less) needed for such images.

Author reply:

While we agree with the reviewer that successful memory encoding might not have primarily relied on relational processing, we cannot rule out the possibility that it has played a role for some participants. We would like to highlight once more that it was not the focus of our work to test relational processing, and we addressed this point more explicitly in the revised discussion.

Discussion (page 12, line 365-370):

“While relational processing has been proposed as a mechanism by which visual grid cells may support memory (Bicanski & Burgess, 2019), this relationship was not clearly evident in our data. However, our studies were not specifically designed to test relational processing, and our stimulus set may have allowed for scene recognition based on visual properties, such as color, perspective, or global features. We thus encourage future work to interrogate the relationship between relational processing and saccade-based entorhinal grid-like codes during memory formation with dedicated paradigms.”

Reviewer 1, comment 3

Results:

L117: “the average number of saccades during scenes that were later remembered (7.95 ± 0.40) was significantly higher than during scenes that were later forgotten”

L174: “Results confirmed that the magnitude of grid-like codes was not related to saccade numbers”

But the magnitude of grid-like codes inversely correlates with d' ?

Author reply:

To briefly summarize, we replicated prior findings showing that higher saccade numbers during memory encoding were associated with better recognition memory performance across participants (Fehlmann et al., 2020; Loftus, 1972; Meister & Buffalo, 2016; Olsen et al., 2016). Next, we showed that the magnitude of grid-like coding was not significantly related to the number of saccades that were produced during encoding. This control analysis is important, as it demonstrates that the observed grid-like codes were not simply a byproduct of higher saccade numbers in some participants (i.e., participants who made more saccades naturally had more trajectories that went into the analysis of grid-like codes; we showed that this potential confound did not drive the results). Lastly, we found a negative association between grid-like coding and recognition memory performance, such that higher grid-like codes during scene viewing were associated with lower recognition memory performance at test. In other words, the relationship between saccade-based entorhinal grid-like codes and recognition memory performance was not confounded by the absolute number of saccades.

To speculate about the brain-behavior relationship, we had outlined several possible explanations in the manuscript: First, it is possible that successful memory performance may depend on specific eye movements, such as lower-velocity or shorter-amplitude saccades that were potentially associated with lower grid-like coding.; but note that we did not observe a significant relationship in our data. Second, some participants may have used alternative encoding strategies (e.g., leveraging prior knowledge) or may have encoded the scenes more efficiently (e.g., focusing on the defining, salient features). Previous findings showed that memory success following memory training is associated with reduced neural activity in brain areas implicated in memory and navigation (e.g., the hippocampus or the retrosplenial cortex; Wagner et al., 2021). Similarly, long-term practice in macaques has been linked to reduced glucose uptake, suggesting more efficient neural processing (Picard et al., 2013). Based on this, we speculate that efficient encoding might be related to lower saccade-based grid-like codes. Variability in gaze patterns or strategy use may account for the presence of grid-like codes during subsequently remembered trials (hits), while also explaining their negative association with overall memory performance. We would like to emphasize that the precise nature of the relationship between grid-like coding, eye movement characteristics, and memory formation remains an open question for future investigation.

Reviewer 1, comment 4

L 256: I remain a bit put off by the use of non-significant results to “confirm” interpretation or assess trends. E.g., a $r_{\text{Pearson}} = -0.04$ with $p = 0.4$ is probably meaningless and should not be used to suggest trends (L256). L305 (a similar (albeit non-significant trend) in the Vienna study ($r = -.44$, $p = .056$;) on the other hand is a fair point.

Author reply:

We completely agree with the reviewer and revised the relevant sections:

Results (page 9, line 249-253):

“For the immediate test, we observed no significant association between grid-like codes in the left entorhinal cortex and recognition memory performance (d-prime; $N = 20$, $r_{\text{Pearson}} = 0.01$, 95% CI = [-0.437, 0.448], $p_{\text{two-tailed}} = 0.98$; Figure 3D). A similar picture emerged for the delayed test (but note that the relationship appeared slightly negative; d-prime; $N = 20$, $r_{\text{Pearson}} = -0.04$, 95% CI = [-1, 0.4127], $p_{\text{one-tailed}} = 0.439$; see Supplementary Results S8 for an analysis of grid-like codes in relation to memory durability).”

Results (page 10, line 294-298):

“To examine the brain-behavior relationship specifically for successful memory encoding, we assessed whether the magnitude of grid-like codes during hits was associated with individual variations in recognition memory performance (d-prime). In both studies, grid-like codes tended to be higher in participants with lower d-prime, but the results were not significant (all $p > 0.05$; Figure 3C & 3F, left panels, Supplementary Information, Table S2).”

Supplementary Information (page 3, line 97-100):

“We first examined whether average fixation and revisitation frequencies were related to recognition memory performance (d-prime) across participants (Donders data set, $N = 32$), but this was not the case (Supplementary Fig. S7A; but note that results indicated a positive trend).”

Supplementary Information (page 5, line 180-184):

“A possible explanation for the negative brain-behavior result is that memory success was associated with differences in eye movements during encoding. For instance, participants with better recognition memory might have been able to memorize the scene images by making saccades with shorter durations, resulting in lower grid-like codes. While this was not the case, we found subtle trends of higher grid-like codes being associated with lower average saccade frequencies, velocities, and amplitudes (Supplementary Fig. S6).”

Supplementary Information (page 5, line 188-190):

"We could not detect a significant relationship between grid-like codes, fixations, revisitations, or recognition memory performance (Supplementary Fig. S7A; ~~individuals who made more fixations during encoding showed marginally lower grid-like codes, Supplementary Fig. S7B).~~"

Reviewer 1, comment 5

L270 and following: this is nice to see, and suggestive of at least some involvement of visual grid codes, despite the imperfect conditions to test of relational processing within images.

Fig3: in panel C it takes a while to find the "hit rate" label vs "d-prime" label due to the common plot title ("hits") and the density of the panels.

Author reply:

We thank the reviewer for bringing this to our attention. To improve readability, we reduced the panel density and increased the label sizes in the figure.

Reviewer 1, comment 6

L344: "the average number of fixations and returns to previously viewed locations (i.e., the revisitations)"

If this is supposed to be informative of memory, revisiting the same image location between study and test should be more informative of memory. Within a trial (as revisitations are defined in Kragel), is also interesting, but more of a short term effect. If revisiting the same locations at test and study correlated correct trials and with 6-fold mod, that would speak to memory involvement. Though of course pure bottom up visual processing could also attract saccades to the same locations at study and test.

Author reply:

We agree with the reviewer that saccades to the same image location during study and test (thus, revisitations) could provide additional evidence for memory-related visual sampling. However, as Kragel et al. (2021) showed, within-trial revisitations during encoding support subsequent recognition memory. This is an analysis that we had already performed as part of the previous round of revisions, but we could not confirm this relationship in our data. Importantly, as the reviewer states correctly, bottom-up visual processing could also attract saccades to the same locations at study and test, which is why revisitations during test are not a perfect indicator of memory.

Since the focus of our work is on saccade-based grid-like codes during memory encoding, we would prefer to stay within the scope of the paper and refrain from this additional analysis. We encourage future research to investigate this aspect, and we touch upon this topic in our updated discussion section.

Discussion (page 14, line 470-473):

"Future studies involving partial occlusions (e.g., Wynn et al., 2020) or an analysis of eye movement reinstatement (Johansson et al., 2022; where gaze during retrieval is returned to locations that were viewed during encoding) could help explain whether relational processing mediates the relationship between memory formation and grid-like codes."

Supplementary Information (page 5, line 186-194):

"Memory performance might also be influenced by the number of fixations or revisitations. The latter were previously linked to better recognition memory performance and may support thorough encoding by redirecting attention to specific elements of a scene (Kragel et al., 2021). We could not detect a significant relationship between grid-like codes, fixations, revisitations, or recognition memory performance (Supplementary Fig. S7A, Supplementary Fig. S7B). To further elucidate the relationship between grid-like codes and memory, future studies may examine the association with the reinstatement of eye movements during retrieval (Johansson et al., 2022). Overall, these findings reflect some inter-individual variability in visual exploration but did not appear to significantly contribute to the observed grid-like codes."

Reviewer 1, comment 7

Discussion:

L401: "While relational processing has been proposed as a mechanism by which visual grid cells may support memory (Bicanski & Burgess, 2019), we could not confirm this relationship..."

I think this also requires a brief, explicit mention of the stimulus set in the present study, which did not necessarily require relational processing. Also, is the correlation of grid signals with hits (even if negative) in principle compatible with that model? (see also L451 comment below)

Author reply:

We now clearly state that our stimulus set was not designed to elicit relational processing.

We still believe that the observed negative association between grid signals (during hits) and recognition memory performance can be reconciled with a relational processing account. It is possible that high-performing participants relied on relational processing when helpful (e.g., when the content was familiar) but otherwise explored the scenes guided by color, perspective, or global visual features (in line with reviewer comment 14). This may help explain why we observed grid-like coding during subsequently correctly recognized trials (hits), even though higher overall grid-like coding was linked to lower recognition memory performance. In line with the reviewer's suggestion (comment 10), lower grid-like coding in participants with better memory performance may also reflect more effective suppression of alternative saccade targets, resulting in lower overall activity. We have now included this interpretation in the revised discussion.

Discussion (page 12, line 365-370):

"While relational processing has been proposed as a mechanism by which visual grid cells may support memory (Bicanski & Burgess, 2019), this relationship was not clearly evident in our data. However, our studies were not specifically designed to test relational processing, and our stimulus may have allowed for scene recognition based on visual properties such as color, perspective, or global features. We thus encourage future work to interrogate the relationship between relational processing and saccade-based entorhinal grid-like codes during memory formation with dedicated paradigms."

Discussion (page 13, line 413-420):

"Similarly, high-performing participants may have explored the scenes more selectively, suppressing saccades toward visual features that were less informative for subsequent recall (Geng, 2014; Henderson & Hayes, 2017) and resulting in lower grid-like codes (but this stands in contrast to our observation that participants who made more saccades recognized the scenes better; see the Supplementary Discussion for a more detailed account of grid-like codes in relation to saccade characteristics and visual exploration patterns)." Overall, the observed negative brain-behavior relationship challenges the idea that higher grid-like codes universally support memory formation."

Reviewer 1, comment 8

L432: see comment above about revisitations

Author reply:

The reviewer touches on a similar topic as in comment 6, and we revised the discussion accordingly:

Discussion (page 14, line 470-473):

"Future studies involving partial occlusions (e.g., Wynn et al., 2020) or an analysis of eye movement reinstatement (Johansson et al., 2022; where gaze during retrieval is returned to locations that were viewed during encoding) could help explain whether relational processing mediates the relationship between memory formation and grid-like codes."

Supplementary Information (page 5, line 186-194):

"Memory performance might also be influenced by the number of fixations or revisitations. The latter were previously linked to better recognition memory performance and may support thorough encoding by redirecting attention to specific elements of a scene (Kragel et al., 2021). We could not detect a significant relationship between grid-like codes, fixations, revisitations, or recognition memory performance (Supplementary Fig. S7A, Supplementary Fig. S7B). To further elucidate the relationship between grid-like codes and memory, future studies may examine the association with the reinstatement of eye movements during retrieval (Johansson et al., 2022). Overall, these findings reflect some inter-individual variability in visual exploration but did not appear to significantly contribute to the observed grid-like codes."

Reviewer 1, comment 9

L443: I don't think that model speaks about schemas. Although if grid cells generalize across scenes that may help.

Author reply:

We thank the reviewer for pointing this out. Our intention was not to suggest that the model by Bicanski & Burgess (2019) explicitly addresses schemas. To avoid any further confusion, we have removed the citation from the sentence (not shown here).

Reviewer 1, comment 10

L451 and following: the efficiency argument is an interesting candidate. The authors could also mention that more effective suppression of alternative saccade targets (for well-remembered items) may explain lower activations.

Author reply:

We agree that more effective suppression of alternative saccade targets could offer a plausible explanation for the observed negative relationship between grid-like codes during encoding and recognition memory performance, although this interpretation stands in contrast to our finding that higher saccade numbers were associated with better memory performance. Nevertheless, we have added this point to the discussion.

Discussion (page 13, line 413-418):

"Similarly, high-performing participants may have explored the scenes more selectively, suppressing saccades toward visual features that were less informative for subsequent recall (Geng, 2014; Henderson & Hayes, 2017) and resulting in lower grid-like codes (but this stands in contrast to our observation that participants who made more saccades recognized the scenes better; see the Supplementary Discussion for a more detailed account of grid-like codes in relation to saccade characteristics and visual exploration patterns)."

Reviewer 1, comment 11

L467 ...: I guess rather than less effective, it would depend on whether or not the task taxes relational processing at encoding AND recall. Relational encoding may of course also be incidental at first, and relied upon later when needed. I would suspect this to be highly task-dependent.

Author reply:

We agree with the reviewer that the original phrasing was suboptimal. Our intention was to convey that tasks requiring little relational processing (e.g., due to a lack of familiar content) may engage grid-like coding less. We rephrased the sentence accordingly to clarify this point.

Discussion (page 13, line 421-425):

"Bicanski and Burgess (2019) argue that grid-like codes mainly contribute to relational memory for familiar content, suggesting they may be less involved in directing gaze when encoding previously unknown scenes. Depending on prior knowledge and task demands, we propose that increased grid-like codes may hinder or support memory formation, which is an interesting aspect warranting future interrogation."

Reviewer 1, comment 12

L481: "Unlike grid-like codes, however, frontal eye field activity showed no clear association with memory performance." Is this missing in the FEF results section?

Author reply:

We had not included this finding initially because the association was not significant. We now added this for completeness.

Results (page 11, line 327-335):

"Results from the whole-brain analysis showed that, as participants viewed scene images, increased saccade-based grid-like codes were coupled to increased activation specifically in the left FEF. In other words, if participants made saccades that were aligned with their individual entorhinal grid orientation (i.e., resulting in larger grid-like codes for that trial), BOLD activation in the left FEF was high as well ($p < 0.05$ FWE-corrected at cluster level using a cluster-defining threshold of $p < 0.001$, cluster size = 116 voxels; $x = 55$, $y = 22$, $z = 31$; Figure 4; please note that we only performed this analysis for the Donders study, as the partial field-of-view in the Vienna study did not allow for reliable FEF coverage). FEF activation was neither significantly tied to the average saccade numbers during encoding ($p > 0.05$) nor related to recognition memory performance across participants (d -prime, $p > 0.05$)."

Reviewer 1, comment 13

Comments on the reply to referees (no line numbers there)

The authors state: "To reiterate, we found that fMRI BOLD activity in the frontal eye fields (FEF) was increased when saccade-based grid-like codes in the entorhinal cortex were increased as well. This suggests that the activity of saccade-based entorhinal grid-like codes was time-locked to activity changes in oculomotor regions as individuals visually explored new content."

"We therefore politely disagree with the reviewer" I politely accept the correction. The new Discussion text is neutral enough here. I am puzzled though, if d -prime correlates negatively with grid signals, and grid signals correlate positively with FEF, how can d -prime correlate positively with the FEF? I was going to say this suggests the need for a joint statistical treatment. However, the authors state: "This was, contrary to the reviewer's comment, not the case (r Pearson = .05, p two-tailed = .80)." I want to try to discourage the authors from arguing based on a 0.05 correlation that is not significant.

Author reply:

We apologize and did not mean to draw strong conclusions based on the absence of a significant correlation – in fact, our intention was to clarify that no such association was observed in our data. Please also note that the correlations in question [grid-like codes x frontal-eye-field (FEF) activity vs. grid-like codes/FEF activity x *d*-prime] were based on the trial- vs. across-participant-level, respectively. These results are hence not directly comparable, which could explain the diverging results. We made sure to rephrase this very carefully.

Results (page 11, line 333-335):

“FEF activation was neither significantly tied to the average saccade numbers during encoding ($p > 0.05$) nor related to recognition memory performance across participants (*d*-prime, $p > 0.05$).”

Reviewer 1, comment 14

On relational processing. I see, it was my misunderstanding then that I assumed relational processing and not just establishing whether grid-like codes play some un-defined role in memory formation. However, targeting a possible mechanism: are there other suggestions besides relational processing? What should engage the network of interest more? Or is relational processing still our best guess for entorhinal grid codes? In either case, I very much welcome the revision of Figure 1 to show more similar stimuli, though one could argue they are still easily discernible based on color, perspective, and global features. The discussion should reflect this a bit more clearly.

The point I was making in my original review is that if entorhinal cortex performs relational processing in general, and specific to vision possibly in the field of view via visual grid cells, then these stimuli are suboptimal, which of course does not necessarily prevent grid signals, as the Buffalo study with covert attention shows (Wilming et al). This is important to note.

Author reply:

We thank the reviewer for this thoughtful comment. Our aim was to determine whether saccade-based grid-like codes in the entorhinal cortex are involved in memory formation, rather than to isolate a specific underlying mechanism. We agree that relational processing is currently the most compelling candidate for how grid-like codes might support visual memory.

Grid-like codes are thought to signal specific positions in space and their relation to each other. In the visual domain, this could help establish a spatial framework of a viewed scene based on successively visited positions. Such a framework could support later recognition by guiding the eye gaze back towards previously sampled positions. The fact that increased grid-like coding was associated with increased frontal eye field activation in our data further supports this idea. Beyond relational processing, we are not aware of a well-supported alternative account in the literature that could explain the link between grid-like codes and memory formation. As mentioned by the reviewer, prior work in monkeys detected entorhinal grid-like activity during covert shifts of attention that was unrelated to eye movements (Wilming et al., 2018). While this does not imply a different mechanism linking grid-like activity and memory formation, it shows that grid-like codes can be observed in the absence of overt visual sampling.

We now make this explicit in the discussion and acknowledge that our stimulus set was not designed to specifically engage relational processing, but that the stimuli were still visually separable based on color, perspective, and global features.

Discussion (page 12, line 365-370):

“While relational processing has been proposed as a mechanism by which visual grid cells may support memory (Bicanski & Burgess, 2019), this relationship was not clearly evident in our data. However, our studies were not specifically designed to test relational processing, and our stimulus may have allowed for scene recognition based on visual properties such as color, perspective, or global features. We thus encourage future work to interrogate the relationship between relational processing and saccade-based entorhinal grid-like codes during memory formation with dedicated paradigms.”

Reviewer 1, comment 15

“Additional analysis of revisitations:

Nevertheless, we performed the additional analyses with regard to whether individuals produced saccades that returned to a previously fixated location (revisitations, please see above for a detailed outline of the methodology). In essence, we did not find a significant correlation between the average number of revisitations per trial and recognition memory performance across participants ...”

Rebuttal letter, Graichen et al.

See similar comment above. If we are looking for long term memory driven effects, revisitations should occur between study and test. Within a trial is less informative. Within a trial a salient part of an image might just as well attract attention repeatedly.

Author reply:

We agree with the reviewer that revisitations occurring between study and test can be informative of memory performance. However, revisitations within a trial have also been associated with successful memory formation (Kragel et al., 2021). While some revisitations (both within-trial and between study and test) may be saliency-driven (as the reviewer also states in comment 6), they can also reflect systematic sampling strategies that support memory (Johansson et al., 2022). We encourage future research to study this relationship in more detail and have incorporated this into our updated discussion.

Reviewer 1, comment 16

Regarding partial occlusions, that was an example, I was not suggesting the authors perform additional experiments. But the authors nicely cover this in the Discussion now.

A clarification regarding the FEF. I do not suggest that “FEF are equally active for all eye-movement vectors”. Of course there are other determinants of eye-movements. However, for those eye-movements primarily instructed by FEF+grids, FEF activity should not depend on the direction of the saccade relative to the grid orientation. This is an odd suggestion because there is no a priori obligation for useful saccade targets to fall on grid axes.

Author reply:

To clarify our original point, we found that FEF activity positively scaled with the overall magnitude of grid-like coding during a given trial (i.e., the extent to which saccades across a 4-second period were, on average, aligned with the underlying grid orientation). We interpreted this as evidence for FEF involvement in generating saccades that are also tied to grid-like codes in the entorhinal cortex, not as reflecting a directional preference per se. In other words, the alignment of a single saccade with the underlying grid orientation may be coincidental and is unlikely to affect FEF activity. However, if multiple saccades within a trial align with the grid orientation, this may indicate that saccades are systematically organized in a grid-like manner, potentially supporting scene encoding and related to increased activation in the FEF.

The hits vs misses analysis is interesting and a nice addition.

The explanation for lack of FEF analysis in the second study is satisfactory

Overall, I thank the referees for the great effort in addressing my comments.

Reviewer #2

I appreciate the authors' comprehensive responses to my previous concerns; the resulting manuscript has greater transparency and clarity, and is, consequently, much improved.

We thank the reviewers very much for taking the time to provide thoughtful and constructive feedback and for recognizing the work invested in preparing and revising this manuscript!

References

- Bicanski, A., & Burgess, N. (2019). A Computational Model of Visual Recognition Memory via Grid Cells. *Current Biology*, 29(6), 979-990.e4. <https://doi.org/10.1016/J.CUB.2019.01.077>
- Fehlmann, B., Coynel, D., Schicklitz, N., Milnik, A., Gschwind, L., Hofmann, P., Papassotiropoulos, A., & De Quervain, D. J.-F. (2020). Visual Exploration at Higher Fixation Frequency Increases Subsequent Memory Recall. *Cerebral Cortex Communications*, 1(1), tga032. <https://doi.org/10.1093/texcom/tga032>
- Johansson, R., Nyström, M., Dewhurst, R., & Johansson, M. (2022). Eye-movement replay supports episodic remembering. *Proceedings of the Royal Society B*, 289(1977), 20220964.
- Kragel, J. E., Schuele, S., VanHaerents, S., Rosenow, J. M., & Voss, J. L. (2021). Rapid coordination of effective learning by the human hippocampus. *SCIENCE ADVANCES*.
- Loftus, G. R. (1972). Eye fixations and recognition memory for pictures. *Cognitive psychology*, 3(4), 525–551.
- Meister, M. L. R., & Buffalo, E. A. (2016). Getting directions from the hippocampus: The neural connection between looking and memory. *Neurobiology of Learning and Memory*, 134, 135–144. <https://doi.org/10.1016/j.nlm.2015.12.004>
- Olsen, R. K., Sebanayagam, V., Lee, Y., Moscovitch, M., Grady, C. L., Rosenbaum, R. S., & Ryan, J. D. (2016). The relationship between eye movements and subsequent recognition: Evidence from individual differences and amnesia. *Cortex*, 85, 182–193. <https://doi.org/10.1016/j.cortex.2016.10.007>
- Picard, N., Matsuzaka, Y., & Strick, P. L. (2013). Extended practice of a motor skill is associated with reduced metabolic activity in M1. *Nature Neuroscience*, 16(9), 1340–1347. <https://doi.org/10.1038/nn.3477>
- Wagner, I. C., Konrad, B. N., Schuster, P., Weisig, S., Repantis, D., Ohla, K., Kühn, S., Fernández, G., Steiger, A., & Lamm, C. (2021). Durable memories and efficient neural coding through mnemonic training using the method of loci. *Science advances*, 7(10), eabc7606.
- Wilming, N., König, P., König, S., & Buffalo, E. A. (2018). Entorhinal cortex receptive fields are modulated by spatial attention, even without movement. *eLife*, 7, e31745. <https://doi.org/10.7554/eLife.31745>